# Telomere attrition becomes an instrument for clonal selection in aging hematopoiesis and leukemogenesis

The mechanisms through which mutations in splicing factor genes drive clonal hematopoiesis (CH) and myeloid malignancies, and their close association with advanced age, remain poorly understood. Here we show that telomere maintenance plays an important role in this phenomenon. First, by studying 454,098 UK Biobank participants, we find that, unlike most CH subtypes, splicing-factor-mutant CH is more common in those with shorter genetically predicted telomeres, as is CH with mutations in *PPM1D* and the *TERT* gene promoter. We go on to show that telomere attrition becomes an instrument for clonal selection in advanced age, with splicing factor mutations 'rescuing' HSCs from critical telomere shortening. Our findings expose the lifelong influence of telomere maintenance on hematopoiesis and identify a potential shared mechanism through which different splicing factor mutations drive leukemogenesis. Understanding the mechanistic basis of these observations can open new therapeutic avenues against splicing-factor-mutant CH and hematological or other cancers.

Advancing age is associated with the development of somatic-mutation-driven clonal expansions in most tissues[1]. In blood, somatic mutations that augment cellular fitness of individual hematopoietic stem cells (HSCs) give rise to CH[2–4]—a common phenomenon associated with an increased risk of hematological cancers and some nonhematological conditions[2,3,5,6]. Mutations in the epigenetic regulator genes *DNMT3A*, *TET2* and *ASXL1* are responsible for ~70% of CH cases, with most of the remaining cases driven by mutations in genes involved in RNA splicing (*SF3B1*, *SRSF2* and *U2AF1*), DNA damage response (DDR) (*TP53* and *PPM1D*), cytokine signaling (*JAK2*) and G-protein signaling (*GNB1* and *GNAS*)[2,3,5].

Despite substantial progress in understanding the natural history of CH[7,8], and identifying many of its causes and consequences[5,9,10], our understanding of its pathogenesis remains limited. An important insight into CH pathogenesis came from the identification of a significant association between polymorphisms at the telomerase reverse transcriptase (*TERT*) locus and CH risk[9,11], with subsequent Mendelian randomization (MR) analyses supporting a causal relationship[5,12]. Similarly, people with a monogenic long telomere syndrome associated with *POT1* mutations displayed a very high prevalence of CH[13]. These findings indicate that normal replication-associated telomere shortening can act to curtail HSC clonal expansion, such that the inheritance of longer telomeres or an enhanced ability to maintain telomere length in HSCs favors CH development[14].

Other insights came from the study of mutations driving CH in particular contexts; such as in autoimmune aplastic anemia, where CH is commonly driven by mutations in *PIGA* or the *HLA* locus, ostensibly because these mutations enable HSCs to evade the immunological attack against them[15]. Similarly, *TP53*-mutant and *PPM1D*-mutant CH arises commonly after cytotoxic chemotherapy, to which these mutations confer HSC chemoresistance by attenuating the DDR[16,17].

We reported previously that *SF3B1* and *SRSF2* mutations cause CH specifically in the elderly[4], suggesting that advanced age provides a specific context within which these mutations confer a clonal advantage[18]. Here we discover that telomere attrition provides such a 'context' and becomes an instrument for clonal selection specifically in advanced age. Our findings demonstrate that mutations in splicing factor genes promote clonal expansion by preventing critical telomere shortening in aged HSCs and that *PPM1D* mutations may

✉e-mail: gsv20@cam.ac.uk

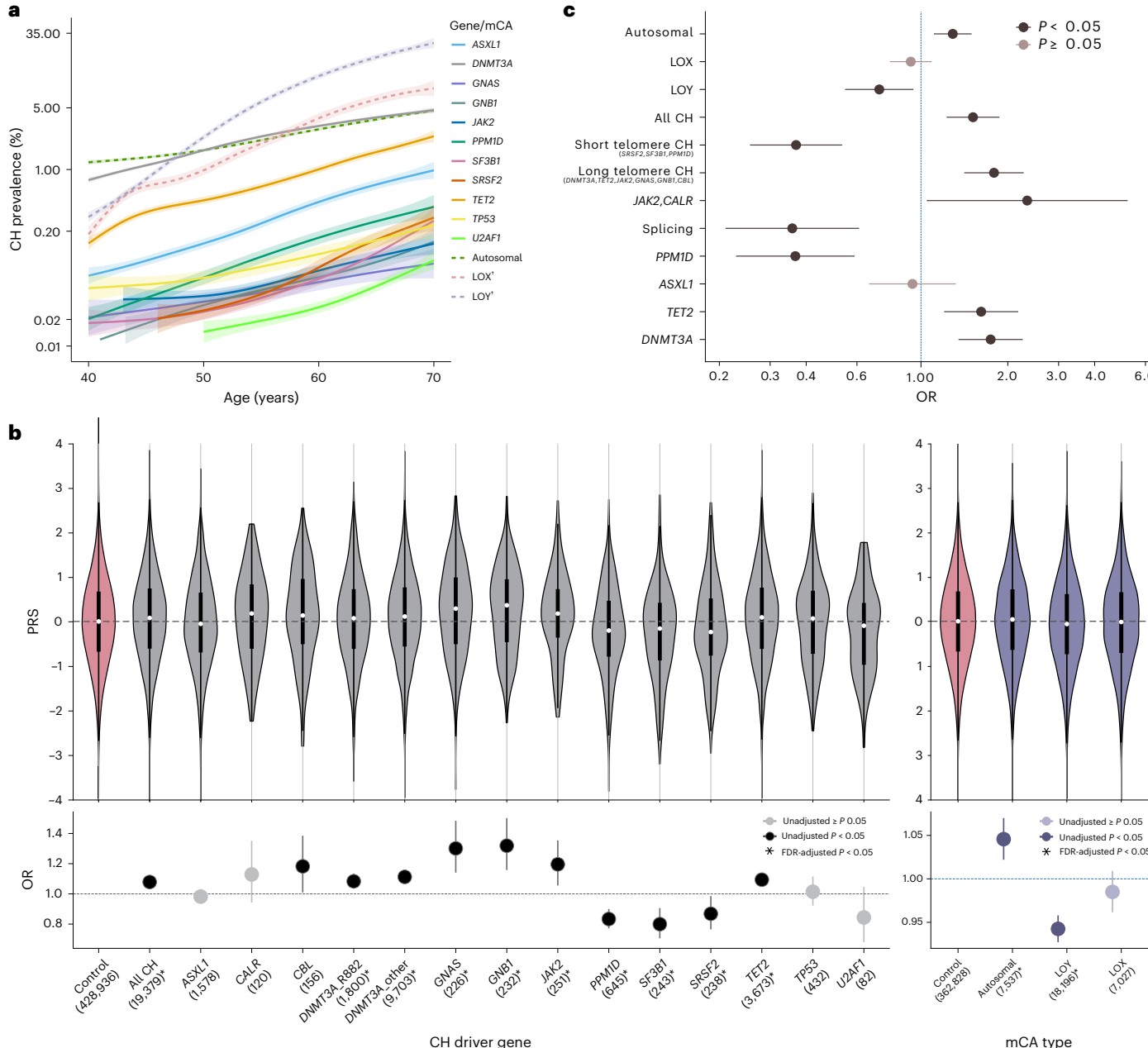

**Fig. 1 | Genetically determined variation in telomere length and CH subtype.**
**a**, Age-related prevalence of CH subtype by driver gene (prevalence shown on log scale). Colored lines depict the smoothed model fitted to a generalized additive model with 95% CIs represented by colored shadows. Dagger (†), prevalence is shown in male and female participants respectively for LOY and LOX mCA. **b**, Top, violin plots of genetically determined LTL-PRS among people with different CH (gray) and mCA (purple) subtypes. White dots and black boxes mark the LTL-PRS median and interquartile range (IQR), respectively. Whiskers extend to the lowest and highest datapoints within Q1 − 1.5 × IQR and Q3 + 1.5 × IQR where Q1 and Q3 represent the first and third quartiles, respectively. The control group (pink) includes all participants without any CH mutation or mCA. Bottom, odds ratio (OR) of developing different CH/mCA subtypes per 1 s.d. increase in LTL-PRS. Dots represent the estimated OR and error bars represent

the 95% CI, both derived from a logistic regression model. ORs for genes with an FDR-adjusted $P < 0.05$ are indicated with an asterisk (*). FDR adjustment was performed using the Benjamini−Hochberg procedure. Number of mutation carriers is indicated in brackets along with the gene names. The y axis is limited to the range [−4,4] to allow for better visualization of PRS differences. **c**, Results of MR showing the OR of developing the depicted subtype of CH per 1 s.d. increase in LTL. Dots represent the estimated OR and error bars represent the 95% CI for each OR. 'Long Telomere CH' refers to CH mutations that are more common among people with longer genetically predicted telomere length (PRS) in **b**, namely *DNMT3A*, *JAK2*, *TET2*, *GNAS*, *GNB1* and *CBL* and 'Short Telomere CH' refers to CH mutations that are more common among people with shorter genetically predicted telomere length (PRS), namely *PPM1D*, *SF3B1* and *SRSF2*.

do so by reducing DDR signaling from short telomeres. This study exposes age-associated telomere attrition as a conduit between aging and leukemogenesis. Although future studies are still needed to delineate the molecular basis of these observations, this work may propose new therapeutic approaches against CH and hematological or other malignancies.

## Results

### Associations between telomere length and subtypes of CH
We noted previously that the age-related prevalence of splicing-factor-mutant CH (SF-CH) differs from that of other CH subtypes[5,18]. To corroborate this in a large cohort, we investigated CH prevalence amongst 454,098 UK Biobank (UKB) participants with CH identified as

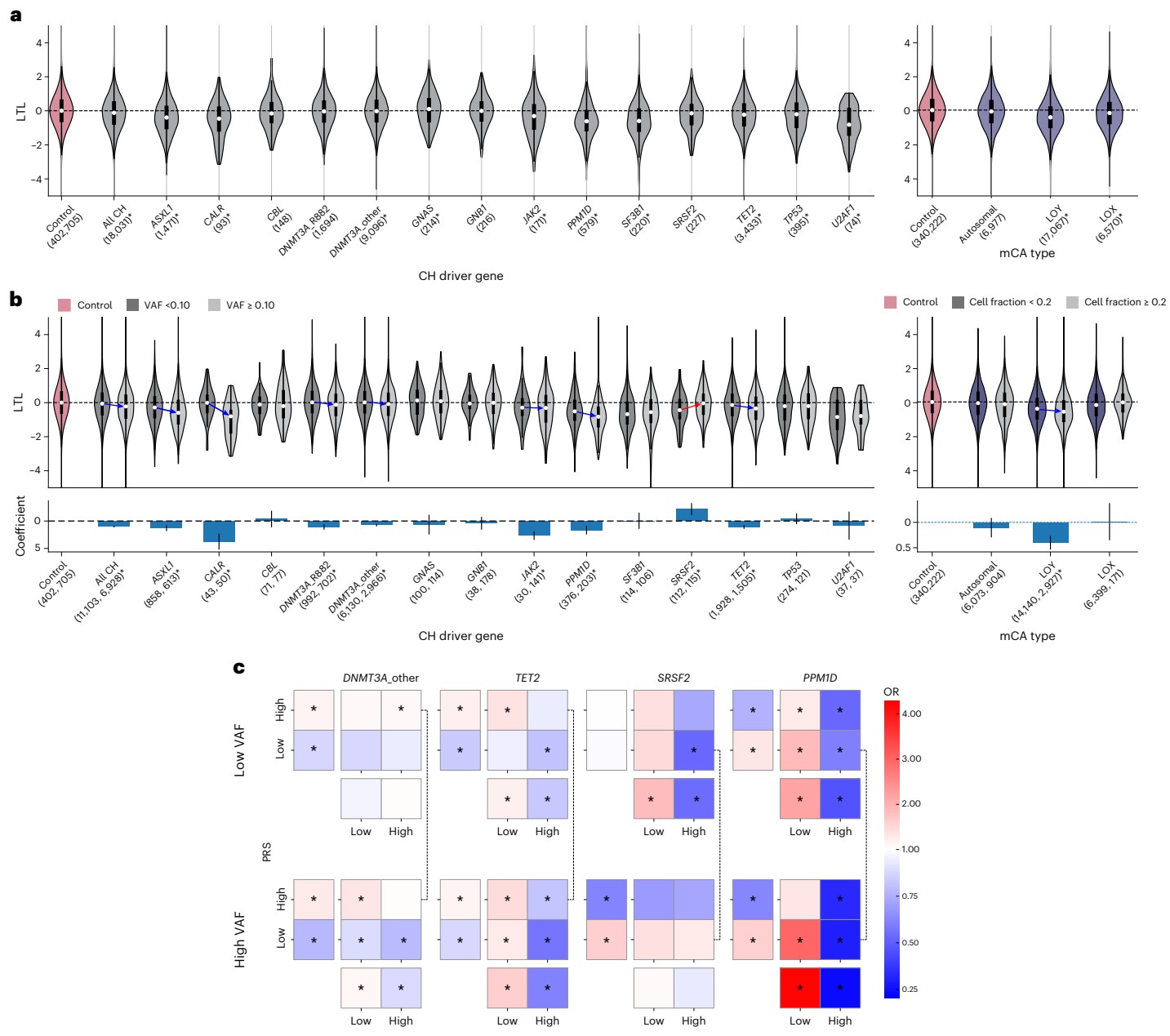

**Fig. 2 | Variation in measured telomere length by CH subtype and clonal size.**
**a**, Variation in measured LTL by CH (gray) or mCA (purple) subtype. White dots and black boxes mark the LTL median and IQR, respectively. For more details on boxplot representation, see Fig. 1. Genes with significantly different LTL (FDR-adjusted $P < 0.05$, Benjamini–Hochberg procedure) compared to controls (pink) are marked by an asterisk in the label, with $P$ values derived from a linear regression model. The $y$ axis is limited to the range [−5,5] for better visualization of LTL differences. **b**, Top, impact of clonal size on the measured LTL across CH/mCA subtypes. Participants in each CH/mCA subgroup were divided into those with small clones (VAF < 0.1 for CH or cell fraction <0.2 for mCA, dark gray/violet) and those with large clones (VAF ≥ 0.1 for CH or cell fraction ≥0.2 for mCA, light gray/violet); data presented as in **a**. Bottom, estimated effects of clone size on LTL. The bars represent the coefficients of CH subtype-specific VAF predictors from a linear regression model of LTL and error bars represent 95% CI. Coefficients that are significantly different from zero (FDR-adjusted

$P < 0.05$, Benjamini–Hochberg procedure) are marked with an asterisk and the corresponding genes are highlighted using colored arrows in the top panel. In **a** and **b**, the number of mutation carriers is shown in brackets below gene names (with separate counts for low and high VAF groups in **b**). For details of boxplot representation, see Fig. 1. **c**, Trends in the association between LTL-PRS and measured LTL among people with *DNMT3A*-other-CH, *TET2*-CH, *SRSF2*-CH and *PPM1D*-CH. Both LTL-PRS and measured LTLs were divided into high/low groups using the median value as cutoff, such that participants were categorized into four quadrants based on high/low status of each. The ORs of finding the specific driver gene mutation in each quadrant are depicted using a color scale. Separate ORs for measured LTL and LTL-PRS groups are shown below and to the left of each square, respectively. The analysis was done by categorizing mutations as low (VAF < 0.1) and high VAF (VAF ≥ 0.1). Significant ORs are indicated by asterisks ($P$ value < 0.05, two-sided Fisher's exact test). Notable differences in ORs between low and high VAF are indicated by connecting the relevant quadrants.

described previously[19]. This confirmed that, unlike other types of CH whose prevalence rose steadily with advancing age, SF-CH remained rare until the age of ~55 years and increased sharply in prevalence beyond that (Fig. 1a).

Noting recent observations that CH driven by splicing factor mutations can arise in young patients with telomere biology disorders (TBD)[20–23], we posited that telomere maintenance may underlie the unusual age distribution of sporadic SF-CH. To investigate this, we

examined the association between previously derived polygenic risk scores (PRS) for leukocyte telomere length (LTL)[5,24] and different CH subtypes. This confirmed the reported association of higher LTL-PRS (predictive of longer LTL) with increased prevalence of several common CH subtypes such as those driven by *DNMT3A*, *TET2* and *JAK2* mutations, and with CH associated with autosomal mosaic chromosome alterations (mCAs) (Fig. 1b and Supplementary Table 1). However, CH driven by mutations in *SF3B1*, *SRSF2* and *PPM1D* was associated with lower LTL-PRS—an association that was not observed in previous studies grouping these genes with other, more common, CH subtypes. A similar trend was seen with *U2AF1*-CH, but did not reach statistical significance (false discovery rate (FDR)-adjusted $P = 0.15$). Consistent with previous observations, CH associated with loss-of-Y (LOY) but not loss-of-X (LOX) chromosome was associated with lower LTL-PRS[25] (Fig. 1b). We also found that high PRS was associated with an increased risk of myeloproliferative neoplasm (MPN), but not myelodysplastic syndrome (MDS) where splicing gene mutations are very common (Extended Data Fig. 1 and Supplementary Table 2).

MR analyses confirmed the reported associations of longer genetically determined telomere length with CH driven by mutant *DNMT3A*, *TET2* or *JAK2* and with autosomal mCAs[5,12]. MR analyses also identified causal associations between shorter genetically determined telomere length and CH due to mutations in *PPM1D*, splicing factor genes (*SRSF2*, *SF3B1* and *U2AF1*) and mosaic LOY (Fig. 1c and Supplementary Table 3). In search of additional evidence for a causative association, we investigated the impact of another heritable influence on telomere length—paternal age[26]—on CH prevalence amongst 101,340 UKB participants for whom paternal age data are available. Logistic regression analysis was performed after grouping CH genes associated with longer telomere length in one group ('Long Telomere CH') and those associated with shorter telomeres in another ('Short Telomere CH'). 'Long Telomere CH' was associated significantly with paternal age, whereas 'Short Telomere CH' showed an opposite trend, although this did not reach statistical significance, reflecting the markedly smaller numbers in this group (Extended Data Fig. 2 and Supplementary Table 4).

To validate our observations regarding LTL-PRS and CH in an external cohort, we applied the UKB-derived LTL-PRS to participants in the All of Us cohort ($n = 133,656$) and examined its association with each CH subtype (Extended Data Fig. 3 and Supplementary Table 5). This revealed significantly higher LTL-PRS values in *DNMT3A*-CH and *CBL*-CH, as well as lower LTL-PRS values in *PPM1D*-CH and *U2AF1*-CH. However, we did not observe differences in LTL-PRS in *SF3B1*-CH and *SRSF2*-CH in the All of Us dataset, potentially owing to the substantially lower numbers of splicing factor-CH (SF-CH) cases in this smaller cohort. Furthermore, we speculated that an LTL-PRS derived using the predominantly European ancestry of the UKB may reflect LTL less accurately in a more genetically diverse cohort, and vice versa. To

test this, we re-analyzed the UKB cohort using an alternative LTL-PRS derived from TOPMed[27]—a cohort with a much higher proportion of people with non-European genetic ancestry (Extended Data Fig. 4 and Supplementary Table 6). This retained the significant associations between LTL-PRS and *DNMT3A*-CH, *TET2*-CH, *JAK2*-CH, *PPM1D*-CH and *SF3B1*-CH. Taken together, analyses using the All of Us and TOPMed datasets replicate many of our core observations, taking into account the limitations imposed by the substantially smaller number of SF-CH cases and more diverse genetic ancestries in these cohorts.

## Variation in LTL by CH subtype

Next, we investigated how the associations identified by LTL-PRS and MR analyses relate to measured LTL (derived previously by quantitative PCR (qPCR)[26]) in the UKB. After accounting for relevant covariates, we found marked differences in LTL between participants with CH and controls (Fig. 2a and Supplementary Table 7). First, as reported before[12], we found that most CH subtypes associated with higher LTL-PRS were not associated with longer measured LTLs and could even display significantly shorter LTL than controls (for example, *JAK2*-CH). In contrast, despite its association with lower LTL-PRS, *SRSF2*-CH was not associated with shorter measured LTL (Fig. 2a).

To better understand these observations, we separated carriers of each CH subtype based on variant allele fraction (VAF) into those with small (VAF < 0.1, 'low-VAF') and those with large (VAF ≥ 0.1, 'high-VAF') CH clones. This was done to capture the LTL of samples composed primarily of non-CH cells (VAF < 0.1) and contrast this with the LTL of samples with higher proportions of CH cells (VAF ≥ 0.1). This revealed that, for most CH subtypes, including *DNMT3A*-CH, *TET2*-CH, *ASXL1*-CH, *JAK2*-CH and *CALR*-CH, low-VAF samples had LTLs close to controls without CH whereas high-VAF samples had shorter LTLs, presumably reflecting telomere attrition during clonal expansion (Fig. 2b). By contrast, low-VAF samples from *PPM1D*-CH, *SF3B1*-CH and *SRSF2*-CH, as well as LOY, (that is, subtypes associated with lower LTL-PRS) had significantly shorter LTLs than controls (Fig. 2b). High-VAF *PPM1D*-CH and LOY samples had shorter LTLs than low-VAF ones, although this was not the case for *SF3B1* and *SRSF2*-CH. In fact, *SRSF2*-CH high-VAF samples had significantly longer LTLs than low-VAF samples (Fig. 2b, top and Supplementary Table 8). In some CH subtypes (*CALR*, *DNMT3A*_R882, *DNMT3A*_other, *JAK2* and *SRSF2*) these effects were driven largely by large clones, whereas other CH subtypes (*ASXL1*, *PPM1D* and *TET2*) showed a more dose-dependent association between VAF and measured LTL (Supplementary Fig. 1 and Supplementary Table 9).

We next used a linear regression model to examine the relationship between clone size (VAF) and LTL after accounting for factors known to affect telomere length, including age, sex, smoking and genetic ancestry (Methods). This revealed that model coefficients for VAF were negative for several common CH driver genes (including *DNMT3A*,

**Fig. 3 | Splicing factor mutations and telomere length of single-HSPC-derived colonies. a**, Ultrametric hematopoietic phylogeny from an 83.8 year old man with splicing-factor-mutant CH. Colonies belonging to *SF3B1*-mutant clades are colored in green, whilst colonies belonging to the *U2AF1*-mutant clade are colored in blue. Corresponding WGS-estimated telomere length for each single-HSPC-derived colony is shown on the right panel. **b**, Pairwise comparisons of WGS-estimated telomere length performed between clades using two-sided Wilcoxon rank sum test without adjustment for multiple comparisons. Independent colonies were used as biological replicates within each individual participant and the number of colonies within each group is shown in brackets below their respective label. Center line represents the median telomere length, upper and lower hinges represent the upper and lower quartiles, respectively, and whiskers represent 1.5× the IQR. Individual datapoints corresponding to telomere lengths of single-HSPC-derived colonies have been overlaid on each plot. **c**, Ultrametric hematopoietic phylogeny from an 73.9 year old woman with *SF3B1*-mutant CH. Colonies belonging to *SF3B1*-mutant clades are colored in green. Corresponding WGS-estimated telomere length for each single-HSPC-

derived colony is shown on the right panel. **d**, Pairwise comparisons of WGS-estimated telomere length performed between clades using two-sided Wilcoxon rank sum test without adjustment for multiple comparisons. Independent colonies were used as biological replicates within each individual participant and the number of colonies within each group is shown in brackets below their respective label. For details of box plot representation, see **b**. **e**, Ultrametric hematopoietic phylogeny from man with *SF3B1*-mutant CCUS/MDS derived from heterochronous samples obtained at 50.2 (CCUS) and 53.8 years of age (progression to MDS). Colonies belonging to *SF3B1*-mutant clades are colored in green. Corresponding WGS-estimated telomere length for each single-HSPC-derived colony is shown on the right panel. **f**, Pairwise comparisons of WGS-estimated telomere length performed within the *SF3B1*-mutant (green) and wild type (red) clades using two-sided Wilcoxon rank sum test without adjustment for multiple comparisons. Independent colonies were used as biological replicates within each individual participant and the number of colonies within each group is shown in brackets below their respective label. For details of box plot representation, see **b**.

*TET2*, *ASXL1*, *JAK2* and *PPM1D*), but positive for *SRSF2*-CH (Fig. 2b and Supplementary Table 8).

To examine the interaction between rising VAF, LTL and LTL-PRS more closely, we divided UKB participants with *DNMT3A*-CH, *TET2*-CH, *SRSF2*-CH and *PPM1D*-CH into upper and lower halves with respect to

measured LTL versus LTL-PRS (Fig. 2c). This confirmed that *DNMT3A*-CH and *TET2*-CH are more common among those in the upper LTL-PRS group, whereas *SRSF2*-CH and *PPM1D*-CH are more common amongst those in the lower LTL-PRS group. Also, by contrasting low-VAF versus high-VAF CH, it becomes evident that a rise in VAF is associated with a

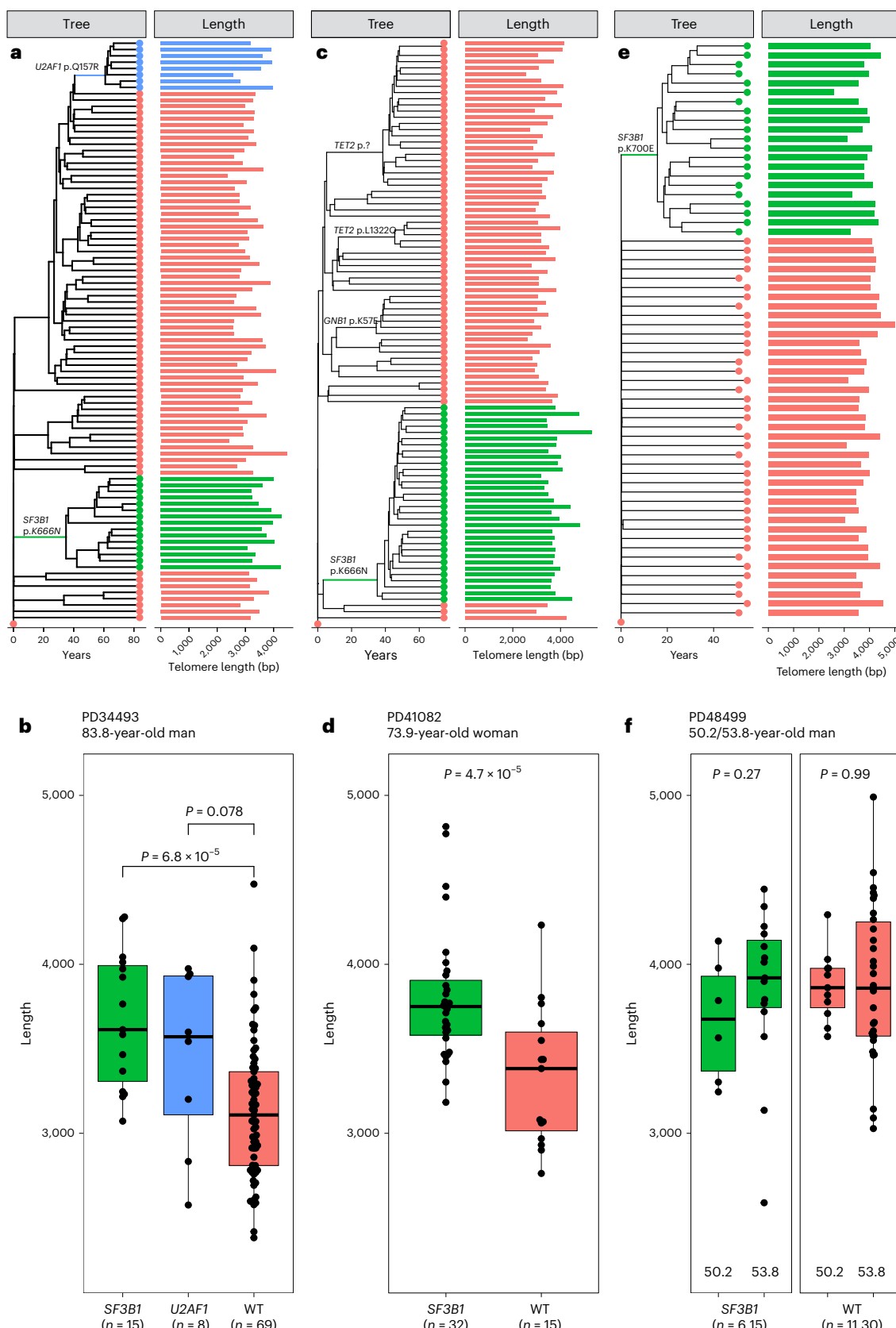

**Table 1 | Association between colony telomere length, driver mutations and age**

| Term | Estimate | Lower | Upper | Confidence |
|---|---|---|---|---|
| Nonsplicing driver | −174.41 | −359.34 | 10.21 | 0.95 |
| Splicing driver | 298.38 | 169.57 | 429.15 | 0.95 |
| Age (years) | −15.94 | −28.04 | −3.37 | 0.95 |

Estimates of the effect size (in base pairs) of each covariate on NGS-estimated telomere length of single-HSPC-derived colonies. Parameter estimates reflect the coefficients in our linear mixed effects model (n=248 colonies from three people; Supplementary Note 1). For categorical covariates (nonsplicing and splicing driver), estimates are with respect to 'driverless' colonies.

shift towards lower LTL for *DNMT3A*-CH, *TET2*-CH and *PPM1D*-CH, but a higher LTL for *SRSF2*-CH. We also examined the interaction between LTL-PRS and VAF (LTL ~ LTL-PRS + VAF + LTL-PRS × VAF) directly and found that the interaction terms did not reach statistical significance in any of our gene-specific models (Supplementary Table 10). Moreover, inclusion of an interaction term did not noticeably improve model performance. This suggests that LTL-PRS and VAF largely affect measured LTL independently.

### Splicing-factor-mutant cells have longer telomeres

Our above findings reveal that, whereas clonal expansion of most common CH subtypes is associated with LTL shortening, expansion of *SRSF2*-CH clones may not be, and may even be associated with telomere elongation. This raises the hypothesis that *SRSF2* mutations, and potentially mutations in other splicing factor genes (higher *SF3B1* and *U2AF1* VAFs were not associated with shorter LTL, unlike other common CH drivers), enable cells to maintain or lengthen their telomeres. This would provide mutant HSCs with a relative clonal advantage only when progressive telomere shortening begins to restrain the replicative potential of normal HSCs, hence the predilection for SF-CH to develop in older people with low LTL-PRS.

To examine this hypothesis directly, we constructed the hematopoietic phylogenies of two older people with SF-CH from whole-genome sequencing (WGS) of 186 single hematopoietic stem and progenitor cell (HSPC)-derived colonies using somatic mutations as barcodes[7] (Supplementary Note 1 and Supplementary Table 11). In parallel, we used the same WGS data to estimate telomere length for each colony and overlaid these estimates over the phylogenetic trees. This revealed that, in both of these individuals, *SF3B1*-mutant colonies had significantly longer telomeres than *SF3B1*-wild type and *U2AF1*-wild type colonies (mean telomere lengths: 3,671 bp versus 3,129 bp and 3,844 bp versus 3,335 bp; $P = 6.82 \times 10^{-5}$ and $P = 4.69 \times 10^{-4}$ for PD34493 and PD41082, respectively; Fig. 3a–d). Furthermore, analysis of hematopoietic phylogenies revealed a nonsignificant trend towards longer telomeres in the sole *U2AF1*-Q157R mutant clade in our cohort ($P = 0.078$; Fig. 3b), with the difference in telomere lengths closely approaching statistical significance ($P = 0.053$) when this clade was compared to ancestrally related colonies, despite rapid recent clonal expansion after *U2AF1*-Q157R acquisition (Supplementary Note 1 and Extended Data Fig. 5). To confirm these findings using a different method, we adapted a protocol for telomere length estimation by qPCR[28] (Supplementary Fig. 2). For this, we generated new colonies from the same individual (PD34493; Methods), genotyped these for *SF3B1*-K666N and *U2AF1*-Q157R, and quantified their telomere length. This confirmed the findings from WGS-derived telomere lengths, with both *SF3B1*-mutant and *U2AF1*-mutant colonies displaying significantly longer telomeres than wild type colonies (Extended Data Fig. 6).

To understand what happens to the telomere length of splicing-factor-mutant HSPCs over time, we derived hematopoietic phylogenies from a third person (PD48499) using WGS of 62 HSPC-derived colonies generated across two timepoints 3.6 years apart. This did not reveal a difference in telomere lengths between the expanded *SF3B1*-mutant clade and polyclonal wild-type colonies, but did reveal a trend toward an increase in telomere length in the *SF3B1*-mutant clade ($P = 0.27$) but not in *SF3B1*-wild type cells, during the 3.6-year interval between samples (Fig. 3e,f), over which time the clinical phenotype progressed from clonal cytopenia of undetermined significance (CCUS) to MDS. Notably, this person had very short telomere lengths for their age (<1st centile) with respect to LTL reference ranges derived using flow-fluorescence in situ hybridization (FISH) (flow-FISH) or Southern blot, and were similar in length to the two older people in our study (PD34493 and PD41082)[29–31].

Pairwise telomere length comparisons of splicing-factor-mutant colonies with other colonies from the same person are confounded by the fact that many colonies are derived from HSPCs with shared ancestry and are therefore not independent. To account for this, we fitted a linear mixed effects model to test the hypothesis that genotype (splicing driver versus nonsplicing driver versus no driver) is associated with telomere length in hematopoietic colonies, adjusting for confounders of next-generation sequencing (NGS)-estimated telomere length (Supplementary Fig. 3 and Supplementary Note 1; Methods). Taking all HSPC colonies into account (n = 248 colonies from three individuals), we found a significant association between colony genotype and telomere length ($P = 9.21 \times 10^{-9}$), with splicing factor mutations showing an increased telomere length (+298 base pairs (bp)), whereas nonsplicing drivers were associated with decreased telomere length (−174 bp) compared to 'no driver' colonies (Table 1), mirroring our UKB findings (Fig. 2).

### Late-acquired splicing factor mutations and telomere length

A potential limitation of colony-derived telomere length measurements is the requirement for HSPCs to both divide and differentiate to give rise to colonies. The impact of these in vitro processes on telomere length may be altered in the presence of splicing factor mutations, leading to potentially erroneous conclusions about their roles in vivo. To circumvent this, we next adapted and validated a flow-FISH protocol[32,33] to estimate the telomere length of individual leukocytes in patients with splicing factor mutations without the need to derive colonies (Extended Data Fig. 7). Since the telomere length of the original HSC acquiring a somatic driver mutation is a principal determinant of the telomere length of its clonal progeny, we focused on samples with a subclonal splicing factor mutation acquired by an ancestral clone driven by other driver mutations, to capture the impact of such late-acquired splicing factor mutations on telomere length.

As there is no current methodology to simultaneously genotype and measure telomere length of individual cells, we instead used flow-FISH to sort cells by telomere length (for example, high versus low) and quantified the proportion of mutant cells within each fraction (Fig. 4a). We first studied a patient with chronic lymphocytic leukemia (CLL) with a late-acquired *SF3B1*-K700E mutation (Supplementary Table 12). After isolating CLL B cells (CD5$^+$CD19$^+$CD3$^−$), we sorted telomere$^{low}$ and telomere$^{high}$ fractions (Extended Data Fig. 8) by percentile telomere length and confirmed telomere length differences by qPCR (Supplementary Fig. 4). Both Sanger sequencing (Fig. 4b) and targeted amplicon sequencing (Fig. 4c) showed that *SF3B1*-K700E cells were enriched within the telomere$^{high}$ fraction compared to the telomere$^{low}$ fraction.

We then analyzed patients with chronic myelomonocytic leukemia (CMML) or acute myeloid leukemia (AML) harboring *TET2* and *SRSF2* comutations. Patients were chosen such that *TET2* mutations were clonal (that is, acquired first) and *SRSF2* mutations subclonal (that is, acquired second). Unlike CLL, we could not isolate CMML cells by cell surface markers so instead sorted bulk mononuclear cells (MNCs) by telomere length and quantified enrichment for the *TET2* versus the *SRSF2* mutations in each fraction. We found that, relative to *TET2*-mutant, *SRSF2*-mutant VAFs were lower in the telomere$^{low}$ and enriched in the telomere$^{high}$ fractions by both Sanger sequencing

(Fig. 4d–g) and NGS (Supplementary Fig. 5). Interestingly, this was also true for a CMML harboring the rare pathogenic mutation *SRSF2*-V18L[34,35] (Fig. 4f). A similar pattern was seen in a single participant with MDS containing *TET2* and *U2AF1* mutations (Fig. 4h). These findings provide further support to the premise that, in contrast to other CH-associated mutations, splicing factor mutations facilitate clonal expansion while maintaining telomere length. Interestingly, we did not see this difference in a single individual whose CMML had been treated with 31 cycles of Azacitidine treatment by the time of sampling (Extended Data Fig. 9), potentially because of selective elimination of faster growing double-mutant cells.

## Somatic TERT promoter mutations drive late-onset CH in UKB

In light of the fact that *PPM1D*-CH and splicing factor-CH are prevalent amongst both TBD patients and older people with lower LTL-PRS, we hypothesized that somatic mutations in the *TERT* gene promoter (*TERT*p), which drive the most common form of CH in TBD patients[23], may also be enriched in people with polygenic short telomeres. To identify putative *TERT*p-CH in the UKB at each of the three known *TERT*p mutational hotspots[23], we performed pileup analysis of WGS data across the *TERT* promoter and implemented several filters to reduce erroneous calls (Supplementary Note 2). This identified 148 people with *TERT*p-CH, as well as three people with both *TERT*p mutations and a prevalent diagnosis of hematological malignancy who were excluded from downstream analyses. Notably, *TERT*p-CH exhibited an age-and sex-related prevalence resembling that of SF-CH (median age, 65 years; 76.4% male), as well as a similar overall prevalence, when SF-CH was identified from WGS using the same pileup analysis pipeline (Supplementary Note 2 and Extended Data Fig. 10). Notably, people with *TERT*p-CH had significantly lower LTL-PRS values relative to UKB participants without *TERT*p-CH (Supplementary Note 2; $P = 0.001$), mirroring what we observed with *PPM1D*-CH and SF-CH. These observations further endorse the main premise of our study, that telomere attrition becomes an instrument for clonal selection in aging hematopoiesis/leukemogenesis.

## Discussion

The dependence of hematopoiesis on normal telomere function is strikingly evident in TBDs, Mendelian disorders involving telomere maintenance genes, in which abnormal telomere shortening is commonly associated with hematopoietic failure/aplastic anemia[36,37]. At the other end of the spectrum lies the increased risk of CH in people with polygenic[5,12] or monogenic[13] inheritance of longer telomeres. Here we uncover another facet of the close interaction between telomeres and hematopoiesis, by discovering that age-related telomere attrition becomes an instrument for clonal selection, particularly amongst people with lower genetically predicted telomere length. Specifically, we find that mutations in splicing factor genes, *PPM1D* and *TERT*p seem

to rescue HSCs from critical telomere shortening or its consequences, conferring a fitness advantage over their unmutated peers in a manner reminiscent of what has been observed in much younger people with TBD[20–23] (Fig. 5a–e).

We reported previously that mutations in splicing factor genes are rare until the seventh decade of life and then rise rapidly in prevalence[4]. This mirrors the age-related prevalence of MDS—a disease group where splicing factor mutations are very common[38,39]. Progress in understanding the molecular consequences of these mutations has identified specific target mRNAs responsible for known MDS-related cellular phenotypes such as ring sideroblasts[40], anemia[41] or aberrant hematopoietic differentiation[42]. However, the mechanisms through which *SF3B1*, *SRSF2* and *U2AF1* mutations drive clonal expansion remain unknown[43]. The fact that these mutations are mutually exclusive[38,39,44,45] suggests that they drive clonal selection through shared mechanisms. However, although mutations in any of these genes lead to mRNA mis-splicing, there is little overlap between mis-spliced[44,46] or differentially expressed[45] mRNAs associated with each. Our findings propose that prevention of telomere attrition is such a shared mechanism, a premise that could explain: (1) the rapid rise in the prevalence of splicing-mutant CH and MDS in old age and as telomeres become critically short in at-risk people (that is, those with the shortest telomeres) and (2) the higher prevalence of splicing factor gene mutations in HSC-derived myeloid malignancies compared to any other cancer, with blood being the tissue that incurs the most marked age-related telomere shortening[47].

*PPM1D* mutations are known to drive context-dependent CH in patients receiving genotoxic therapies, imparting a relative fitness advantage on mutant HSCs by attenuating DDR[17]. However, most cases of *PPM1D*-CH arise in people without such history, and this was also the case in the UKB. In particular, of 647 cases of *PPM1D*-CH, only 32 had a documented history of previous chemotherapy, five had radiotherapy and only 98 had a documented history of cancer (excluding nonmelanoma skin cancer). The total number of cases after amalgamating all three categories (chemotherapy + radiotherapy + malignant neoplasm) was 100. This finding indicates that a substantial proportion of cases of *PPM1D*-CH are not related to genotoxic therapy, but instead arise in the context of replicative senescence linked to telomere attrition, as happens with TBDs[20,21]. It is also noteworthy that cisplatin, the chemotherapeutic agent linked most strongly to *PPM1D*-CH, can directly bind and shorten telomeres[48,49]. Similarly, it is notable that we did not see an increase in *TP53*-CH amongst people with lower LTL-PRS, proposing a specific link between *PPM1D* and replicative senescence that is endorsed by findings of direct physical and functional interactions between *PPM1D* and the shelterin complex[50].

Collectively, the above findings reveal that telomere attrition constrains normal hematopoiesis in old age and becomes an instrument for clonal selection by mutations that either prevent telomere attrition

---

**Fig. 4 | Impact of splicing factor mutations on telomere length in chronic leukemias. a**, Peripheral blood samples were collected from patients with clonal blood disorders and sorted by percentile telomere length using flow-FISH. Sorted populations were prepared for sequencing to quantify the enrichment of mutant cells within each fraction. **b,c**, Sanger sequencing (**b**) and targeted amplicon sequencing (**c**) of unsorted (left), telomere^low (middle, ≤33rd percentile telomere length) and telomere^high (right, ≥66th percentile telomere length) populations of MNCs from a patient with CLL with a subclonal *SF3B1*-K700E mutation. Black arrows indicate mutated base. **d**, Ratio of *SRSF2*-mutant and *TET2*-mutant cells in MNCs sorted by percentile telomere length from an individual with AML, with VAF quantified by Sanger sequencing. **e**, Ratio of *SRSF2*-mutant and *TET2*-mutant cells in MNCs sorted by percentile telomere length from an individual with CMML, with VAF quantified by Sanger sequencing. **f**, Ratio of *SRSF2*-mutant and *TET2*-mutant cells in MNCs sorted by percentile telomere length from an individual with CMML, with VAF quantified by Sanger sequencing. For panels **d**–**f**, 'unsorted' populations were derived from the same population

as sorted fractions but were gated on live single cells only (that is, not sorted by telomere length). Diagnosis and clinically reported VAF (assessed by clinical NGS performed on bone marrow DNA) is shown above each plot and may differ from the VAF of 'Unsorted' cells (MNCs) due to differences in the cellular composition of these fractions. Absence of bars in 'Telomere^<10%' (**d**–**f**) and 'Telomere^10–33%' (**f**) indicates that no *SRSF2*-mutant cells were detected within those fractions. **g**, Summary of data from the three participants shown in **d**–**f**. Points represent the ratio of *SRSF2*-mutant to *TET2*-mutant cells in a single person, bar heights represent the mean *SRSF2*/*TET2* ratio within each telomere length fraction and error bars represent the mean values ± s.e.m. *P* values were derived by one-way ANOVA followed by Tukey's multiple comparison test with individual participants treated as biological replicates. Only adjusted *P* values below the significance threshold ($P < 0.05$) are shown. **h**, Ratio of *U2AF1*-mutant and *TET2*-mutant cells in MNCs sorted by percentile telomere length from a patient with MDS-single lineage dysplasia (SLD) with VAF quantified by Sanger sequencing. Panel **a** created using BioRender.com.

(splicing factor genes) or attenuate replicative senescence-related DDR signaling (*PPM1D*). These observations mirror what is seen in TBD[20–23], but with significant differences. For example, *U2AF1* is the splicing factor gene most commonly mutated in TBD[20–23], whereas *SF3B1* and *SRSF2*

mutations are more common in aging-related telomere attrition[2,3,5]. The basis of these differences is unknown, but may relate to the fact that single gene defects underlie the abnormal telomere attrition in TBDs, whilst age-related attrition is multifactorial and polygenic in origin.

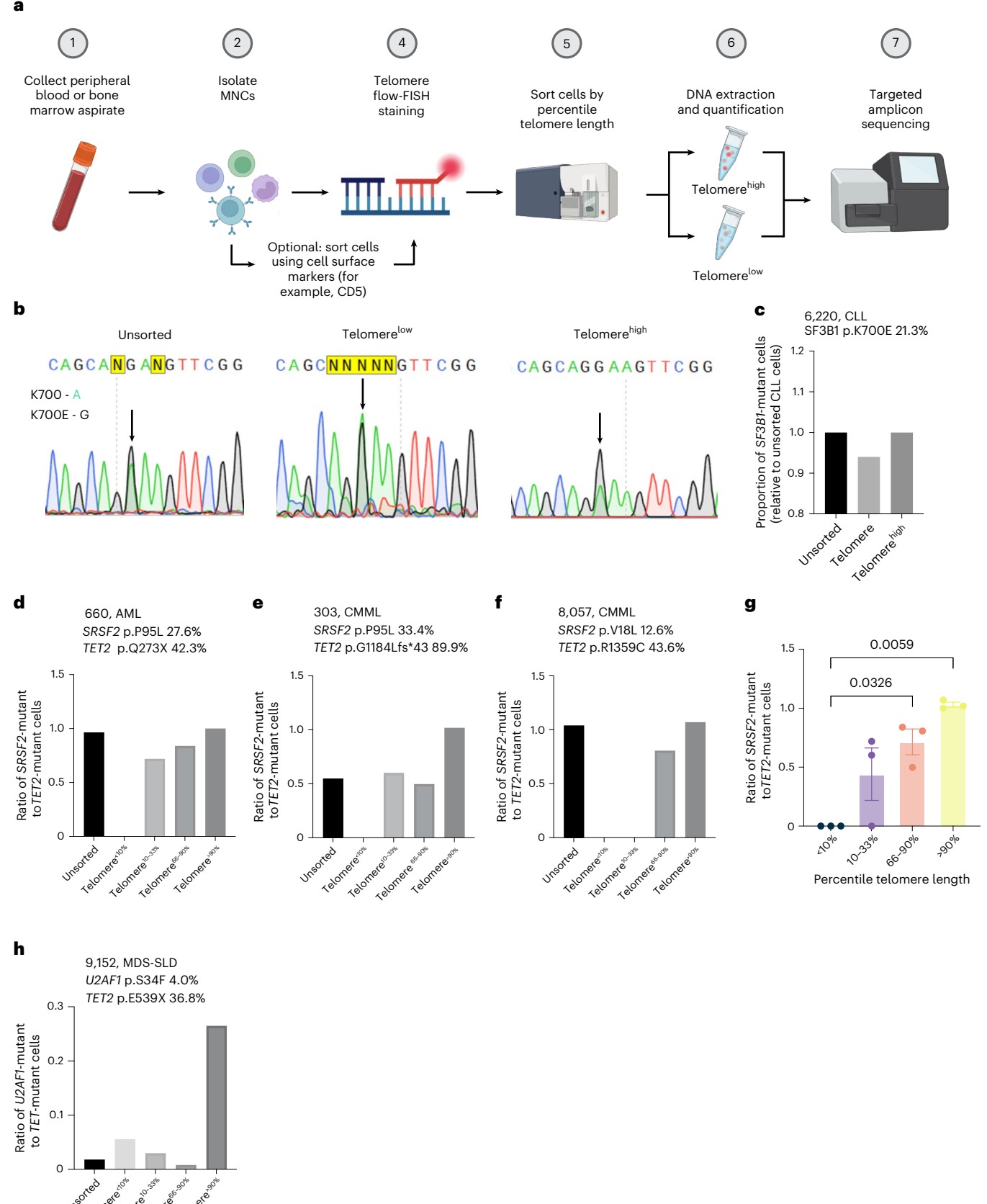

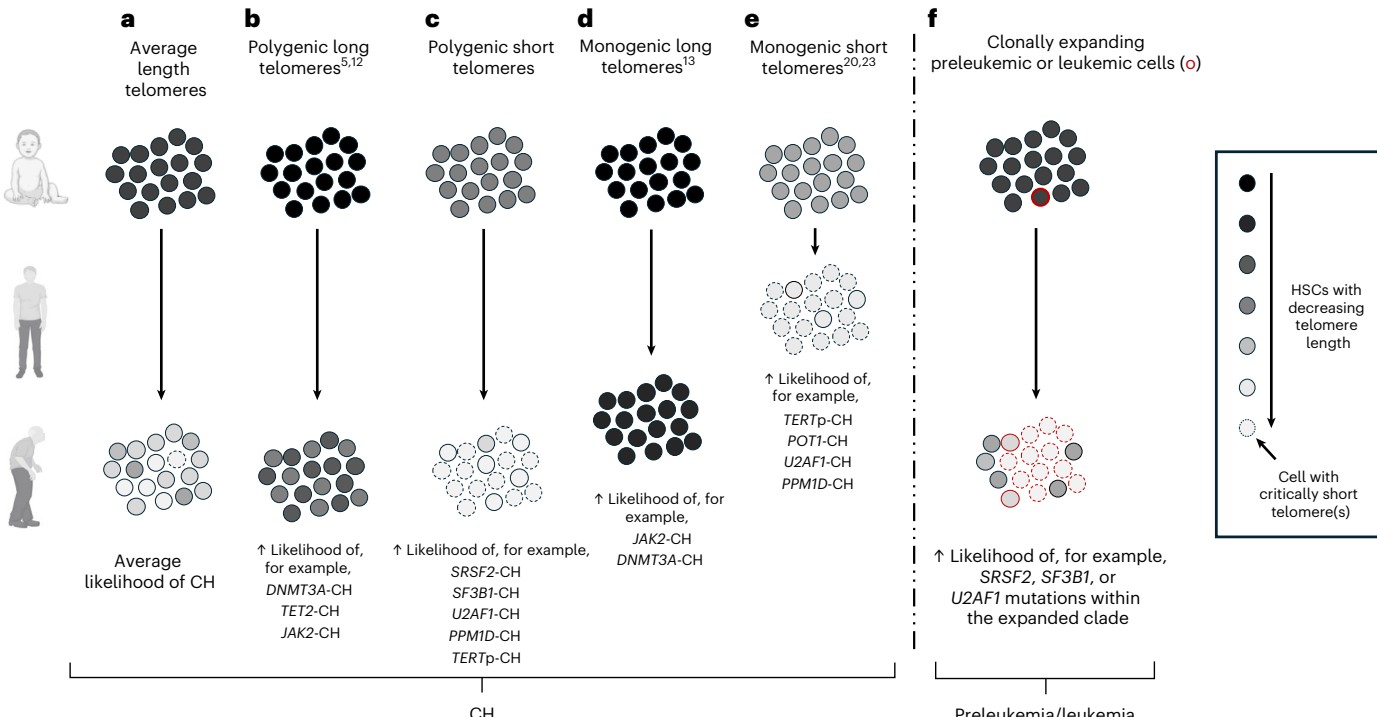

**Fig. 5 | Proposed model for the impact of telomere length on the development of different types of CH.** Hypothesis explaining how telomere length shapes the global landscape of clonal hematopoiesis throughout life. **a**, HSCs of people inheriting average/near-average length telomeres (through polygenic inheritance of variants associated with long telomeres) that shorten with age. These people have an average risk of developing CH. **b**, People inheriting longer telomeres than average have a greater propensity to develop CH driven by mutations in genes like *DNMT3A*, *TET2* and *JAK2*, which are associated with significant telomere shortening during clonal expansion. Initiation in an HSC with longer telomeres allows clones to expand for longer before their telomeres become critically short[5,12]. **c**, People inheriting shorter telomeres than average are more likely to develop critical telomere shortening, triggering DDR signaling and replicative senescence in a large proportion of their HSCs. In this context, there is selection for mutations that enable HSCs to avoid this fate, either by promoting telomere maintenance (for example, splicing factor mutations) or by attenuating DDR signaling (*PPM1D*). **d**, People with monogenic inheritance of variants in

genes such as *POT1*, which are associated with sustained long HSC telomeres with advancing age, are at highly increased risk of early-onset CH driven by mutations in genes such as *JAK2* that normally drive marked telomere shortening (for example, Fig. 2b)[13,14]. **e**, People with TBD develop marked telomere shortening in their HSCs (and other tissue stem cells) at a young age. In this setting, the global shortening of telomeres presents a strong selection pressure that favors CH driven by adaptive mutations that 'reverse' the monogenic defect (*TERTp*, *POT1*) or maladaptive mutations mirroring those seen in people with polygenic short telomeres (*U2AF1*, *PPM1D*)[20–23]. **f**, Mirroring what is observed in the context of inherited short telomeres leading to a 'field' of HSCs with critical telomere shortening (**c** and **e**), clonal expansion-mediated telomere attrition can also become an instrument for clonal selection within previously expanded HSC clones (red border). In this context, mutations that prevent telomere attrition (*SRSF2 or SF3B1*) can restore clonal fitness and facilitate further expansion or leukemic progression of clones previously expanded by driver mutations associated with telomere shortening. Figure created using BioRender.com.

Finally, we examined the role of splicing gene mutations as later (noninitiating) events in blood cancer development. Our findings show that in this context too, splicing gene mutations seem to prevent telomere attrition, ostensibly restoring the fitness of a clone whose previous expansion led to substantial telomere shortening (Fig. 5f). It is also plausible that splicing gene mutations may act in this same way to restore fitness of apparently 'driverless' clones (for example *U2AF1*-mutant clone in Fig. 3a), which become very common after the seventh decade of life[51]. In line with this is the observation that splicing factor gene mutations (predominantly *SF3B1*) are late events in CLL[52] and solid cancers[53], which may reflect the fact that telomere attrition is slower in the cells of origin of these malignancies compared to HSCs[47], such that telomere shortening only becomes critical after substantial clonal expansion. Collectively, our findings propose that urgent studies are required to decipher how splicing gene mutations act to prevent telomere attrition as this may lead to new therapeutic approaches targeting splicing factor-mutant myeloid and other cancers.

## Online content

Any methods, additional references, Nature Portfolio reporting summaries, source data, extended data, supplementary information,

acknowledgements, peer review information; details of author contributions and competing interests; and statements of data and code availability are available at https://doi.org/10.1038/s41588-025-02296-x.

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

Matthew A. McLoughlin [1,2,3,22], Sruthi Cheloor Kovilakam[1,2], William G. Dunn [1,2,3,22], Muxin Gu[1,2,3], Jake Tobin [1,2,3], Yash Pershad [4], Nicholas Williams [5], Daniel Leongamornlert [5], Kevin Dawson[5], Laura Bond[1,2], Ludovica Marando[1,2,6], Sean Wen[2,7], Rachael Wilson [1,2], Giampiero Valenzano[1,2], Vasiliki Symeonidou[1,2], Justyna Rak[1,2], Aristi Damaskou[1,2], Malgorzata Gozdecka [1,2], Xiaoxuan Liu[1,2], Clea Barcena[8], Josep Nomdedeu [9], Paul Costeas[10], Ioannis D. Dimitriou[11], Edoardo Fiorillo [12], Valeria Orrù[12], Jose Guilherme de Almeida[13], Thomas McKerrell[14,15], Matthew Cullen[2], Irina Mohorianu [1], Theodora Foukaneli[2,3], Alan J. Warren [1,2,3,16], Chi Wong[2,3], George Follows[3], Anna L. Godfrey[3], Emma Gudgin[3], Francesco Cucca [12,17], Eoin McKinney [18], E. Joanna Baxter [1,2], Moritz Gerstung [19], Jonathan Mitchell [7], Daniel Wiseman [20], Alexander G. Bick [4], Margarete Fabre [2,3,7], Pedro M. Quiros [8], Jyoti Nangalia [1,2,3,5], Siddhartha Kar[21] & George S. Vassiliou [1,2,3,10,11] ✉

[1]Cambridge Stem Cell Institute, University of Cambridge, Cambridge, UK. [2]Department of Haematology, University of Cambridge, Cambridge, UK. [3]Department of Haematology, Cambridge University Hospitals NHS Trust, Cambridge, UK. [4]Department of Medicine, Division of Genetic Medicine, Vanderbilt University Medical Center, Nashville, TN, USA. [5]Wellcome Sanger Institute, Hinxton, Cambridge, UK. [6]Department of Hematology, Mayo Clinic, Rochester, MI, USA. [7]Centre for Genomics Research, Discovery Sciences, BioPharmaceuticals R&D, AstraZeneca, Cambridge, UK. [8]Department of Biochemistry and Molecular Biology, Instituto Universitario de Oncología (IUOPA),Universidad de Oviedo, Oviedo, Spain. [9]Hospital Sant Pau, Barcelona, Spain. [10]Centre for the Study of Haematological Malignancies, Nicosia, Cyprus. [11]Cyprus Cancer Research Institute, Nicosia, Cyprus. [12]Institute for Genetic and Biomedical Research, National Research Council, Lanusei, Italy. [13]Computational Clinical Imaging Group, Champalimaud Foundation, Lisbon, Portugal. [14]Department of Haematology, University Hospital Geelong, Geelong, Victoria, Australia. [15]School of Medicine, Deakin University Geelong, Geelong, Victoria, Australia. [16]Cambridge Institute for Medical Research, University of Cambridge, Cambridge, UK. [17]Department of Biomedical Science, University of Sassari, Sassari, Italy. [18]Cambridge Institute of Therapeutic Immunology and Infectious Disease, University of Cambridge, Cambridge, UK. [19]Division of AI in Oncology, German Cancer Research Centre DKFZ, Heidelberg, Germany. [20]Cancer Research UK, Manchester Institute, Manchester, UK. [21]Department of Oncology, Early Cancer Institute, University of Cambridge, Cambridge, UK. [22]These authors contributed equally: Matthew A. McLoughlin, Sruthi Cheloor Kovilakam, William G. Dunn. ✉e-mail: gsv20@cam.ac.uk

## Methods

### Ethical regulations

This study was conducted under approved UKB application no. 56844. Clinical samples were obtained with informed written consent from the Cambridge Blood and Stem Cell Biobank with approval by the Cambridge East Research Ethics Committee (REC) (REC 18/EE/0199 and 24/EE/0116), from the SardiNIA longitudinal study of immune senescence (REC 15/EE/0327) with approval by the East of England (Essex) REC, or from the Manchester Cancer Research Centre Biobank with approval by the South Manchester REC (REC 07/H1003/161+5; HTA license 30004).

### Statistics and reproducibility

In this project, we included 454,340 UKB participants with somatic variant call data from our previous study[19]. From this group, participants who had withdrawn consent, had a mismatch between genetic and self-reported sex or had differences in the dates of attending the assessment center and the blood sample collection, were excluded from the study resulting in n = 454,098 participants. Power calculations were conducted to determine the minimum number of cases for inclusion ('Mutation Calling'). These analyses were not randomized, and the investigators were not blinded to allocation during experiments and outcome assessment.

### Mutation calling

Mutations in 41 CH driver genes (Supplementary Table 13) were called using Mutect2 GATK v.4.1.3.0 from whole-exome sequencing (WES) data of peripheral blood DNA from 454,340 UKB participants and filtered as described previously[19] (Methods). A specific VAF cutoff was not used to define participants with CH. Mutations in *DNMT3A* at the hotspot R882 were grouped as 'DNMT3A_R882' and the rest as 'DNMT3A_other.' *U2AF1* mutations were identified using Samtools mpileup (v.1.15.1) and the variants with at least three alternate allele reads and a VAF ≥ 0.05 were included in the analysis. Participants who were diagnosed with hematological malignancy before recruitment were removed from all analyses involving LTL. Participants harboring mutations in several genes, or mutations in less frequently mutated genes (<100 cases), were excluded from LTL and LTL-PRS analyses. The threshold of 100 cases was chosen following power calculations performed using the 'samplesizelogisticcasecontrol' package (v.2.0.2) in R (Methods). An exception to this threshold was made for mutations in the splicing factor gene *U2AF1* (n = 82) in light of its recently reported association with CH in TBD[23]. We also excluded *ATM*, *BRCC3* and *STAT3* from downstream analysis as these are not widely recognized as drivers of myeloid CH[54]. Somatic mutations in the 'All of Us' cohort[55] were identified as described previously[56].

### Mosaic chromosomal alterations

mCA calls were obtained from Loh et al.[57]. Before analyses, participants carrying several mCAs or any CH driver gene mutations or mCAs of unknown copy number change/cell fraction were filtered out. Based on the chromosome and the type of copy number change, mCAs were grouped into autosomal mCAs (any type of copy number change), LOX and LOY.

### PRS calculation

We used PRSice-2 (v.2.3.5) to compute PRS associated with telomere length based on the 131 SNPs identified in the GWAS by Codd et al.[24] with beta coefficients from the same study serving as weights in the PRS computation. Imputed genotypes available in the UKB were used for this analysis. Calculated PRS were *Z*-normalized. Participants with a prevalent hematological diagnosis were not excluded for LTL-PRS analyses, as those people would have developed CH at some stage before development of malignancy.

### Myeloid malignancy phenotypes

UKB participants with a prevalent diagnosis of hematological malignancy were defined using ICD codes (Supplementary Table 14). If a participant had several myeloid neoplasms, only the first diagnosed disease was considered for analysis. People who had chemotherapy before diagnosing myeloid malignancies were excluded from the association analysis with LTL and LTL-PRS.

### Regression analyses

All linear and logistic regression analyses were performed using the Python (v.3.9.7) module statsmodels (v.0.12.2). First, the association between the presence of a CH mutation and LTL was investigated using a linear regression model on LTL with binary predictor variables representing presence/absence (1/0) of mutations in each of the CH driver genes and covariates. For quantifying the variation in telomere length with respect to VAF, a linear regression model for predicting telomere length was built with the variables $(Gene + Gene\ VAF)_{for\ all\ genes}$ where $(Gene + Gene\ VAF)_{for\ all\ genes}$ = DNMT3A + DNMT3A VAF + ASXL1 + ASXL1 VAF + TET2 + TET2 VAF and so on for all genes and covariates. DNMT3A, ASXL1 and so on are variables that represent whether a mutation is present (1) or not (0) in the specific gene. The covariates used were sex, age, smoking status, genetic principal components from one to ten, white blood cell counts and percentages of types of white blood cell. Blood-count-related parameters were winsorized to 99% before regression. Similar analysis was performed for mCAs using cell fraction instead of VAF. Correction for multiple testing was performed using the Benjamini–Hochberg procedure and applying a threshold of FDR < 0.05.

Logistic regression analyses were performed to quantify the association between polygenic risk scores and CH/mCA. Age, sex, smoking status and first ten genetic principal components were used as the covariates in the regression. Correction for multiple testing was performed using the Benjamini–Hochberg procedure and applying a threshold of FDR < 0.05.

### Mendelian randomization

The same set of variants as used in PRS calculation were employed as genetic instruments in the MR analyses to identify causal associations between telomere length and various types of CH. Coefficients quantifying the association between each of the genetic instruments and each of CH types were obtained by Firth's logistic regression analysis performed using the logistf function in R (v.4.2.1). MR analyses were performed using the TwoSampleMR package (v.0.5.7) in R (v.4.3.0) using these coefficients along with the coefficient estimates for association between genetic instruments and telomere length from Codd et al.[24] and the results were reported for the inverse-variance-weighted method. Correction for multiple testing was performed using the Benjamini–Hochberg procedure and applying a threshold of FDR < 0.05.

### Analysis of *TERT*p mutations in the UKB

*TERT* promoter mutations we identified from WGS of blood DNA from 488,364 UKB participants as this region is not captured adequately by the UKB WES panel. We used samtools mpileup (v.1.15.1) to identify single nucleotide variants (SNVs) across the entire *TERT* promoter (chr5:129489–1295157) with high sensitivity and then applied several manual filters (depth ≥ 15 bp, at least three supporting reads, VAF ≥ 30%). This approach was used in place of somatic variant calling pipelines due to the low depth of WGS across the promoter (median 34×). We then focused our subsequent analysis on three mutational hotspots identified previously as somatic rescue mutations in TBD (chr5:1295046:T:G, chr5:1295113:G:A and chr5:1295135:G:A)[22,23]. To benchmark our approach for calling *TERT*p hotspot mutations from WGS data, we used the same approach to call hotspot mutations in *SF3B1* (R625, K666 and K700) and *SRSF2* (P95) from WGS, filtered them as described above, and compared their age-related prevalence to

*TERT*p-CH, as well as *SF3B1*/*SRSF2*-CH identified from WES. A detailed outline of the approach used to call *TERT*p mutations and subsequent benchmarking is contained in Supplementary Note 2.

## Construction of phylogenetic trees from WGS of hematopoietic cell colonies

We analyzed data from a man aged 83.8 years with SF-CH detected in blood DNA (PD34493: *U2AF1*-Q157R 10.3%, *SF3B1*-K666N 8.7%, *NOTCH1*-L441L 0.3%), studied previously by phylogenetic analysis using WGS of single-HSPC-derived colonies[7]. Specifically, for this study, we also performed colony WGS and phylogenetic analyses on samples from a woman aged 73.9 years with SF-CH (PD41082: *TET2*-Q1825X 33.8%, *SF3B1*-K666N 7.1%, *TET2*-S315fs 3.2%, *GNB1*-K57E 1.5%, *TET2*-L1322Q 1.3%, *TET2*-H435fs 1.2%, *TET2*-Q1274R 1.1%, *TET2*-Q1542X 0.8%). Both were participants in the SardiNIA study[58] and were studied because they harbored SF-CH with sizeable clones[7]. Ninety-six colonies per person were picked from methylcellulose-based medium previously plated with peripheral blood mononuclear cells (PBMCs) and used for WGS as described previously[7,51]. To investigate trends in clonal expansion and telomere length over time, heterochronous peripheral blood samples were taken from a man with *SF3B1*-CCUS (PD48499) aged 50.2 years (*n* = 24 colonies, *SF3B1*-K700E 42.4% on clinical NGS of bone marrow DNA) and *SF3B1*-MDS at age of 53.8 years (*n* = 72 colonies, *SF3B1*-K700E 42.8% on clinical NGS of bone marrow DNA). This man was selected because of the presence of SF-CH and availability of longitudinal blood samples.

Phylogenetic relationships were derived from colony WGS data as described previously[7,59,60]. Briefly, reads were aligned to the human reference genome (GRCh38) using BWA-MEM (https://github.com/lh3/bwa). Variant calling was performed using CaVEMAN[61] (SNV) and Pindel[62] (indels) against an in silico generated unmatched normal. Colonies with low sequencing depth (<6×) or low clonality (median VAF < 0.4) were removed from downstream analyses. Filtering was performed to remove germline variants and artefacts arising from low DNA input, using pooled information across per-person colonies as outlined previously[59,60]. For all mutations passing quality filters in at least one colony, matrices were generated of mutant and normal reads at each site for every colony from the same person, using vafCorrect (https://github.com/cancerit/vafCorrect) to correct for reference bias arising during alignment of reads containing indels. Genotype matrices of SNVs were used as input to MPBoot[63] to infer the phylogenetic relationships between colonies using a maximum parsimony approach with bootstrap approximation. The treeMut package (https://github.com/nangalialab/treemut) was then used to assign mutations (SNVs and indels) to branches and estimate branch lengths. To convert the *x* axis of each phylogenetic tree from number of mutations to chronological age, where the tips of the tree are the age of the person at sampling, we used the package Rtreefit[59] (https://github.com/nangalialab/rtreefit) to scale branch lengths, accounting for differences in mutation rate across the human lifespan and intersample variation in the sensitivity of detecting somatic variants.

## Telomere length estimation from WGS data

Telomere length estimates were estimated from the NovaSeq-sequenced colony WGS data described above using Telomerecat[64]. Novaseq's two-dye technology interprets the absence of signal from a failed cluster as a run of 'G' base calls that can confound Telomerecat due to its resemblance to the telomere sequence (TTAGGG). The likelihood of cluster failure increases with read length; hence, we ran Telomerecat with the '-trim 75' argument to estimate telomere lengths from the first 75 bp of each read and avoid the higher error regions towards the end of the read. Phylogenetic trees were then annotated with telomere length estimates using the ggtree (v.3.8.2) package in R[65].

Pairwise comparison of telomere length estimates were performed using the Wilcoxon rank sum test. Alongside this, we also fitted a linear

mixed effects model using the lme4 package (v.1.1) in R[66] to model colony telomere length with sequencing batch as a random effect and genotype and age as fixed effects (Supplementary Note 1):

$$\text{Colony } telomere\ length \sim \text{Age} + \text{Genotype} + (1|\text{Batch})$$

This model was fitted on all colonies passing filters and included in the final phylogenetic trees (*n* = 248). To test the hypothesis that genotype (splicing factor driver mutation/other driver mutation/driverless) is associated with colony telomere length at a cohort level, we compared linear mixed effects models with and without genotype as a fixed effect and compared both models using one-way analysis of variance (ANOVA). Confidence intervals (CIs) for fixed effect coefficients were estimated using bootstrap resampling with 10,000 resamples and calculating the 95% CI for each coefficient based on the first 5,000 converged models.

## Cell-line culture

K562 were cultured in IMDM (Gibco, cat no. 12440053) supplemented with 10% FBS (Gibco, catalogue number SH30071.03), 2 mM L-glutamine and 1% penicillin/streptomycin. OCI-AML2 were cultured in α-MEM (Gibco, catalogue number 12571063) supplemented with 20% FBS, 2 mM L-glutamine and 1% penicillin/streptomycin. HEK293FT were cultured in DMEM (Gibco, catalogue number 11960085) supplemented with 10% FBS, 2 mM L-glutamine and 1% penicillin/streptomycin and passaged using trypsin. Cells were maintained at 37 °C and 5% $CO_2$ in a humidified incubator and passaged every 2–3 days. Cas9-expressing cell lines were generated using lentivirus generated from pKLV2-EF1aBsd2ACas9-W plasmid (Addgene, catalogue number 67978) as described below.

## Lentivirus generation and transduction

Tissue culture plates (15 cm$^2$) were coated in 0.1% gelatin for 37 °C for 30 min. Plates were washed with PBS (Sigma, catalogue number D8537-500) and seeded with $8 \times 10^6$ HEK293FT cells. Vector plasmid (7.5 µg) was mixed with 18.5 µg of pPAX2 (Addgene, catalogue number 12260), 4 µg of pMD2.G (Addgene, catalogue number 12259), 30 µl of PLUS reagent and 7.5 ml of Opti-MEM (Gibco, catalogue number 51985026) and incubated at room temperature for 5 min. Lipofectamine LTX (180 µl; Invitrogen, cat no. 15338030) was added, and the mixture was incubated at room temperature for an additional 30 min. After this, the transfection mixture was added dropwise to cells followed by 20 ml of HEK293FT medium (prepared as above) and placed in a humidified incubator overnight. Medium was changed the following morning. On day 2, viral supernatant was filtered with 0.45 µM low-protein binding filter (Nalgene, catalogue number 190-2545), mixed with Lenti-X (Takara Bio, catalogue number 631232) and kept at 4 °C overnight. Viral supernatant was then spun at 1,500*g* for 45 mins at 4 °C and the pellet was resuspended in 300 µl of ice-cold PBS.

Concentrated virus (15 µl) was added to $1 \times 10^5$ cells in 1 ml of medium supplemented with 6.7 µg ml⁻¹ polybrene. Cells were centrifuged at 870*g* and 37 °C for 1 h and returned to the incubator. Following 2 days in culture, transduced cells were selected by supplementing medium with 10 µg ml⁻¹ blasticidin or 1 µg ml⁻¹ puromycin for 5 days.

## *TERT* knockout and validation

Two gRNAs targeting *TERT* exon 2 (Supplementary Table 15) were cloned into the pKLV2-U6gRNA5(BbsI)-PGKpuro2ABFP-W vector (Addgene, catalogue number 67974) and lentivirus was generated and transduced as described above. Transduced cells were selected using 1 µg ml⁻¹ puromycin and maintained in culture for a total of 14 days. Cells ($1 \times 10^6$) cells were transferred to a 1.5 ml tube and centrifuged at 300*g* for 5 min and supernatant was discarded. Genomic DNA was extracted from the pellet using the DNeasy Blood and Tissue Kit (Qiagen, catalogue number 69504). DNA was quantified and diluted in

UltraPure DNase/RNase-Free Distilled Water (Invitrogen, catalogue number 11538646). *TERT* gRNA activity was validated using PCR with primers spanning the region of interest (Supplementary Table 15) followed by Sanger sequencing.

PCR was performed on 1 ng of diluted gDNA using HiFi HotStart ReadyMix (Kapa, catalogue number 07958927001) and primers spanning the *TERT* region of interest using the following reaction conditions: 95 °C for 3 min, 35 cycles of (98 °C for 20 s, 60 °C for 15 s, 72 °C for 30 s) and 72 °C for 5 min. PCR product was purified using QIA quick PCR Purification Kit (Qiagen, catalogue number 28104) and submitted for Sanger sequencing with the forward primer using GeneWiz. Sequencing traces were analyzed in SnapGene to confirm *TERT* gRNA activity.

### Clinical samples

Peripheral blood was collected into lithium heparin tubes (Sarstedt, catalogue number 02.1065.001) and bone marrow aspirate was collected in RPMI (Gibco, catalogue number 21875034) supplemented with 1% penicillin/streptomycin and 10 IU ml$^{-1}$ sodium heparin (Merck, catalogue number H3149-10KU). Samples were processed using Ficoll (Merck, catalogue number GE17-1440-02) and/or PharmLyse (catalogue number BD 555899) to isolate MNCs, leukocytes or granulocytes. Cells were used immediately in experiments or cryopreserved in FBS supplemented with 50% human serum albumin and 10% dimethylsulfoxide and stored for future use.

### Colony-derived WGS

Samples were plated to form colonies and prepared for WGS as described previously[7,51]. Briefly, peripheral blood or bone marrow MNCs were plated at $3 \times 10^6$ cells ml$^{-1}$ in MethoCult H4034 (Stemcell Technologies, catalogue number 04034) and cultured in a humidified incubator at 37 °C and 5% CO$_2$ for 14 days. Colonies were picked and resuspended in RLT (Qiagen, catalogue number 79216). Libraries were prepared using a low-input pipeline and 150 bp paired-end sequencing was performed on a NovaSeq 6000 at 15× coverage.

### DNA extraction and quantification

For cell lines, genomic DNA was isolated using DNeasy Blood and Tissue Kit and quantified using the Qubit dsDNA HS Kit (Invitrogen, catalogue number Q32851). Telomere qPCR (Supplementary Methods) on colonies lysed in RLT was attempted but yielded poor and inconsistent results, particularly at higher RLT concentrations and low DNA input (Supplementary Fig. 2). Instead, cells were plated as described above and picked after 14 days into 17 µl of PicoPure (Applied Biosystems, catalogue number KIT0103) buffer supplemented with Proteinase K according to the manufacturer's instructions, lysing the cells. Lysate was placed in a thermocycler under the following conditions: 65 °C for 6 h, 75 °C for 30 min, 4 °C hold. Volume was made up to 50 µl with UltraPure H$_2$O and DNA was quantified using the Quant-iT PicoGreen dsDNA Assay Kit (Invitrogen, catalogue number P7589).

### Flow-FISH

Cryopreserved cells were thawed and washed twice in warmed RPMI supplemented with 10% FBS. Cells were centrifuged at 300*g* for 5 min and resuspended in FACS buffer (PBS supplemented with 0.1% BSA (Fisher, catalogue number BP9702-100)). Cells were counted and $1–3 \times 10^6$ cells were aliquoted into 1.5 ml tubes. Cells were centrifuged at 300*g* for 5 min and resuspended in 1 ml PBS containing 1:1,000 Fixable Viability Dye eFluor 780 (eBioscience, catalogue number 65-0865-14) and incubated at 4 °C in the dark for 20 min. Following this, cells were washed twice in FACS buffer. For the CLL sample only, cells were then centrifuged at 300*g* for 5 min and resuspended in FACS buffer supplemented with the following antibodies: 1:100 CD3-BUV395, 1:160 CD19-BV421, 1:160 CD11b-PE, 1:100 CD33-BV510, 1:50 CD5-FITC (Supplementary Table 16). Cells were incubated at 4 °C in the dark for 20 min and washed twice with FACS buffer and sorted as described below.

Following sorting (CLL sample) or viability staining (remaining samples), cells were centrifuged at 300*g* for 5 min at resuspended in 250 µl of hybridization buffer (70% formamide (Thermo Scientific, catalogue number 17899), 20 mM Tris (Thermo Scientific, catalogue number AM9850G) and 0.1% BSA in water) containing 0.3 µg ml$^{-1}$ TelC-Alexa647 (PNA Bio F1013) and 0.3 µg ml$^{-1}$ CENPB-Alexa488 (PNA Bio, catalogue number F3004) PNA probes which had been heated briefly at 55 °C for 5 min and vortexed before addition. Cells were heated at 80 °C for 10 min and incubated overnight at room temperature in the dark.

The following morning, cells were centrifuged at 300*g* for 7 min at 16 °C and resuspended gently in 1 ml of formamide wash buffer (70% formamide, 10 mM Tris, 0.1% Tween 20 (Sigma, catalogue number P1379) and 0.1% BSA in water). This step was repeated once more. After this, cells were centrifuged at 300*g* for 7 min at 16 °C and resuspended gently in 1 ml of PBS wash buffer (PBS supplemented with 0.1% Tween 20 and 0.1% BSA). Finally, cells were centrifuged at 300*g* for 5 min at 16 °C, resuspended in 500 ml of FACS buffer supplemented with 10 mg ml$^{-1}$ RNase A (Invitrogen, catalogue number 12091021) and transferred to FACS tubes through a 40-µm cell strainer (Fisher, catalogue number 22363547). Cells were sorted using a BD FACSAria Fusion flow cytometer. For each sample, a small proportion of cells were analyzed to give the distribution of telomere lengths within that sample and then sorting gates were set by the specified percentile telomere length ranges.

DNA was extracted from sorted populations using PicoPure and quantified as described above. Purified DNA was prepared for Sanger sequencing (as described above, PCR annealing temperature optimized for each primer pair) alongside targeted amplicon sequencing (Supplementary Table 17; Supplementary Methods).

### Reporting summary

Further information on research design is available in the Nature Portfolio Reporting Summary linked to this article.

### Data availability

UKB data is publicly available, and access can be requested at https://www.ukbiobank.ac.uk. All of Us data can be accessed at https://researchallofus.org/. WGS data used to generate hematopoietic phylogenies is available at the European Genome–phenome Archive (EGA) with accession EGAS00001004280 (https://ega-archive.org/studies/EGAS00001004280). Targeted amplicon sequencing data has been uploaded to SRA (PRJNA1121075). Additional data is included as Supplementary Tables. Source data are provided with this paper.

### Code availability

No custom software, tools or packages were used in the UKB analysis. The code used to generate phylogenies from single-HSPC-derived colonies is available on GitHub: SNVs were called using the cancer variants through expectation maximization (CaVEMan) algorithm (https://github.com/cancerit/CaVEMan), indels were called using Pindel (https://github.com/cancerit/cgpPindel) and allele counts at SNV/indel sites were calculated using vafCorrect (https://github.com/cancerit/vafCorrect). Rtreemut and rtreefit, used to assign mutations to branches and infer temporal branch lengths/mutation rates, respectively, are available from https://github.com/nangalialab/treemut and https://github.com/nangalialab/rtreefit. Telomerecat, used to estimate colony telomere lengths from WGS, can be accessed from: (https://github.com/cancerit/telomerecat). The pairwise comparisons of splicing mutant clades, construction of linear mixed effects models and analysis of TERTp mutations from UKB WGS data were carried out using custom R scripts available at: https://github.com/billydunn/telomeres-ch (ref. 67). Python code used to perform the primary LTL-CH and PRS-CH association analyses in the UKB and the R script for MR analysis are available at https://github.com/cksruthi/telomeres-CH (ref. 68). Scripts used for the analysis of targeted sequencing data can also be found at https://github.com/cksruthi/telomeres-CH.

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

## Acknowledgements

We thank the participants and investigators involved in the UKB resource and in the other genome-wide association studies cited in this work who collectively made this research possible. This work was funded by a Consolidator Award from the European Research Council (819956), an Early Detection Project Grant from Cancer Research UK (EDDCPJT\100010) and a joint Leukemia and Lymphoma Society—Blood Cancer UK Specialized Centre of Research Grant (7035-24) awarded to G.S.V. The Cambridge Stem Cell Institute is supported by the Wellcome Trust (203151/Z/16/Z, 203151/A/16/Z) and the UKRI Medical Research Council (MC_PC_17230). M.A.M. is funded by a Wellcome Trust PhD Studentship (102160/B/13/Z). W.G.D. is funded by a Clinical Research Fellowship from the Cancer Research UK Cambridge Centre (CTRQQR-2021\100012). S.K. is supported by a UK Research and Innovation (UKRI) Future Leaders Fellowship (MR/T043202/2). P.M.Q. is supported by the Ramon y Cajal Program

(RYC2022-036793-I) funded by MICIU/AEI/10.13039/501100011033 and cofunded by FSE+, and his work is funded by ISCIII (PI22/00218) cofunded by the EU. C.B. is supported by the Ramon y Cajal program (RYC2021-031291-I) and funded by MICU/AEI/10.13039/50100011033 and cofunded by European Union NextGenerationEU/PRTR. A.J.W. is supported by Cancer Research UK (DRCNPG-Jun24/100002), the UK Medical Research Council (UKRI1443) and the Rosetrees Trust. Y.P. is supported by the National Institutes of Health (T32 GM007347). A.G.B. is supported by a Burroughs Wellcome Fund Career Award for Medical Scientists, a Pew Charitable Trusts and Alexander and Margaret Stewart Trust Pew-Stewart Scholar for Cancer Research award and a Hevolution/AFAR New Investigator Award in Aging Biology and Geroscience Research. G.S.V. is supported by a Cancer Research UK Senior Cancer Fellowship (C22324/A23015) and work in his laboratory is also funded by the Kay Kendall Leukemia Fund, AstraZeneca, Blood Cancer UK and the Wellcome Trust. This research was supported by the Cambridge NIHR BRC Cell Phenotyping Hub, Cambridge Blood and Stem Cell Biobank and the Manchester Cancer Research Centre Biobank. This research was conducted using the UKB resource under approved application 56844.

## Author contributions

G.S.V. conceived, designed and supervised the study. M.A.M. designed and performed the majority of experiments. J.T., L.B., R.W. and G.V. performed additional experimental work. S.C.K. and M.G. performed UKB analysis. Y.P. and A.G.B. performed All of Us analysis. W.G.D, N.W., D.L. and K.D. performed phylogenetic analysis. S.C.K, W.G.D., M.G., and S.W. performed additional bioinformatics analysis. N.W., D.L., K.D., V.S., L.M., S.W., J.R., A.D., M. Gozdecka, X.L., C.B., J. Nomdedeu, P.C., I.D.D., J.G.dA., I.M., M.F., P.M.Q., M. Gerstung, J.M., J. Nangalia and S.K. provided technical advice and expertise. E.F., V.O., T.M., T.F., A.J.W., C.W., G.F., A.L.G., E.G., M.C., F.C., E.M., E.J.B. and D.W. provided patient material used in the study. G.S.V., M.A.M., S.C.K and W.G.D. prepared the paper and all authors reviewed the final paper.

## Competing interests

G.S.V. is a consultant to STRM.BIO and holds a research grant from AstraZeneca for research unrelated to that presented here. M.A.F., S.W. and J.M. are employees and stockholders of AstraZeneca. A.J.W. is a consultant for SDS Therapeutics. The other authors declare no competing interests.

## Additional information

**Extended data** is available for this paper at https://doi.org/10.1038/s41588-025-02296-x.

**Correspondence and requests for materials** should be addressed to George S. Vassiliou.

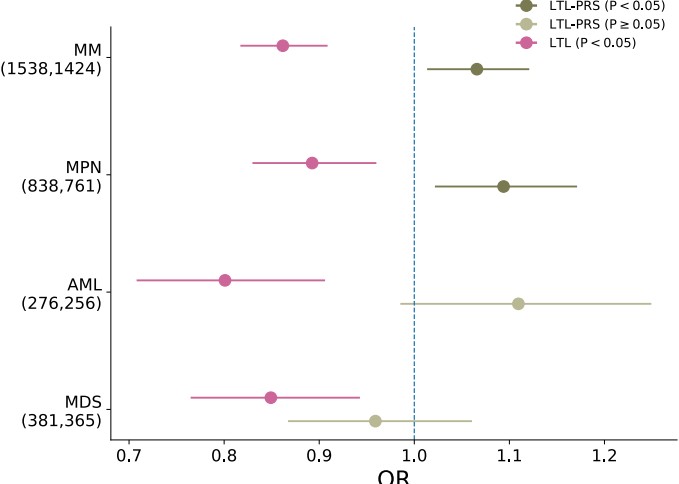

**Extended Data Fig. 1 | Association of telomere length with myeloid malignancy risk.** Odds ratio (OR) of developing different myeloid malignancies with every 1 standard deviation increase in leukocyte telomere length polygenic risk score (LTL-PRS) and measured telomere length (LTL). OR was calculated using logistic regression in which the outcome variable is the development of myeloid malignancy, and the input variable is LTL or LTL-PRS along with the covariates sex, age, smoking status and the first ten genetic principal components. Dots represent the estimated OR and error bars represent the 95% CI for each OR. Color indicates ORs with a P < 0.05 (pink for LTL, dark green for LTL-PRS) or P≥0.05 (light green). 'MM' represents myeloid malignancies, namely AML, MDS, MPN and CMML. The numbers of UK Biobank participants with myeloid malignancy included in the LTL-PRS and LTL association analyses are indicated in brackets, respectively.

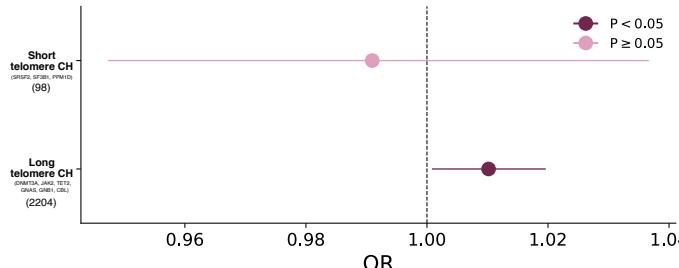

**Extended Data Fig. 2 | Association of paternal age at birth with clonal hematopoiesis risk.** Odds ratio (OR) of developing CH per one year increase in paternal age at birth. 'Long Telomere CH' refers to CH mutations that are more common amongst individuals with longer genetically predicted telomere length (PRS), namely *DNMT3A*, *JAK2*, *TET2*, *GNAS*, *GNB1* and *CBL* (n = 2204) and 'Short Telomere CH' refers to CH mutations that are more common amongst individuals with shorter genetically predicted telomere length (PRS), namely *PPM1D*, *SF3B1* and *SRSF2* (n = 98). Dots represent the estimated OR and error bars represent the 95% CI for each OR. OR was calculated using logistic regression in which the outcome variable is the development of CH, and the input variable is paternal age at birth along with the covariates sex, age, smoking status and the first ten genetic principal components. Color indicates ORs with a P < 0.05 (purple) or P≥0.05 (pink). Number of UKB participants in each of the 'Long Telomere CH' and 'Short Telomere CH' groups is shown in brackets.

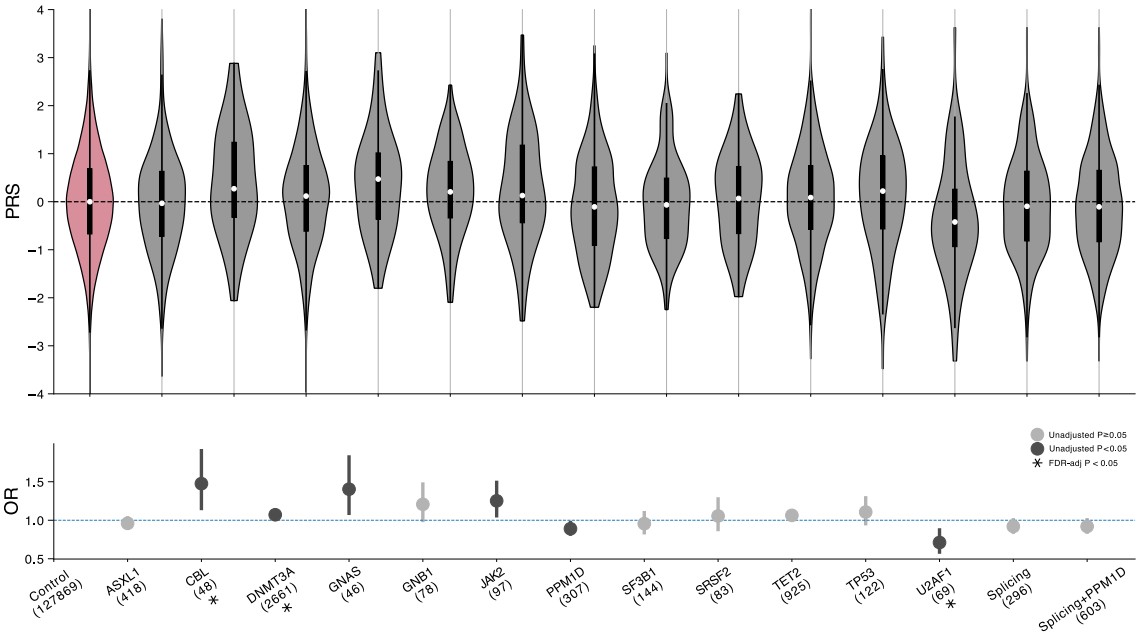

**Extended Data Fig. 3 | Distribution of UKB-derived LTL-PRS scores amongst All of Us participants.** Upper panel: Violin plots of genetically determined LTL-PRS derived from the UKB[24] and applied to All of Us participants (top panel). White dots and black boxes mark the LTL-PRS median and interquartile range (IQR), respectively. Whiskers extend to the lowest and highest data points within Q1-1.5×IQR and Q3 + 1.5×IQR where Q1 and Q3 represent the first and third quartiles, respectively. The control group includes all participants without any CH mutation. Y-axis is limited to the range [−4,4] to allow for better visualization of PRS differences. Lower panel: Odds ratio (OR) of developing different CH subtypes per 1 standard deviation increase in LTL-PRS obtained using linear regression models. Dots represent the estimated OR and bars represent the 95% CI for each OR. Color indicates ORs with an unadjusted P < 0.05 (dark grey) or unadjusted P≥0.05 (light grey). ORs for genes with an FDR-adjusted P < 0.05 are indicated with an asterisk (*). FDR correction was performed using the Benjamini-Hochberg procedure. The number of individuals in each group is indicated in brackets next to the gene names.

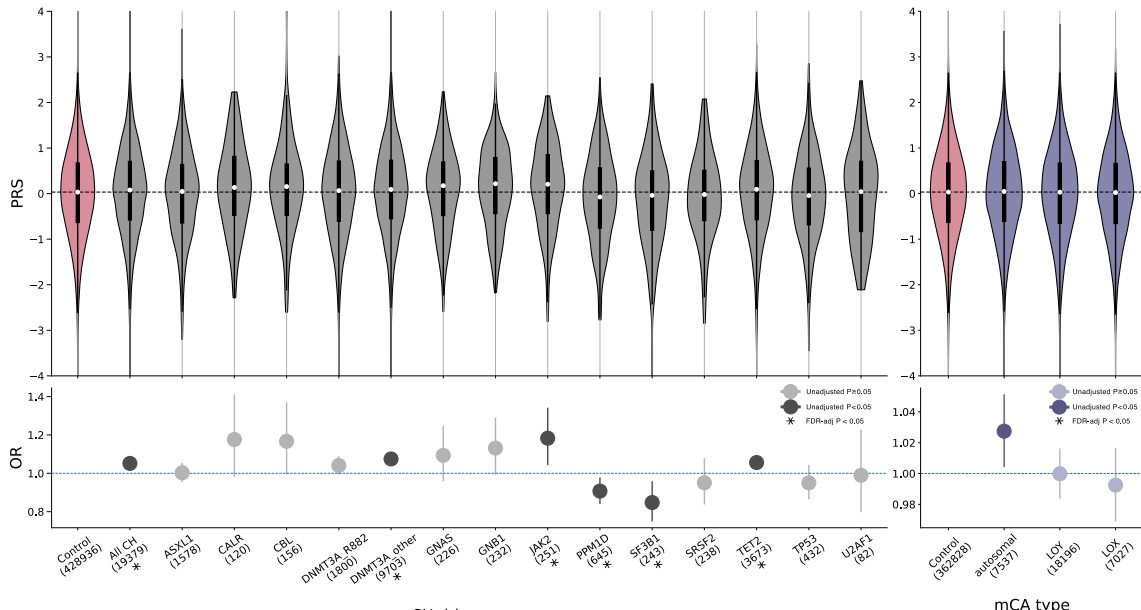

**Extended Data Fig. 4 | Distribution of TOPMed-derived LTL-PRS scores amongst UKB participants.** Upper panel: Distribution of TOPMed-derived LTL-PRS scores[27] amongst UKB participants with or without different types of CH and mCA. White dots and black boxes mark the LTL-PRS median and interquartile range (IQR), respectively. Whiskers extend to the lowest and highest data points within Q1-1.5×IQR and Q3 + 1.5×IQR where Q1 and Q3 represent the first and third quartiles, respectively. The control group includes all participants without any CH mutation or mCA. Lower panel: Odds ratio (OR) of developing different CH/ mCA subtypes per 1 standard deviation increase in LTL-PRS with values derived from a linear regression model. Dots represent the estimated OR and bars represent the 95% CI for each OR. Color indicates ORs with an unadjusted P < 0.05 (dark grey, dark purple) or unadjusted P≥0.05 (light grey, light purple). ORs for genes with an FDR-adjusted P < 0.05 are indicated with an asterisk (*). FDR correction was performed using the Benjamini-Hochberg procedure. The number of individuals in each group is indicated in brackets next to the gene names.

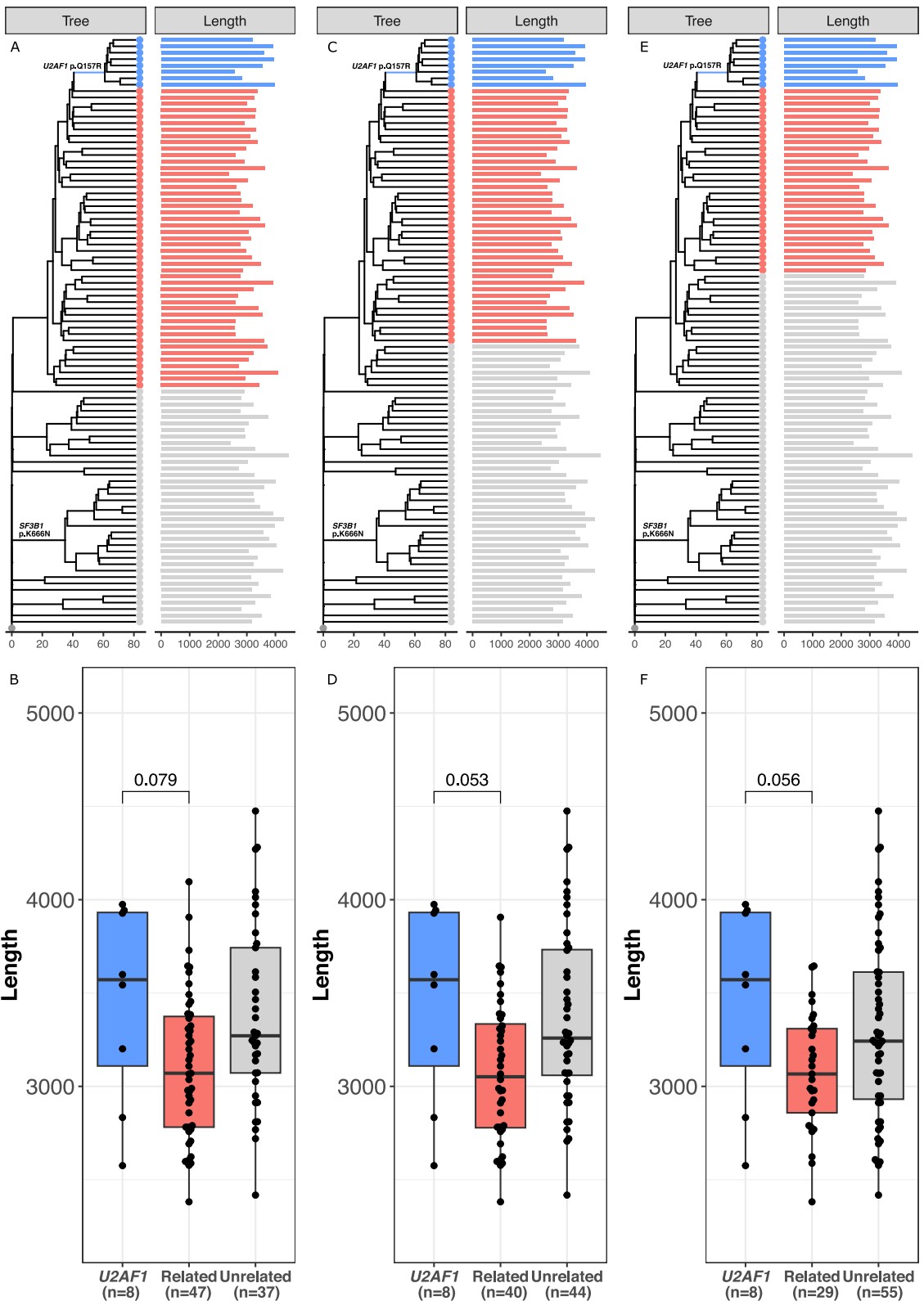

**Extended Data Fig. 5 | Telomere lengths in U2AF1-mutant and closely related non-mutant clades.** Comparison of telomere lengths of HSPC-derived colonies from the *U2AF1*-mutant clade (blue) in individual PD34493 with iteratively more closely related colonies of non-mutant clades (pink), showing that the observed trend for longer telomeres is maintained when comparing with more closely ancestrally related colonies, despite the *U2AF1*-mutant clade having undergone recent clonal expansion. Panels **a, c** and **e** highlight (pink) the clade being compared with the *U2AF1* clade (blue). Panels **b, d** and **f** show boxplots and p values of pairwise comparisons of NGS-estimated telomere lengths of the clades highlighted in Panels **a, c** and **e** respectively (two-sided Wilcoxon Rank Sum Test). Center line represents the median telomere length, upper and lower hinges the upper and lower quartiles respectively, and whiskers represent 1.5*the inter-quartile range. Individual data points corresponding to telomere lengths of single-HSPC derived colonies have been overlaid on each plot.

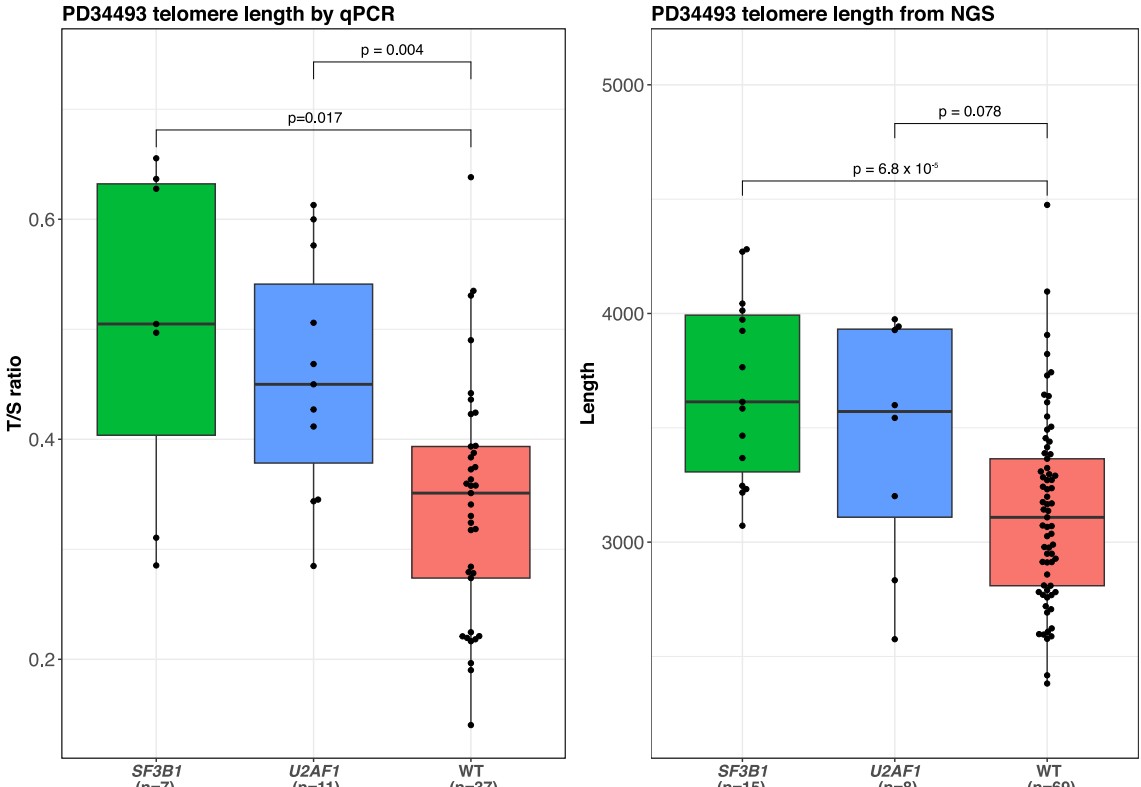

**Extended Data Fig. 6 | Orthogonal validation of NGS-estimated telomere lengths by qPCR.** Telomere lengths for colonies cultured from PD34493 were estimated using quantitative real-time PCR (qPCR) as T/S ratios (left panel). These recapitulate the pattern seen when telomere length is estimated from NGS data (right panel). By capturing more *U2AF1*-mutant colonies, the qPCR results demonstrate a significant difference in telomere length between *U2AF1*-mutant and wild-type colonies. P values were calculated using a two-sided Wilcoxon rank sum test comparing *SF3B1*- or *U2AF1*-mutant colonies to WT colonies. Center line represents the median T/S ratio or telomere length, upper and lower hinges the upper and lower quartiles respectively, and whiskers represent 1.5 X the IQR. Individual data points corresponding to T/S ratios/telomere lengths of single-HSPC derived colonies have been overlaid on each plot.

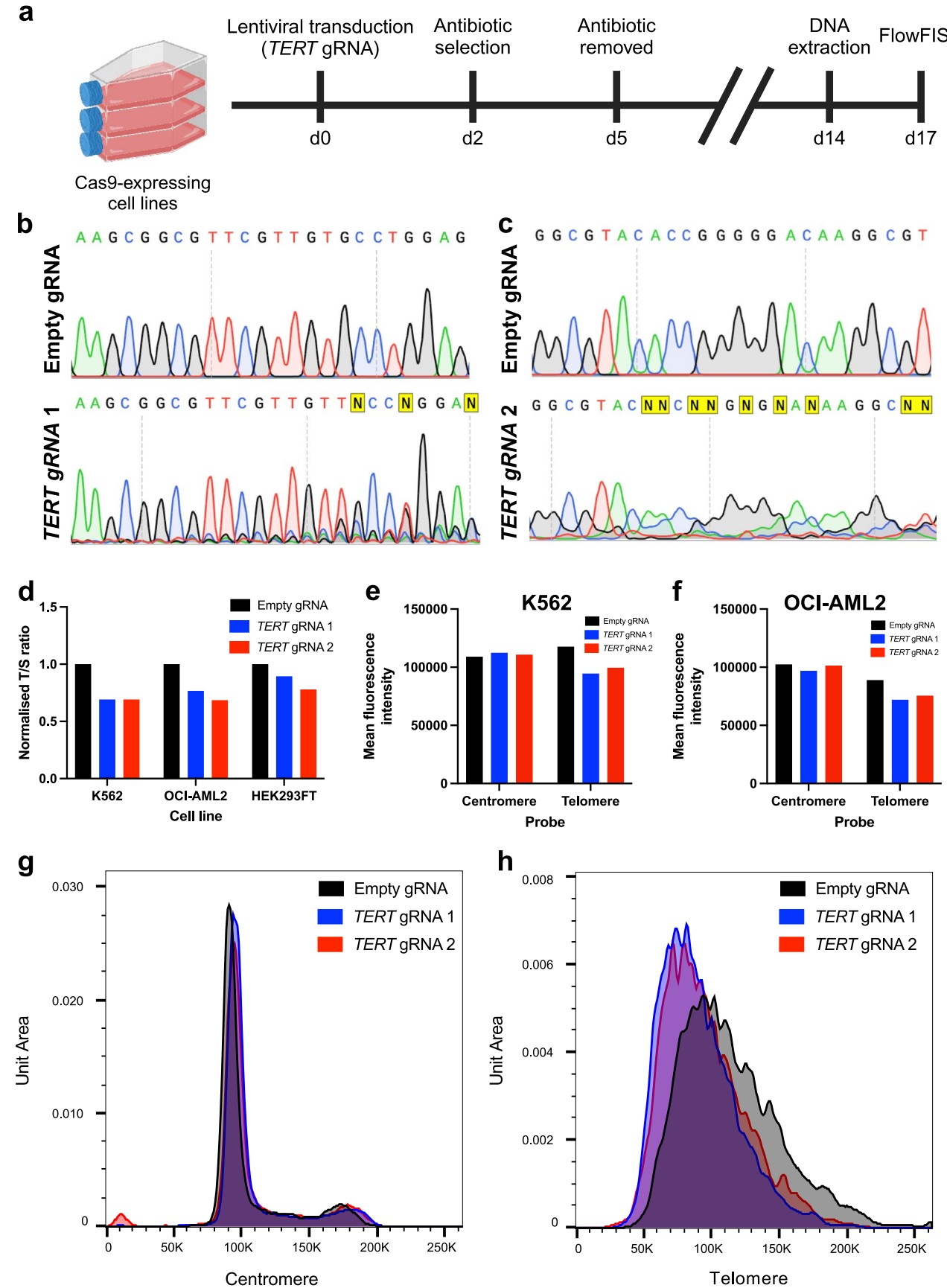

**Extended Data Fig. 7 | See next page for caption.**

**Extended Data Fig. 7 | Validation of the sensitivity of telomere qPCR and flow-FISH in *TERT* KO cells. a**, Timeline for *TERT* knockdown experiment (d = day). **b-c**, Sanger sequencing of *TERT* exon 2 from K562-Cas9 cells at day 14 following transduction with empty gRNA (top) or gRNAs targeting *TERT* exon 2 (bottom, *TERT* gRNA 1 (**b**) or *TERT* gRNA 2 (**c**)). **d**, Telomere qPCR of three Cas9-expressing cell lines (K562, OCI-AML2 and HEK293FT) 14 days after *TERT* knockout (gRNA). **e-h**, Telomere flow-FISH results at day 17 following *TERT* knockdown: **e,f**, Mean fluorescence intensity of telomere and centromere probes by flow-FISH for K562 (**e**) and OCI-AML2 (**f**). **g-h**, Example histograms for centromere (**g**) and telomere (**h**) probes in K562. Panel **a** created using BioRender.com.

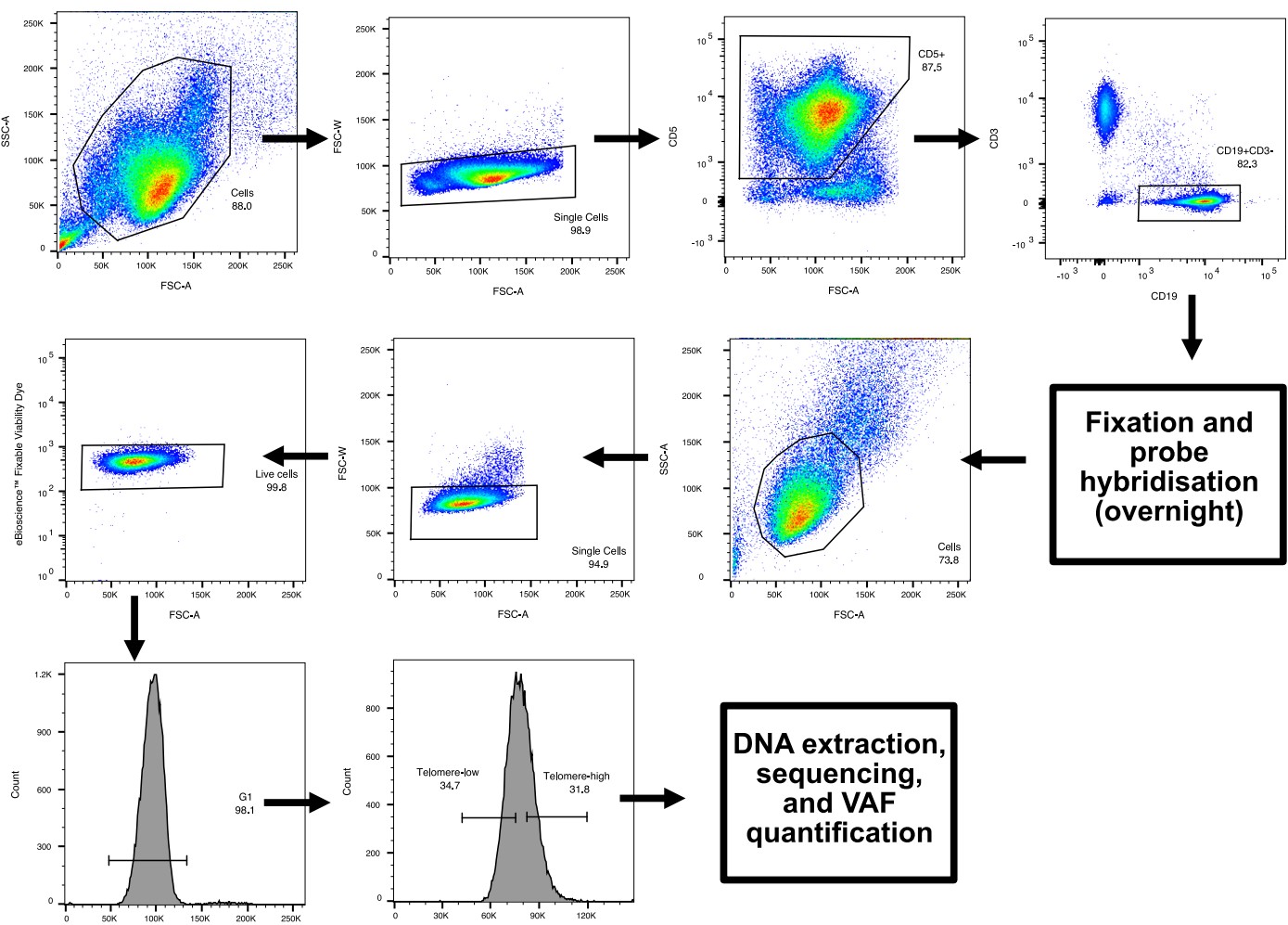

**Extended Data Fig. 8 | Isolation of CLL cells from peripheral blood.** Example gating strategy for isolating CD5+ CLL B cells with different telomere lengths from peripheral blood. Starting with mononuclear cells isolated from peripheral blood, live CD5+ CD19+ CD3− single cells were sorted, fixed and hybridized overnight with flow-FISH probes. The following day, cells in G1 (assessed by centromere staining) were sorted by telomere length into telomere^low (≤33rd percentile telomere length) and telomere^high (≥66th percentile telomere length)

fractions as depicted. Sorted cells were prepared for DNA extraction and sequencing and results are shown in Fig. 4b/c. For samples in Fig. 4d–g, isolated mononuclear cells underwent fixation and overnight probe hybridization and then G1 cells were sorted by percentile telomere length into 4 fractions (telomere^<10%, telomere^10–33%, telomere^66–90%, telomere^>90%). An increased number of fractions was used due to the availability of higher cell numbers from these samples.

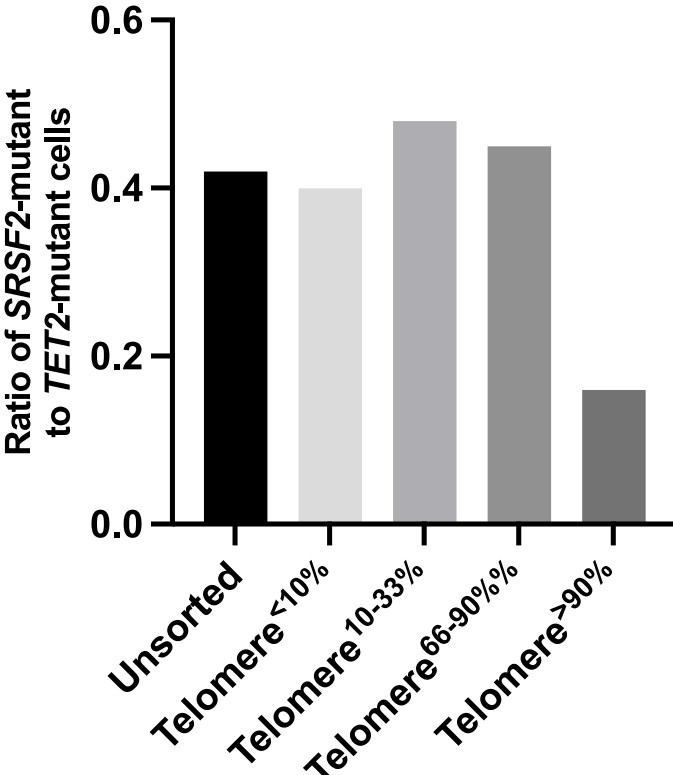

**Extended Data Fig. 9 | Patient treated with Azacitidine does not show enrichment of splicing mutant cells in telomere<sup>high</sup> cells.** Ratio of *SRSF2*-mutant to *TET2*-mutant cells in an individual with CMML following 31 cycles of Azacitidine treatment.

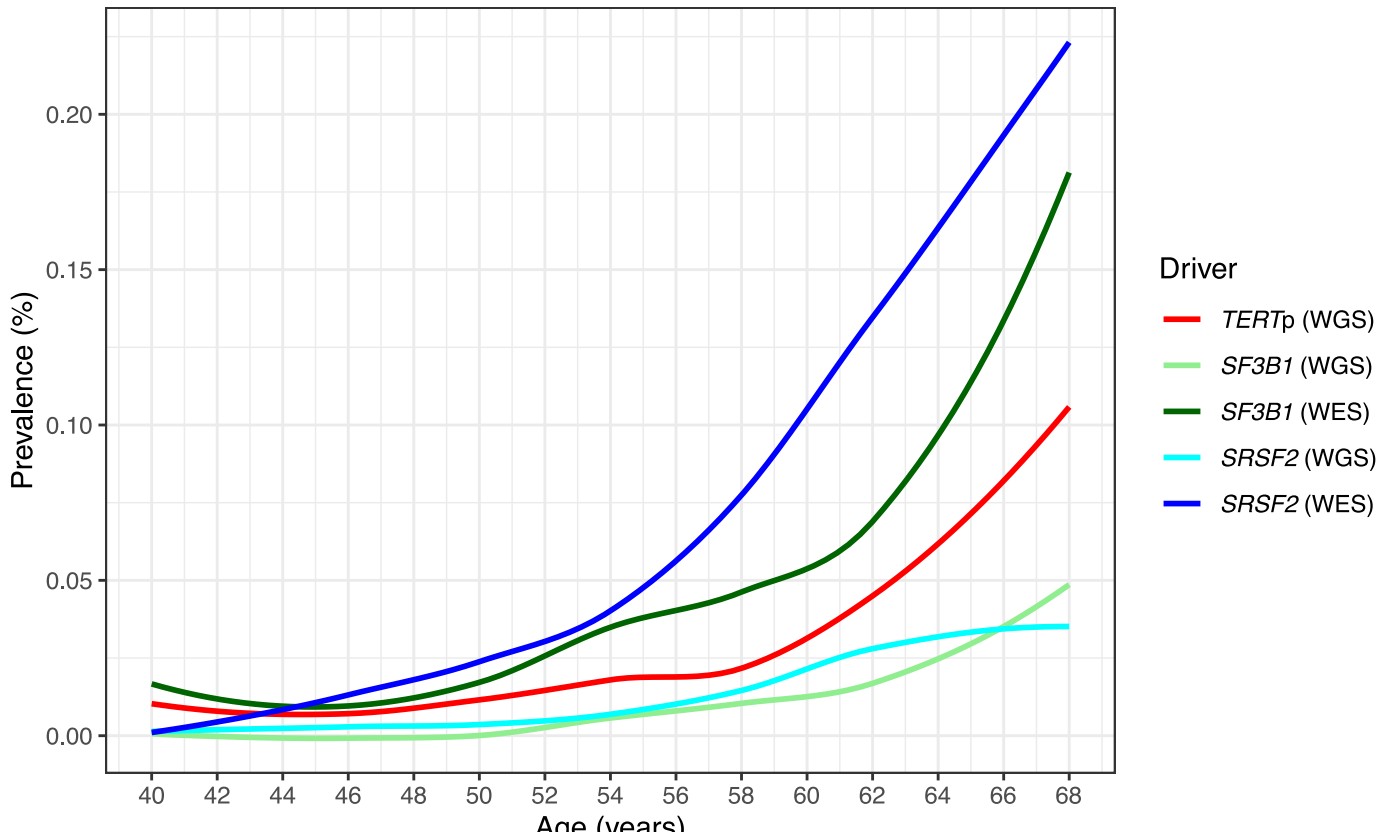

**Extended Data Fig. 10 | Age-related prevalence of *SF3B1*, *SRSF2* and *TERT*p-CH amongst UK Biobank participants.** The prevalence of *SF3B1* and *SRSF2* hotspot mutations increases sharply after the sixth decade of life. The sensitivity to detect CH clones is a function of sequencing depth and modality. Since *TERT*p is not adequately captured by WES, we identified *TERT*p-CH (red line) from blood DNA WGS data of (methodology described in Supplementary Note 2). When we call *SF3B1* and *SRSF2* hotspot mutations from WGS using the same filtering method, the prevalence of *TERT*p-CH is comparable to the combined prevalence of *SF3B1*-CH (light green line) and *SRSF2*-CH (light blue line) and exhibits a similar association with advancing age.

# Reporting Summary

## Statistics

For all statistical analyses, confirm that the following items are present in the figure legend, table legend, main text, or Methods section.

| n/a | Confirmed | |
|---|---|---|
| ☐ | ☒ | The exact sample size (*n*) for each experimental group/condition, given as a discrete number and unit of measurement |
| ☐ | ☒ | A statement on whether measurements were taken from distinct samples or whether the same sample was measured repeatedly |
| ☐ | ☒ | The statistical test(s) used AND whether they are one- or two-sided <br> *Only common tests should be described solely by name; describe more complex techniques in the Methods section.* |
| ☐ | ☒ | A description of all covariates tested |
| ☐ | ☒ | A description of any assumptions or corrections, such as tests of normality and adjustment for multiple comparisons |
| ☐ | ☒ | A full description of the statistical parameters including central tendency (e.g. means) or other basic estimates (e.g. regression coefficient) AND variation (e.g. standard deviation) or associated estimates of uncertainty (e.g. confidence intervals) |
| ☐ | ☒ | For null hypothesis testing, the test statistic (e.g. *F*, *t*, *r*) with confidence intervals, effect sizes, degrees of freedom and *P* value noted <br> *Give P values as exact values whenever suitable.* |
| ☒ | ☐ | For Bayesian analysis, information on the choice of priors and Markov chain Monte Carlo settings |
| ☒ | ☐ | For hierarchical and complex designs, identification of the appropriate level for tests and full reporting of outcomes |
| ☒ | ☐ | Estimates of effect sizes (e.g. Cohen's *d*, Pearson's *r*), indicating how they were calculated |

*Our web collection on statistics for biologists contains articles on many of the points above.*

## Software and code

Policy information about availability of computer code

| Data collection | Data collected by the UK Biobank was either downloaded or accessed through the UK Biobank research analysis platform. Data collected by the All of Us study can be accessed at https://researchallofus.org/. Data used to generate phylogenetic trees was collected from individuals with informed consent obtained from either the Cambridge Blood and Stem Cell Biobank (REC 18/EE/0199) or from the SardiNIA longitudinal study, who had single-HSPC derived colonies collected as part of a previously approved study into immunosenescence (REC 15/EE/0327). |
|---|---|
| Data analysis | We used Mutect2 GATK v. 4.1.3.0 to identify somatic variants in the CH driver genes. All UK Biobank associated linear and logistic regression analyses were performed using the Python (v. 3.9.7) module statsmodels (v. 0.12.2). Polygenic risk scores were calculated with the help of PRSice-2 (v. 2.3.5). Association between genetic variants and CH subtypes were determined by Firth's logistic regression analysis performed using the logistf function in R (v. 4.2.1) TwoSampleMR package (v. 0.5.7) in R (v. 4.3.0) was used to perform the Mendelian Randomization analysis. For phylogenetic analyses, all code used to generate phylogenies from single-HSPC derived colonies is available on GitHub: single nucleotide variants (SNVs) were called using the cancer variants through expectation maximization (CaVEMan) algorithm (https://github.com/cancerit/CaVEMan), indels were called using Pindel (https://github.com/cancerit/cgpPindel), whilst allele counts at SNV/indel sites were calculated using vafCorrect (https://github.com/cancerit/vafCorrect). Rtreemut and rtreefit, used to assign mutations to branches and infer temporal branch lengths/mutation rates respectively, are available from https://github.com/nangalialab/treemut and https://github.com/nangalialab/rtreefit. Telomerecat, used to estimate colony telomere lengths from WGS, can be accessed from: (https://github.com/telomerecat). The pairwise comparisons of splicing mutant clades and construction of linear mixed-effects models were carried out using custom R scripts available at: https://github.com/billydunn/telomeres-ch. Python code used to perform the primary LTL-CH and PRS-CH association analyses in the UKB and the R script for MR analysis are available at https://github.com/cksruthi/telomeres-CH. Scripts used for the analysis of targeted sequencing data can also be found at https://github.com/cksruthi/telomeres-CH. |

For manuscripts utilizing custom algorithms or software that are central to the research but not yet described in published literature, software must be made available to editors and reviewers. We strongly encourage code deposition in a community repository (e.g. GitHub). See the Nature Portfolio guidelines for submitting code & software for further information.

## Data

Policy information about availability of data

All manuscripts must include a data availability statement. This statement should provide the following information, where applicable:
- Accession codes, unique identifiers, or web links for publicly available datasets
- A description of any restrictions on data availability
- For clinical datasets or third party data, please ensure that the statement adheres to our policy

UKB data is publicly available, and access can be requested at https://www.ukbiobank.ac.uk. All of Us data can be accessed at https://researchallofus.org/. Targeted amplicon sequencing data has been uploaded to SRA (PRJNA1121075). Other data generated or analyzed in this study are available on reasonable request.

## Research involving human participants, their data, or biological material

Policy information about studies with human participants or human data. See also policy information about sex, gender (identity/presentation), and sexual orientation and race, ethnicity and racism.

| | |
|---|---|
| Reporting on sex and gender | For UK Biobank based analyses participants with matching self-reported and genetic sex were included. 54% of the participants were women. Sex was used as a covariate in all association analyses. For clinical samples, sex- or gender-based analysis was not performed due to the small sample size used. |
| Reporting on race, ethnicity, or other socially relevant groupings | First ten principal components (PCs) of genetic ethnicity were used as covariates in several of the UK Biobank analyses. Details on how the PCs were determined can be found at Bycroft, Clare, et al. BioRxiv (2017): 166298. For clinical samples, information on race, ethnicity and other socially relevant groupings were not available. |
| Population characteristics | For UK Biobank based analyses we used data of n=454,098 participants (age range:, 37-73, 54% female) for whom whole exome sequencing data was available as of November 2021. For clinical samples, patient characteristics are provided in the Supplementary Tables. |
| Recruitment | For clinical samples, recruitment was carried out by the Manchester Cancer Research Centre (MCRC) Biobank or the Cambridge Blood and Stem Cell Biobank (CBSB) and involved obtaining informed consent from patients undergoing investigation or treatment of a suspected/confirmed hematological condition. Details of participant recruitment for the UK Biobank can be found at https:/www.ukbiobank.ac.uk and in Sudlow C, et al. (2015) PLoS Med 12(3):e1001779. This study involves data of 454,098 (age range:, 37-73, 54% females) individuals from the UK Biobank with whole exome sequencing data released as of November 2021. |
| Ethics oversight | Clinical samples were obtained with informed consent from the Cambridge Blood and Stem Cell Biobank (CBSB) with approval from the Cambridge East Research Ethics Committee (REC: 18/EE/0199 and 24/EE/0116), from individuals in the SardiNIA longitudinal study of immunosenescence (REC 15/EE/0327) or from the Manchester Cancer Research Centre (MCRC) Biobank with approval of the South Manchester Research Ethics Committee (REC 07/H1003/161+5; HTA license 30004). The UK Biobank resource was approved by the North West Multi-centre Research Ethics Committee under reference number 21/NW/0157 and all participants provided written, informed consent to participate. Participants in the UK Biobank are volunteers and did not receive compensation for their involvement. Data for this study were accessed under approved application number 56844 from the UK Biobank resource. |

Note that full information on the approval of the study protocol must also be provided in the manuscript.

# Field-specific reporting

Please select the one below that is the best fit for your research. If you are not sure, read the appropriate sections before making your selection.

☒ Life sciences ☐ Behavioural & social sciences ☐ Ecological, evolutionary & environmental sciences

For a reference copy of the document with all sections, see nature.com/documents/nr-reporting-summary-flat.pdf

# Life sciences study design

All studies must disclose on these points even when the disclosure is negative.

| | |
|---|---|
| Sample size | For UK Biobank based analyses we used data of n=454,098 participants (age range:, 37-73, 54% females for whom whole exome sequencing data was available as of November 2021. For the analyses of clinical samples in Figures 3 and 4, we included all available samples which met our criteria. |
| Data exclusions | Individuals with no leukocyte telomere length measurements and imputed genotypic data were excluded. Additionally, participants who had withdrawn consent, or had a mismatch between genetic and self-reported sex, or had differences in the dates of attending the assessment center and the blood sample collection, were excluded from the study. |
| Replication | Replication was performed by applying the UK Biobank-derived leukocyte telomere length polygenic risk score (LTL-PRS) score to the All of Us cohort, and by applying the TOPMed-derived LTL-PRS score to the UK Biobank cohort. |

| Randomization | The experiments were not randomized. Age at recruitment, sex, smoking status, first ten genetic principal components were included as covariates in the association analysis with polygenic risk score and telomere length. WBC counts and percentages of WBC types were also included as covariates in analyses involving telomere length. |
|---|---|
| Blinding | The experiments were conducted without blinding due to feasibility. In UK Biobank analyses, CH mutations vary substantially in prevalence, such that the number of individuals with a given CH mutation can reveal the nature of the mutation and make blinding ineffective. |

# Behavioural & social sciences study design

All studies must disclose on these points even when the disclosure is negative.

| Study description | *Briefly describe the study type including whether data are quantitative, qualitative, or mixed-methods (e.g. qualitative cross-sectional, quantitative experimental, mixed-methods case study).* |
|---|---|
| Research sample | *State the research sample (e.g. Harvard university undergraduates, villagers in rural India) and provide relevant demographic information (e.g. age, sex) and indicate whether the sample is representative. Provide a rationale for the study sample chosen. For studies involving existing datasets, please describe the dataset and source.* |
| Sampling strategy | *Describe the sampling procedure (e.g. random, snowball, stratified, convenience). Describe the statistical methods that were used to predetermine sample size OR if no sample-size calculation was performed, describe how sample sizes were chosen and provide a rationale for why these sample sizes are sufficient. For qualitative data, please indicate whether data saturation was considered, and what criteria were used to decide that no further sampling was needed.* |
| Data collection | *Provide details about the data collection procedure, including the instruments or devices used to record the data (e.g. pen and paper, computer, eye tracker, video or audio equipment) whether anyone was present besides the participant(s) and the researcher, and whether the researcher was blind to experimental condition and/or the study hypothesis during data collection.* |
| Timing | *Indicate the start and stop dates of data collection. If there is a gap between collection periods, state the dates for each sample cohort.* |
| Data exclusions | *If no data were excluded from the analyses, state so OR if data were excluded, provide the exact number of exclusions and the rationale behind them, indicating whether exclusion criteria were pre-established.* |
| Non-participation | *State how many participants dropped out/declined participation and the reason(s) given OR provide response rate OR state that no participants dropped out/declined participation.* |
| Randomization | *If participants were not allocated into experimental groups, state so OR describe how participants were allocated to groups, and if allocation was not random, describe how covariates were controlled.* |

# Ecological, evolutionary & environmental sciences study design

All studies must disclose on these points even when the disclosure is negative.

| Study description | *Briefly describe the study. For quantitative data include treatment factors and interactions, design structure (e.g. factorial, nested, hierarchical), nature and number of experimental units and replicates.* |
|---|---|
| Research sample | *Describe the research sample (e.g. a group of tagged Passer domesticus, all Stenocereus thurberi within Organ Pipe Cactus National Monument), and provide a rationale for the sample choice. When relevant, describe the organism taxa, source, sex, age range and any manipulations. State what population the sample is meant to represent when applicable. For studies involving existing datasets, describe the data and its source.* |
| Sampling strategy | *Note the sampling procedure. Describe the statistical methods that were used to predetermine sample size OR if no sample-size calculation was performed, describe how sample sizes were chosen and provide a rationale for why these sample sizes are sufficient.* |
| Data collection | *Describe the data collection procedure, including who recorded the data and how.* |
| Timing and spatial scale | *Indicate the start and stop dates of data collection, noting the frequency and periodicity of sampling and providing a rationale for these choices. If there is a gap between collection periods, state the dates for each sample cohort. Specify the spatial scale from which the data are taken* |
| Data exclusions | *If no data were excluded from the analyses, state so OR if data were excluded, describe the exclusions and the rationale behind them, indicating whether exclusion criteria were pre-established.* |
| Reproducibility | *Describe the measures taken to verify the reproducibility of experimental findings. For each experiment, note whether any attempts to repeat the experiment failed OR state that all attempts to repeat the experiment were successful.* |
| Randomization | *Describe how samples/organisms/participants were allocated into groups. If allocation was not random, describe how covariates were controlled. If this is not relevant to your study, explain why.* |

| Blinding | *Describe the extent of blinding used during data acquisition and analysis. If blinding was not possible, describe why OR explain why blinding was not relevant to your study.* |
|---|---|

Did the study involve field work? ☐ Yes ☒ No

# Reporting for specific materials, systems and methods

We require information from authors about some types of materials, experimental systems and methods used in many studies. Here, indicate whether each material, system or method listed is relevant to your study. If you are not sure if a list item applies to your research, read the appropriate section before selecting a response.

## Materials & experimental systems

| n/a | Involved in the study |
|---|---|
| ☐ | ☒ Antibodies |
| ☐ | ☒ Eukaryotic cell lines |
| ☒ | ☐ Palaeontology and archaeology |
| ☒ | ☐ Animals and other organisms |
| ☒ | ☐ Clinical data |
| ☒ | ☐ Dual use research of concern |
| ☒ | ☐ Plants |

## Methods

| n/a | Involved in the study |
|---|---|
| ☒ | ☐ ChIP-seq |
| ☐ | ☒ Flow cytometry |
| ☒ | ☐ MRI-based neuroimaging |

## Antibodies

| Antibodies used | CD3-BUV395 (clone: SK7, supplier: BD Biosciences, catalogue number: 564001); CD19-BV421 (clone: HIB19, supplier: BD Biosciences catalogue number: 562440); CD5-FITC (clone:UCHT2, supplier: BD Biosciences, catalogue number: 555352); CD11b-PE (clone: ICRF44, supplier: eBioscience, catalogue number: 12-0118-42); CD33-BV510 (clone: WM53, supplier: BD Biosciences, catalogue number: 563257) |
|---|---|
| Validation | All antibodies used have had reactivity against human immunogen confirmed during quality control by the manufacturer and this is stated on the manufacturer's website. These clones have also been used in several published studies. Example references for each clone can be found below:<br>CD3 - van Dongen et al. (1988) Cytoplasmic Expression of the CD3 Antigen as a Diagnostic Marker for Immature T-Cell Malignancies. Blood 71:3.<br>CD19 - Caulier et al. (2024) CD37 is a safe chimeric antigen receptor target to treat acute myeloid leukemia. Cell Reports Medicine 5:6<br>CD5 - Li et al. (2017) Targeted Disruption of TCF12 Reveals HEB as Essential in Human Mesodermal Specification and Hematopoiesis. Stem Cell Reports 9:3<br>CD11b - Nicosia et al. (2022) Pharmacological inhibition of LSD1 triggers myeloid differentiation by targeting GSE1 oncogenic functions in AML. Oncogene 41<br>CD33 - Kyttälä et al. (2016) Genetic Variability Overrides the Impact of Parental Cell Type and Determines iPSC Differentiation Potential. Stem Cell Reports 6:2 |

## Eukaryotic cell lines

Policy information about cell lines and Sex and Gender in Research

| Cell line source(s) | The cell lines used in this study were K562 (human, female, ATCC CCL-243), HEK293-FT (human, female, ATCC CRL-1573) and OCI-AML2 (human, male, DSMZ ACC 99) |
|---|---|
| Authentication | None of the cell lines used were authenticated in this study. |
| Mycoplasma contamination | All cell lines used in the study were routinely tested for mycoplasma contamination and were negative. |
| Commonly misidentified lines (See ICLAC register) | No commonly misidentified cell lines were used in the study |

## Palaeontology and Archaeology

| Specimen provenance | *Provide provenance information for specimens and describe permits that were obtained for the work (including the name of the issuing authority, the date of issue, and any identifying information). Permits should encompass collection and, where applicable, export.* |
|---|---|
| Specimen deposition | *Indicate where the specimens have been deposited to permit free access by other researchers.* |
| Dating methods | *If new dates are provided, describe how they were obtained (e.g. collection, storage, sample pretreatment and measurement), where* |

| Dating methods | *they were obtained (i.e. lab name), the calibration program and the protocol for quality assurance OR state that no new dates are provided.* |

☐ Tick this box to confirm that the raw and calibrated dates are available in the paper or in Supplementary Information.

| Ethics oversight | *Identify the organization(s) that approved or provided guidance on the study protocol, OR state that no ethical approval or guidance was required and explain why not.* |

Note that full information on the approval of the study protocol must also be provided in the manuscript.

# Animals and other research organisms

Policy information about studies involving animals; ARRIVE guidelines recommended for reporting animal research, and Sex and Gender in Research

| Laboratory animals | *For laboratory animals, report species, strain and age OR state that the study did not involve laboratory animals.* |
| Wild animals | *Provide details on animals observed in or captured in the field; report species and age where possible. Describe how animals were caught and transported and what happened to captive animals after the study (if killed, explain why and describe method; if released, say where and when) OR state that the study did not involve wild animals.* |
| Reporting on sex | *Indicate if findings apply to only one sex; describe whether sex was considered in study design, methods used for assigning sex. Provide data disaggregated for sex where this information has been collected in the source data as appropriate; provide overall numbers in this Reporting Summary. Please state if this information has not been collected. Report sex-based analyses where performed, justify reasons for lack of sex-based analysis.* |
| Field-collected samples | *For laboratory work with field-collected samples, describe all relevant parameters such as housing, maintenance, temperature, photoperiod and end-of-experiment protocol OR state that the study did not involve samples collected from the field.* |
| Ethics oversight | *Identify the organization(s) that approved or provided guidance on the study protocol, OR state that no ethical approval or guidance was required and explain why not.* |

Note that full information on the approval of the study protocol must also be provided in the manuscript.

# Clinical data

Policy information about clinical studies

All manuscripts should comply with the ICMJE guidelines for publication of clinical research and a completed CONSORT checklist must be included with all submissions.

| Clinical trial registration | *Provide the trial registration number from ClinicalTrials.gov or an equivalent agency.* |
| Study protocol | *Note where the full trial protocol can be accessed OR if not available, explain why.* |
| Data collection | *Describe the settings and locales of data collection, noting the time periods of recruitment and data collection.* |
| Outcomes | *Describe how you pre-defined primary and secondary outcome measures and how you assessed these measures.* |

# Dual use research of concern

Policy information about dual use research of concern

## Hazards

Could the accidental, deliberate or reckless misuse of agents or technologies generated in the work, or the application of information presented in the manuscript, pose a threat to:

No | Yes

☐ ☐ Public health
☐ ☐ National security
☐ ☐ Crops and/or livestock
☐ ☐ Ecosystems
☐ ☐ Any other significant area

## Experiments of concern

Does the work involve any of these experiments of concern:

No | Yes
--- | ---
☐ | ☐ Demonstrate how to render a vaccine ineffective
☐ | ☐ Confer resistance to therapeutically useful antibiotics or antiviral agents
☐ | ☐ Enhance the virulence of a pathogen or render a nonpathogen virulent
☐ | ☐ Increase transmissibility of a pathogen
☐ | ☐ Alter the host range of a pathogen
☐ | ☐ Enable evasion of diagnostic/detection modalities
☐ | ☐ Enable the weaponization of a biological agent or toxin
☐ | ☐ Any other potentially harmful combination of experiments and agents

# Plants

| | |
|---|---|
| Seed stocks | *Report on the source of all seed stocks or other plant material used. If applicable, state the seed stock centre and catalogue number. If plant specimens were collected from the field, describe the collection location, date and sampling procedures.* |
| Novel plant genotypes | *Describe the methods by which all novel plant genotypes were produced. This includes those generated by transgenic approaches, gene editing, chemical/radiation-based mutagenesis and hybridization. For transgenic lines, describe the transformation method, the number of independent lines analyzed and the generation upon which experiments were performed. For gene-edited lines, describe the editor used, the endogenous sequence targeted for editing, the targeting guide RNA sequence (if applicable) and how the editor was applied.* |
| Authentication | *Describe any authentication procedures for each seed stock used or novel genotype generated. Describe any experiments used to assess the effect of a mutation and, where applicable, how potential secondary effects (e.g. second site T-DNA insertions, mosiacism, off-target gene editing) were examined.* |

# ChIP-seq

## Data deposition

☐ Confirm that both raw and final processed data have been deposited in a public database such as GEO.

☐ Confirm that you have deposited or provided access to graph files (e.g. BED files) for the called peaks.

| | |
|---|---|
| Data access links<br>*May remain private before publication.* | *For "Initial submission" or "Revised version" documents, provide reviewer access links.  For your "Final submission" document, provide a link to the deposited data.* |
| Files in database submission | *Provide a list of all files available in the database submission.* |
| Genome browser session<br>(e.g. UCSC) | *Provide a link to an anonymized genome browser session for "Initial submission" and "Revised version" documents only, to enable peer review.  Write "no longer applicable" for "Final submission" documents.* |

## Methodology

| | |
|---|---|
| Replicates | *Describe the experimental replicates, specifying number, type and replicate agreement.* |
| Sequencing depth | *Describe the sequencing depth for each experiment, providing the total number of reads, uniquely mapped reads, length of reads and whether they were paired- or single-end.* |
| Antibodies | *Describe the antibodies used for the ChIP-seq experiments; as applicable, provide supplier name, catalog number, clone name, and lot number.* |
| Peak calling parameters | *Specify the command line program and parameters used for read mapping and peak calling, including the ChIP, control and index files used.* |
| Data quality | *Describe the methods used to ensure data quality in full detail, including how many peaks are at FDR 5% and above 5-fold enrichment.* |
| Software | *Describe the software used to collect and analyze the ChIP-seq data. For custom code that has been deposited into a community repository, provide accession details.* |

# Flow Cytometry

## Plots

Confirm that:

☒ The axis labels state the marker and fluorochrome used (e.g. CD4-FITC).

☒ The axis scales are clearly visible. Include numbers along axes only for bottom left plot of group (a 'group' is an analysis of identical markers).

☒ All plots are contour plots with outliers or pseudocolor plots.

☒ A numerical value for number of cells or percentage (with statistics) is provided.

## Methodology

| | |
|---|---|
| Sample preparation | Cryopreserved mononuclear cells (MNCs) were thawed and washed twice in warmed RPMI supplemented with 10% FBS. For cell lines, cells were maintained in culture and transferred to a separate tube for downstream processing. Cells were centrifuged at 300g for 5 minutes and resuspended in FACS buffer (PBS supplemented with 0.1% BSA (Fisher BP9702-100)). Cells were counted and 1-3x106 cells were aliquoted into 1.5mL tubes. Cells were centrifuged at 300g for 5 minutes and resuspended in 1mL PBS containing 1:1000 Fixable Viability Dye eFluor™ 780 (eBioscience 65-0865-14) and incubated at 4°C in the dark for 20 minutes. Following this, cells were washed twice in FACS buffer. For the CLL sample only, cells were centrifuged at 300g for 5 minutes and resuspended in FACS buffer supplemented with the following antibodies: 1:100 CD3-BUV395, 1:160 CD19-BV421, 1:160 CD11b-PE, 1:100 CD33-BV510, 1:50 CD5-FITC (Supplementary Table 8). Cells were incubated at 4°C in the dark for 20 minutes and washed twice with FACS buffer and sorted as described below. Following sorting (CLL sample) or viability staining (remaining samples), cells were centrifuged at 300g for 5 minutes at resuspended in 250μL of hybridization buffer (70% formamide (Thermo Scientific 17899), 20mM Tris (Thermo Scientific AM9850G) and 0.1% BSA in water) containing 0.3μg/mL TelC-Alexa647 (PNA Bio F1013) and 0.3μg/mL CENPB-Alexa488 (PNA Bio F3004) PNA probes which had been briefly heated at 55°C for 5 minutes and vortexed prior to addition. Cells were heated at 80°C for 10 minutes and incubated overnight at room temperature in the dark. The following morning, cells were centrifuged at 300g for 7 minutes at 16°C and gently resuspended in 1mL of formamide wash buffer (70% formamide, 10mM Tris, 0.1% Tween 20 (Sigma P1379) and 0.1% BSA in water). This step was repeated once more. After this, cells were centrifuged at 300g for 7 minutes at 16°C and gently resuspended in 1mL of PBS wash buffer (PBS supplemented with 0.1% Tween 20 and 0.1% BSA). Finally, cells were centrifuged at 300g for 5 minutes at 16°C, resuspended in 500mL of FACS buffer supplemented with 10mg/mL RNase A (Invitrogen 12091021) and transferred to FACS tubes through a 40μm cell strainer (Fisher 22363547). Cells were sorted using a BD FACSAria Fusion flow cytometer. |
| Instrument | BD FACSAria Fusion |
| Software | Data was collected using BD FACSDiva software (v9.0) and analyzed in FlowJo (v10.10.0) |
| Cell population abundance | Cell population abundance was not determined post-sort. For telomere flow-FISH sorting, telomere qPCR was used to validate differences in telomere length between telomere-low and telomere-high groups |
| Gating strategy | For sorting of patient samples outlined in figure 4, the gating strategy is shown in Extended Data Fig. 6. Briefly, cells were gated using forward scatter (FSC) and side scatter (SSC). FSC-area and FSC-width were used to gate single cells. Following this, live cells were gated as Fixable Viability Dye-negative cells. In the CLL sample, additional gatings were used to first isolate CD5 + cells then CD19+CD3- cells. All samples were then gated to select cells in G1 using centromere probe then sorted by percentile telomere length. A small portion of the sample was used to establish the distribution of telomere lengths and then gates were set to sort cells in the following percentile telomere length ranges: <10%, 10-33%, 66-90%, >90%. In the CLL case, a more limited number of cells allowed only two gates to be sorted (<33%, >66%). |

☒ Tick this box to confirm that a figure exemplifying the gating strategy is provided in the Supplementary Information.

# Magnetic resonance imaging

## Experimental design

| | |
|---|---|
| Design type | *Indicate task or resting state; event-related or block design.* |
| Design specifications | *Specify the number of blocks, trials or experimental units per session and/or subject, and specify the length of each trial or block (if trials are blocked) and interval between trials.* |
| Behavioral performance measures | *State number and/or type of variables recorded (e.g. correct button press, response time) and what statistics were used to establish that the subjects were performing the task as expected (e.g. mean, range, and/or standard deviation across subjects).* |

## Acquisition

**Imaging type(s)**
*Specify: functional, structural, diffusion, perfusion.*

**Field strength**
*Specify in Tesla*

**Sequence & imaging parameters**
*Specify the pulse sequence type (gradient echo, spin echo, etc.), imaging type (EPI, spiral, etc.), field of view, matrix size, slice thickness, orientation and TE/TR/flip angle.*

**Area of acquisition**
*State whether a whole brain scan was used OR define the area of acquisition, describing how the region was determined.*

**Diffusion MRI**   ☐ Used   ☐ Not used

## Preprocessing

**Preprocessing software**
*Provide detail on software version and revision number and on specific parameters (model/functions, brain extraction, segmentation, smoothing kernel size, etc.).*

**Normalization**
*If data were normalized/standardized, describe the approach(es): specify linear or non-linear and define image types used for transformation OR indicate that data were not normalized and explain rationale for lack of normalization.*

**Normalization template**
*Describe the template used for normalization/transformation, specifying subject space or group standardized space (e.g. original Talairach, MNI305, ICBM152) OR indicate that the data were not normalized.*

**Noise and artifact removal**
*Describe your procedure(s) for artifact and structured noise removal, specifying motion parameters, tissue signals and physiological signals (heart rate, respiration).*

**Volume censoring**
*Define your software and/or method and criteria for volume censoring, and state the extent of such censoring.*

## Statistical modeling & inference

**Model type and settings**
*Specify type (mass univariate, multivariate, RSA, predictive, etc.) and describe essential details of the model at the first and second levels (e.g. fixed, random or mixed effects; drift or auto-correlation).*

**Effect(s) tested**
*Define precise effect in terms of the task or stimulus conditions instead of psychological concepts and indicate whether ANOVA or factorial designs were used.*

**Specify type of analysis:**   ☐ Whole brain   ☐ ROI-based   ☐ Both

**Statistic type for inference**
*Specify voxel-wise or cluster-wise and report all relevant parameters for cluster-wise methods.*

(See Eklund et al. 2016)

**Correction**
*Describe the type of correction and how it is obtained for multiple comparisons (e.g. FWE, FDR, permutation or Monte Carlo).*

## Models & analysis

| n/a | Involved in the study |
|-----|-----------------------|
| ☐ | ☐ Functional and/or effective connectivity |
| ☐ | ☐ Graph analysis |
| ☐ | ☐ Multivariate modeling or predictive analysis |

**Functional and/or effective connectivity**
*Report the measures of dependence used and the model details (e.g. Pearson correlation, partial correlation, mutual information).*

**Graph analysis**
*Report the dependent variable and connectivity measure, specifying weighted graph or binarized graph, subject- or group-level, and the global and/or node summaries used (e.g. clustering coefficient, efficiency, etc.).*

**Multivariate modeling and predictive analysis**
*Specify independent variables, features extraction and dimension reduction, model, training and evaluation metrics.*

