## [Peer Review File · Nature Genetics]

Telomere attrition becomes an instrument for clonal selection in aging hematopoiesis and leukemogenesis

Corresponding Author: Professor George Vassiliou

Version 0:

Decision Letter:

31st Jul 2024

Dear Professor Vassiliou,

Your Letter, "Telomere attrition becomes an instrument for clonal selection in aging hematopoiesis and leukemogenesis" has now been seen by 3 referees. You will see from their comments below that while they find your work of interest, some important points are raised. We are interested in the possibility of publishing your study in Nature Genetics, but would like to consider your response to these concerns in the form of a revised manuscript before we make a final decision on publication.

We therefore invite you to revise your manuscript taking into account all reviewer and editor comments. Please highlight all changes in the manuscript text file. At this stage we will need you to upload a copy of the manuscript in MS Word .docx or similar editable format.

*2) If you have not done so already please begin to revise your manuscript so that it conforms to our Letter format instructions, available

[here](http://www.nature.com/ng/authors/article_types/index.html).

*3) Include a revised version of any required Reporting Summary: <https://www.nature.com/documents/nr-reporting-summary.pdf>

Link Redacted

We hope to receive your revised manuscript within four to eight weeks. If you cannot send it within this time, please let us

know.

Sincerely,

Safia Danovi, PhD
Senior Editor, Nature Genetics
ORCID: 0009-0007-7822-5479

Referee expertise:

Referee #1: CH genetics

Referee #2: CH genetics

Referee #3: CH/leukemia genetics

Reviewers' Comments:

Reviewer #1:

Remarks to the Author:

In "Telomere attrition becomes an instrument for clonal selection in aging hematopoiesis and leukemogenesis", McLoughlin and colleagues pair population genetics observations in the UK Biobank with phylogenetic analysis to develop a model whereby telomere attrition acts as a selection pressure on clonal hematopoiesis. They start with the observation that clonal hematopoiesis driven by splicing factor genes and PPM1D are more common in those with shorter genetically predicted telomeres. Then, they show how clones from splicing factor driver mutations have increased telomere length compared to those with non-splicing driver mutation. Overall, the paper is a valuable addition to the field.

Analytic critiques:

- Typically in genetic studies replication across two separate cohorts is considered the gold standard. Are the authors able to replicate the core observations about either measured or predicted telomere length and CH in another biobank - for example in their recently preprinted Mexico City Prospective Study cohort (<https://doi.org/10.1101/2024.02.07.24302442>) or elsewhere (Genomics England, All of Us, TOPMed etc).
- The analysis of the interaction between rising VAF, LTL and LTL-PRS is very interesting. However, a more direct way to test this would be to, for each CH driver gene, create a linear models between: 1) $LTL \sim LTL-PRS$, 2) $LTL \sim LTL-PRS + VAF$, and 3) $LTL \sim LTL-PRS + VAF + VAF * LTL-PRS$. This would create a more quantitative model of the interaction as VAF increases rather than the descriptive categories as described. This could be a sensitivity analysis as extended data figure.
- I believe the authors have performed what is essentially "one sample" Mendelian randomization - using the same sample (UK Biobank) to identify genetic variants as instruments to estimate the effect of interest. One sample MR is more prone to data overfitting and weak instrument bias. Two sample mendelian randomization is often preferable. In this case does a LTL-PRS derived from samples other than the UK Biobank (eg Taub Cell Genomics 2022, PMID 35530816) and then applied to samples in the UK Biobank yield the same result?
- In the low vs high VAF analysis shown in Fig 2b, it would be valuable to understand if this observed discordant trend for increased LTL with increased VAF is "dose-dependent" on VAF or if it is driven by large clones with $VAF > 0.2$. For example as a sensitivity analysis for SRSF2 one could plot the LTL for three groups ($VAF 0-0.1$, $VAF 0.1-0.2$, $VAF > 0.2$).

Stylistic critiques

- In Figure 1a, it is not clear what VAF cutoff is being used to define prevalent CH here, presumably 2%. It would be helpful to include this in the methods.
- In Figure 1b and 1c, please make it more clear what the odds ratio represents - is it increased odds per increase in 1 standard deviation of the PRS? It is labeled as such in extended data fig 2. For both Fig 1b and 1c, adjust the alpha value of 0.05 to account for multiple hypothesis testing (14 CHIP gene categories and 3 mCA categories).
- Fig 1c - for the sake of clarity, would be valuable to group the "Short" (splicing and and "Long" driver genes under the short and long headers
- Fig 2a - what was the statistical cutoff here? $P=0.05$? Should account for multiple hypothesis
- I would temper this claim as a heading: "Splicing factor gene mutations enable cells to maintain their telomere length across cell divisions." Perhaps "Clones from splicing factor driver mutations have increased telomere length compared to

those with non-splicing driver mutation" might be more fitting.

- lines 203-214 "SRSF2-first" and "TET2-first" imply a temporal component which is not measured. How is which mutation occurs 'first' being established? Do any of these individuals have concurrent loss of heterozygosity?

- In Figure 3E/F - how do the authors explain the observation that the SF3B1 mutated cells have the same telomere length as the WT cells?

- In Fig 4c - the unsorted ratio of SRSF2 to TET2 was 1.0, and discordant from the ratio of the VAFs of SRSF2 to TET2. In the other plots, the unsorted bar roughly approximates the ratio of VAFs. How do the authors account for this?

- Line 248 - please clarify that "at the lower end of the spectrum" means "lower genetically predicted telomere length"

- Figure quality of Figure 4 is low resolution

-If Figure 5 is a conceptual hypothesis - it perhaps may be better as an extended data figure. The data supporting 5D and 5E may not be the most robust. 5D is from one paper that characterized 17 patients and 21 controls. The data supporting 5E is from 1 published paper of 42 patients and 2 ASH abstracts from 2021 that have not yet been published.

Reviewer #2:

Remarks to the Author:

McLoughlin et al. describe the relationship between clonal hematopoiesis (CH) in specific CH driver genes and telomere length. As most prior studies investigated CH-TL associations at the aggregate level, this study provides novel insights into CH driven by splicing factor genes suggesting telomere attrition could select for CH clones harboring splicing factor gene mutations. The study data is from both a large population and specific individuals with clinical information and repeated samples. Approaches and methods used are appropriate, conclusions are supported by the data and the manuscript clearly presents the findings. I have a few comments worth consideration for further strengthening this already terrific investigation:

-Details on CH driver and mCA calling were sparse. As small changes in calling approach or filtering methods could have a substantial impact on study results I recommend a more detailed description of the methods used to facilitate readers in evaluating the approach and promote reproducibility of future investigations.

-The Codd et al investigation used to calculate the telomere length PRS and perform the Mendelian randomization was performed in the UK Biobank. As such the same dataset was used for GWAS discovery and phenotype testing, potentially resulting in overfitting or bias. For example, CH clones in UK Biobank could cause deviations in the UK Biobank TL measures and impact TL loci identified. This in turn could impact results between CH and TL in the current analysis. A sensitivity analysis employing the Taub et al TL GWAS in TOPMed showing very similar overall results with CH would help minimize any concerns. At minimum, I suggest a discussion of the current TL loci being discovered in the same population as a study limitation.

-The population-based analyses employ multiple tests of different CH clones, but use 0.05 as the statistical significance threshold. I recommend a more conservative threshold that factors in the number of tests performed.

-It is unclear how individuals were selected for the WGS analysis of mutation carriers. Were these individuals chosen from multiple individuals with splicing factor gene mutations, or were they the only individuals with these alterations? Please better describe selection criteria and include a brief description of the mutation frequencies in the clinical sample from which the individuals for phylogenetic analyses were drawn.

-While the relationship between senescent phenotypes and PPM1D is suggested as a mechanism for the relationship between PPM1D mutations and telomere length, the relationship between PPM1D and telomere length is not explored outside of the population-level data like the splicing factor mutations. Were samples available for analysis? Were findings null?

-Analyses of cases with double mutant CH assume the same CH clone has both mutations. This might not always be the case and should be stated.

-In Figure 3 and Extended Figure 3, consider moving the mutation label to a location that does not cover the branch points on the dendrogram. It is difficult to see the estimated age at divergence when the label is covering the branch points.

-There are several discrepancies in the PD identifiers and descriptions of the WGS sequenced individuals in the Methods (pages 10-11) and Figure 3. In the Methods, there is a male aged 83.8 years with a SF3B1-mutant CH clone denoted with (PD48499), but this is the label for the male with CCUS to MDS. In Figure 3 panel a, the individual with PD34493 is listed as male, while page 11 describes PD34493 as female. Please correct.

-The abbreviation HSPCs was not defined in text.

Reviewer #3:

Remarks to the Author:

In their manuscript McLoughlin et al., study the relationship between telomere length and clonal hematopoiesis showing that while most CH mutations are associated with longer telomere lengths (as has been previously shown) specific mutations including PPM1D and splicing mutations are more common among individuals with shorter telomere lengths. As would be

predicted based on the finding of an excess of PPM1D and spliceosome CH among individuals with telomere biology disorders, they demonstrate that splicing factor CH appears to protect against telomere attrition. The absence of mechanistic data limits extrapolation to whether this might be harnessed therapeutically. Overall though the analysis was well executed and novel, the manuscript well-written with appropriate and interesting conclusions being drawn.

Major comments:

- 1) They do not present any mechanistic data regarding how spliceosome and PPM1D mutations might protect against telomere attrition and whether this plays a role in leukemogenesis. Because of this, their data does not directly support statements that their findings offer clear therapeutic approaches in splicing factor mutant myeloid neoplasms. This includes the last statement in the abstract, introduction and discussion. These statements should be modified emphasizing that future work is needed to determine whether this might have therapeutic relevance.
- 2) It is interesting in Figure 1b and c, that there is no association between TP53 CH and telomere length. Is there an association between other common DDR CH genes and telomere length including ATM and CHEK2? Similarly, it would be interesting to compare the strength of the association between telomere length and U2AF1. This could provide some additional biologic insight into differences between genes of the same class.
- 3) In their analysis of individuals with double mutant CH, they state that they used the VAF to determine mutation order. However, VAF is both a reflection of the time of mutation acquisition and mutation fitness. If the authors restricted their double hit analysis to individuals in whom mutant clones must be co-occurring based on the pigeonhole principle (using the cell fraction), VAF can be a proxy for order. However, if the clones are independent VAF cannot be used as a proxy. This should be clarified.
- 4) The authors state in the discussion that only 32 of the 640 cases of PPM1D-CH had documented history of prior chemotherapy. Is complete ascertainment of chemotherapy available in the UKBB including both oral and IV administration? Is radiation therapy data available? This would be important in determining the true number of PPM1D carriers that have no prior chemotherapy/radiation therapy.
- 5) In Figure 4, are there differences in proportion of mutant cells in the telomere high and low populations statistically significant? It is unclear whether there is formal hypothesis testing performed here.
- 6) The authors note that they did not observe CH in the TERT promoter as has been observed in TBD. Did they query the WGS of the UKBB for this region? Only analysis of the WES data is mentioned

Minor:

- 1) In the introduction, the association between SNPs at the TERT locus and CH are described as "powerful". Qualifiers on strength are normally used in regard to the OR, not the p-value. But in the Kar et al paper the top hit rs2853677 was modestly associated with CH with an OR of 1.24.
- 2) The results they show in Figure 3F for the patient with longitudinal sampling are not significant. I would advise emphasizing this on line 186-7 by emphasizing this was a "non-significant trend".
- 3) In the methods section they state that "People with mutations in less frequently mutated genes...were excluded". What is the rationale for excluding individuals with CH in less common genes?
- 4) The code used to generate all main and extended/supplemental datasets should be included.

Version 1:

Decision Letter:

Our ref: NG-LE65770R

28th Mar 2025

Dear Dr Vassiliou,

Thank you for submitting your revised manuscript "Telomere attrition becomes an instrument for clonal selection in aging hematopoiesis and leukemogenesis" (NG-LE65770R). It has now been seen by Reviewers #2 and #3 (Reviewer was unable to re-review) and their comments are below. The reviewers find that the paper has improved in revision, and therefore we'll be happy in principle to publish it in Nature Genetics, pending minor revisions to satisfy the referees' final requests and to comply with our editorial and formatting guidelines.

Sincerely,

Safia Danovi, PhD
Senior Editor, Nature Genetics
ORCID: 0009-0007-7822-5479

Reviewer #2 (Remarks to the Author):

I commend the authors on a thorough response that adequately addressed my concerns. I have no further comments.

Reviewer #3 (Remarks to the Author):

The additional analyses in the revised manuscript are very interesting and well done. In particular the finding of TET2 promotor mutations is quite novel and I agree adds to the insightful conclusions that can be drawn from this important contribution to the field.

My only comment is in regards to the mutational occurrence analysis. The authors state that it is unlikely that two CH mutations would be present in separate clones. However it has been shown by Ross Levine's group using single cell sequencing that multiple CH mutations in the same individual are generally present in different clones (Miles et al., Nature 2020). Thus I do not think this is a reasonable assumption. Furthermore, if SRSF2 mutations were in a separate clone from TET2 and if SRSF2 mutant CH was present at a higher VAF than TET2 CH (and therefor impacting a larger number of cells), one would expect to observe longer telomere length in the bulk DNA than in the converse scenario when TET2 was impacting a larger number of cells. Because of this, I do not agree with the statement that any misclassification would bias their results towards the null. One solution would be to limit the analysis to individuals where the sum of the VAF of multiple mutations is greater than 50%. Another would be to simply remove this as I don't believe it adds much to this excellent manuscript.

Kelly Bolton

Reviewer #1:

Remarks to the Author:

In "Telomere attrition becomes an instrument for clonal selection in aging hematopoiesis and leukemogenesis", McLoughlin and colleagues pair population genetics observations in the UK Biobank with phylogenetic analysis to develop a model whereby telomere attrition acts as a selection pressure on clonal hematopoiesis. They start with the observation that clonal hematopoiesis driven by splicing factor genes and *PPM1D* are more common in those with shorter genetically predicted telomeres. Then, they show how clones from splicing factor driver mutations have increased telomere length compared to those with non-splicing driver mutation. Overall, the paper is a valuable addition to the field.

We thank the reviewer for the succinct overview of our study and for finding it a valuable addition to the field.

Analytic critiques:

1.1. Typically in genetic studies replication across two separate cohorts is considered the gold standard. Are the authors able to replicate the core observations about either measured or predicted telomere length and CH in another biobank - for example in their recently preprinted Mexico City Prospective Study cohort (<https://doi.org/10.1101/2024.02.07.24302442>) or elsewhere (Genomics England, All of Us, TOPMed etc).

We agree with the reviewer that replication of our genetic findings in an additional cohort would greatly strengthen our manuscript. As the prevalence of splicing-factor (SF)-mutant CH in the Mexico City Prospective Study was low¹ and most individuals were of Indigenous American ancestry², we used the larger and more genetically diverse All of Us cohort³ for this analysis.

By applying the UK Biobank (UKB) LTL-PRS to the All of Us cohort (**Figure R1**), we found a significantly higher LTL-PRS amongst carriers of *DNMT3A*-CH, *JAK2*-CH, *CBL*-CH, and *GNAS*-CH, and a significantly lower LTL-PRS amongst those with *PPM1D*-CH and *U2AF1*-CH (*U2AF1* was added to the analysis in response to a request from reviewer #3 - comment 3.2). However, only the associations between UKB LTL-PRS and *DNMT3A*-CH, *CBL*-CH, and *U2AF1*-CH retained significance in the All of Us cohort following correction for multiple testing (see response to comment 1.6). We did not find significant associations for other CH subtypes, likely due to the much smaller numbers of cases in the All of Us cohort. This may have limited our ability to detect differences in LTL-PRS, as proposed by our power analysis outlined in response to comment 3.9. It is also possible that the use of an LTL-PRS derived from the UKB, which is predominantly European in genetic ancestry, may be slightly less able to accurately reflect LTL when applied to a cohort with more diverse genetic ancestry such as All of Us.

Note that *U2AF1*-CH was not included in our initial submission due to an insufficient number of cases, but is included in the revised manuscript as this was specifically

requested by reviewer #3 (see response to comment 3.2) and also because of the recent finding that it is a frequent driver of CH in telomere biology disorders (TBDs)⁴.

Overall, our analysis of the All of US cohort supports our core observations that higher LTL-PRS is associated with the most common forms of CH (e.g. *DNMT3A*, *JAK2*), whereas *PPM1D*-CH and SF-CH was associated with lower LTL-PRS. We thank the reviewer for this suggestion and now include these data in Extended Data Figure 3 and discuss the results in the text (lines 128-135).

Figure R1. Distribution of UKB-derived LTL-PRS scores amongst All of Us participants.

Upper panel: Violin plots of genetically determined LTL-PRS derived from the UKB⁵ and applied to All of Us participants (top panel). White dots and black boxes mark the LTL-PRS median and interquartile range (IQR), respectively. Whiskers extend to the lowest and highest data points within $Q1-1.5 \times IQR$ and $Q3+1.5 \times IQR$ where Q1 and Q3 represent the first and third quartiles, respectively. The control group includes all participants without any CH mutation. Y-axis is limited to the range [-4,4] to allow for better visualization of PRS differences. Lower panel: Odds ratio (OR) of developing different CH subtypes per 1 standard deviation increase in LTL-PRS. Bars represent the 95% CI for each OR. ORs for genes with an unadjusted $P < 0.05$ are depicted in dark grey and those with unadjusted $P \geq 0.05$ in light grey. ORs for genes with an FDR-adjusted $P < 0.05$ are indicated with an asterisk (*). FDR correction was performed using the Benjamini-Hochberg procedure. The number of individuals in each group is indicated in brackets next to the gene names.

1.2. The analysis of the interaction between rising VAF, LTL and LTL-PRS is very interesting. However, a more direct way to test this would be to, for each CH driver gene,

create a linear models between: 1) $LTL \sim LTL\text{-}PRS$, 2) $LTL \sim LTL\text{-}PRS + VAF$, and 3) $LTL \sim LTL\text{-}PRS + VAF + VAF * LTL\text{-}PRS$. This would create a more quantitative model of the interaction as VAF increases rather than the descriptive categories as described. This could be a sensitivity analysis as extended data figure.

We thank the reviewer for this suggestion and have developed gene-specific linear models to address this. In the model $LTL \sim LTL\text{-}PRS + VAF$, the VAF term was a statistically significant coefficient for *DNMT3A_R882*, *DNMT3A_other*, *TET2*, *PPM1D*, *ASXL1*, *SRSF2*, *CALR* and *JAK2*. When including an interaction term in the model ($LTL \sim LTL\text{-}PRS + VAF + LTL\text{-}PRS * VAF$), none of the interaction terms reached statistical significance in any of our gene-specific models following correction for multiple testing (Supplementary Table 10). Moreover, model performance (assessed by adjusted R^2) was not improved by inclusion of an interaction term. This suggests that LTL-PRS and VAF have largely independent effects on measured LTL.

We have also performed a sensitivity analysis assessing the change in predicted LTL for difference combinations of LTL-PRS and VAF by building individual gene models for four CH genes (*DNMT3A*, *TET2*, *PPM1D*, *SRSF2*) (**Figure R2**). This mirrors the observations shown in Figure 2, wherein LTL decreases with rising VAF for *DNMT3A*, *TET2* and *PPM1D*, but increases with rising VAF for *SRSF2*.

We appreciate the reviewer's suggestions and have included the outputs from these models as Supplementary Table 10 and discussed them in the main text (line 177-181). However, we have opted not to include the sensitivity analysis in the manuscripts as it is similar and does not add to the data shown in Figure 2.

Figure R2. Effect of LTL-PRS and VAF on predicted LTL using sensitivity analysis. Individual gene models were built for LTL as $LTL \sim PRS + VAF$. Sensitivity of predicted LTL to variation in LTL-PRS and VAF was assessed by variation of input values whilst keeping all continuous covariates at their mean values. The color scale shows the mean LTL prediction from several models using different values of the following categorical variables: sex, smoking status.

1.3. I believe the authors have performed what is essentially "one sample" Mendelian randomization - using the same sample (UK Biobank) to identify genetic variants as instruments to estimate the effect of interest. One sample MR is more prone to data overfitting and weak instrument bias. Two sample mendelian randomization is often preferable. In this case does a LTL-PRS derived from samples other than the UK Biobank (eg Taub Cell Genomics 2022, PMID 35530816) and then applied to samples in the UK Biobank yield the same result?

We thank the reviewer for this excellent suggestion and have repeated the analysis shown in Figure 1 of our manuscript, now applying the LTL-PRS derived from the TOPMed database⁶ to individuals in the UK Biobank (**Figure R3**). This replicated many significant associations between CH and either increased LTL-PRS (All CH, *DNMT3A*_other, *JAK2*, *TET2*) or decreased LTL-PRS (*SF3B1*, *PPM1D*). These findings endorse those derived using the UKB-derived LTL-PRS, with the lack of statistical significance in some genes likely resulting from: i) the small numbers of cases in some categories and ii) the diverse genetic ancestry of TOPMed participants, making the TOPMed-derived LTL-PRS a slightly less apt estimate of LTL-PRS amongst the predominantly European UKB participants.

We now include this analysis as Extended Data Figure 4 and discuss the results in the text (lines 135-142).

Figure R3. Distribution of TOPMed-derived LTL-PRS scores amongst UKB participants.

Boxplots showing variation in measured telomere length by CH subtype and clonal size. Telomere length was adjusted for several covariates (sex, age, smoking status, genetic principal components from 1 to 10, WBC counts and percentages of types of WBC) using multiple variable regression. Individuals within each CH subgroup were divided into those with VAF < 0.1 (dark blue), VAF 0.1-0.2 (turquoise), and VAF > 0.2 (green). Red lines and boxes mark the median and interquartile range, respectively. Whiskers extend to the lowest and highest data points within $Q1 - 1.5 \times (Q3 - Q1)$ and $Q3 + 1.5 \times (Q3 - Q1)$ where Q1 and Q3 represent the first and third quartiles, respectively. The control group includes all participants without any CH mutation. Number of mutation carriers in each group are shown in brackets below the gene names. Horizontal bars and "*" indicates groups with significant differences (adjusted P < 0.05) in LTL with P values derived by pairwise Wilcoxon rank sum tests with Bonferroni correction for multiple testing within each CH

subtype. Y-axis is limited to the range [-5,5] to allow for better visualization of LTL differences.

1.4. In the low vs high VAF analysis shown in Fig 2b, it would be valuable to understand if this observed discordant trend for increased LTL with increased VAF is "dose-dependent" on VAF or if it is driven by large clones with VAF>0.2. For example as a sensitivity analysis for SRSF2 one could plot the LTL for three groups (VAF 0-0.1, VAF 0.1-0.2, VAF >0.2).

We thank the reviewer for this comment and have performed the suggested analysis with the results depicted in **Figure R4**. These show that certain CH subtypes (*ASXL1*, *PPM1D*, and *TET2*) displayed a dose-dependent association, with their LTLs decreasing significantly between consecutive clone size bins (i.e. VAF<0.1 vs 0.1-0.2 and VAF 0.1-0.2 vs >0.2). For other CH subtypes (*CALR*, *DNMT3A_R882*, *DNMT3A_other*, *JAK2* & *SRSF2*), the LTLs of the two smaller VAF bins (VAF<0.1 vs 0.1-0.2) did not differ significantly, but those of the largest bin (VAF>0.2) differed significantly, in keeping with the premise that large clones drove the observed LTL trends (i.e. decreased LTL for *DNMT3A_R882* & *JAK2* and increased LTL for *SRSF2*). We now include this analysis as **Extended Data Figure 5** and discuss the results in the text (line 163-165).

Figure R4. Impact of increasing VAF on LTL by CH subtype.

Boxplots showing variation in measured LTL by CH subtype and VAF. Individuals within each CH subgroup were divided into bins with VAF<0.1 (dark blue), VAF 0.1-0.2 (turquoise), and VAF>0.2 (green). Red lines and boxes mark the LTL median and IQR, respectively. Whiskers extend to the lowest and highest data points within $Q1-1.5 \times IQR$ and $Q3+1.5 \times IQR$ where $Q1$ and $Q3$ represent the first and third quartiles, respectively. The control group is composed of all participants without any CH mutation or mCA. The number of mutation carriers in each group is shown in brackets below the gene names. Horizontal bars and “*” indicates groups with significant differences (adjusted $P < 0.05$) in

LTL with P-values derived by pairwise Wilcoxon rank sum tests with Bonferroni correction for multiple testing within each CH subtype.

Stylistic critiques

1.5. In Figure 1a, it is not clear what VAF cutoff is being used to define prevalent CH here, presumably 2%. It would be helpful to include this in the methods.

We included all mutations identified by Mutect2 (without a VAF cut-off) or pile-up (in case of *U2AF1* only) and passing relevant quality and mutation filters as described previously⁷. Samtools mpileup (v. 1.15.1) was used to identify *U2AF1* mutations and the variants with a minimum of 3 alternate allele reads and a $VAF \geq 0.05$ were included in the analysis. Whilst we did not use a VAF cut-off for the remaining genes, only 50 of 23,179 calls had a $VAF < 0.2$. Of these, 45 had a VAF of 0.17-0.19. We now clarify the absence of VAF cut-off in the Methods (lines 426-427).

1.6. In Figure 1b and 1c, please make it more clear what the odds ratio represents - is it increased odds per increase in 1 standard deviation of the PRS? It is labeled as such in extended data fig 2. For both Fig 1b and 1c, adjust the alpha value of 0.05 to account for multiple hypothesis testing (14 CHIP gene categories and 3 mCA categories).

Our reviewer is correct, and the odds ratio represents the odds per 1 standard deviation increase in the PRS, as we described in the legend of the original Extended Data Figure 2 (now Extended Data Figure 1). We have now updated the legend of Figure 1 (line 738-745) to reflect this this.

With regards to multiple hypothesis testing, we had not corrected for this in our original submission as we had specifically set out to investigate a possible association between telomere length and SF-mutant CH, rather than search for such an association across multiple CH subtypes. Moreover, several groups including ours had previously used genome-wide association studies (GWAS) and MR to show that common types of CH, including *DNMT3A*-CH, *TET2*-CH, overall CH, *JAK2*-CH and mCA are associated with germline variants associated with longer telomeres⁸⁻¹³. In fact, the association of long telomeres with most types of CH is so well established that an enrichment of somatic mutations amongst individuals with *TERT* polymorphisms linked to longer telomeres is accepted as evidence that these mutations are CH drivers¹⁴. In this light, we did not view our analysis as one setting out to examine multiple hypothesis, but one setting out specifically to determine whether SF-mutant CH is associated with inheritance of short telomeres. The inclusion of other types of CH in our study was used to verify our method and to contrast SF-mutant CH with other forms of CH.

Nevertheless, as multiple reviewers raised this point, we have now performed multiple hypothesis correction. To do this, we applied a false discovery rate (FDR) control using the Benjamini-Hochberg procedure to account for comparisons. After this adjustment, most significant comparisons remained significant in Figure 1 and 2, with the exception of *CBL* in panel 1b (FDR-adjusted $P = 0.052$). We have included the FDR-adjusted P values as an additional column in Supplementary Tables 1-10, modified figures to

highlight statistical significance with and without adjustment for multiple testing and reference this in figure legends and methods (lines 472-473, 739-741, 749-751, 761-762, 771).

1.7. Fig 1c - for the sake of clarity, would be valuable to group the "Short" (splicing and and "Long" driver genes under the short and long headers

Thank you. We agree that this would make the message clearer and have now amended Figure 1c accordingly.

1.8. Fig 2a - what was the statistical cutoff here? $P=0.05$? Should account for multiple hypothesis

We thank the reviewer for noting this omission. We have updated Figure 2 legend to state the statistical cutoff used ($P<0.05$) (lines 761-762, 771). Please see response to comment 1.6 for a discussion of adjustments for multiple testing.

1.9. I would temper this claim as a heading: "Splicing factor gene mutations enable cells to maintain their telomere length across cell divisions." Perhaps "Clones from splicing factor driver mutations have increased telomere length compared to those with non-splicing driver mutation" might be more fitting.

We thank the reviewer for their suggestion to temper this claim. We have now changed the relevant heading to: "Splicing factor-mutant cells have longer telomeres than cells lacking such mutations" (line 183).

1.10. lines 203-214 "SRSF2-first" and "TET2-first" imply a temporal component which is not measured. How is which mutation occurs 'first' being established? Do any of these individuals have concurrent loss of heterozygosity?

We classified individuals as "SRSF2-first" or "TET2-first" on the basis that the mutation with the highest VAF would have arisen first, with the second mutation being subclonal. Whilst we cannot rule out the possibility that the two mutations mark separate clones, this is highly unlikely to occur by chance for these two genes. In fact, considering the frequency of *TET2* mutations ($n=4284$) and *SRSF2* mutations ($n=373$) CH amongst 454,098 individuals in the UKB, we expect only 4 instances of such a chance co-occurrence. In additional support of the premise that *SRSF2* and *TET2* mutations occurred in the same cell, the VAF of both mutations was almost always large (>0.1) and, in 10/34 individuals, the sum of the VAFs of the two mutations exceeded 0.5, precluding the possibility that they marked separate clones.

The reviewer correctly notes that these observations could also be confounded by loss of heterozygosity (LOH) events. However, in the case of *SRSF2* mutations, LOH or homozygosity is thought to be detrimental and has not been observed in humans^{15,16},

mirroring the fact that mice with homozygous hematopoietic-specific *SRSF2*-P95H mutations develop bone marrow aplasia¹⁷. *TET2* mutations can exhibit LOH, and this could increase the estimated VAF and cause some individuals to be incorrectly classified as *TET2*-first. We already ruled out one such individual from our analysis as they had a *TET2* mutation at a VAF of 80%. We had only one other instance of a *TET2* VAF above 50% (54.1%). Out of 17 *TET2*-first individuals, 6 had a *TET2*-mutant VAF at least two times higher than the *SRSF2*-mutant VAF and would be correctly classified as *TET2*-first even in the presence of an LOH event. Of the 11 remaining individuals, the likelihood that a significant proportion are incorrectly classified due to concurrent LOH at the *TET2* locus is low and unlikely to confound our findings. **It is also important to note that, if the timing of mutation acquisition was miscategorized for some individuals, this would have acted to reduce the likelihood of detecting the significant difference we observed between the “*TET2*-first” and “*SRSF2*-first” groups. We have now added an explanation of the above in Methods (lines 493-499).**

1.11. In Figure 3E/F - how do the authors explain the observation that the *SF3B1* mutated cells have the same telomere length as the WT cells?

We hypothesize that CH-associated mutations in *SF3B1* (as well as *SRSF2* and *U2AF1*) may promote telomere maintenance, enabling mutant HSCs to continue dividing whilst avoiding replicative senescence. This can drive clonal expansion when telomere-associated replicative senescence affects the majority of HSCs, as could happen in some elderly individuals with globally shortened telomeres (mirroring what is seen in TBDs). However, we also provide evidence that splicing factor mutations can also act to rescue the fitness of a highly fit clone whose expansion is curtailed by telomere attrition, as we report for individuals with *TET2/SRSF2* double-mutant myeloid disorders (Figure 4). A possible explanation for the lack of a difference in telomere lengths between *SF3B1*-mutant and *SF3B1*-WT cells in Figure 3E/F is that the clone was expanding due to a different/unknown driver or as a “driverless” clone (leading to telomere shortening), with the *SF3B1* mutation affording telomere protection and restoring clonal fitness within this clone. In this context, the telomere length of splicing factor-mutant cells need not be longer than unmutated cells, but merely long enough to avoid replicative senescence and thus maintain the fitness of the clone. In keeping with this, in Fig 3E/F we observe that the median telomere length of *SF3B1*-mutated colonies, but not *SF3B1*-WT colonies, increases between timepoints 1 and 2 to resemble the telomere length of the *SF3B1*-WT colonies (which are not clonally expanded), although this trend does not reach statistical significance ($p=0.27$). Figure 3A may be showing a similar phenomenon, where the acquisition of a *U2AF1* mutation by a HSC belonging to an apparently driverless clone, restores clonal fitness and drives further clonal expansion (evidenced by a burst of coalescences on the phylogenetic tree) without evidence of telomere attrition.

1.12. In Fig 4c - the unsorted ratio of *SRSF2* to *TET2* was 1.0, and discordant from the ratio of the VAFs of *SRSF2* to *TET2*. In the other plots, the unsorted bar roughly approximates the ratio of VAFs. How do the authors account for this?

We thank the reviewer for raising this point, which we presume refers to Fig 4E (as Fig 4C relates to a case of *SF3B1*-mutant CLL). In 4E (4F in updated figure), the clinically reported VAFs for *SRSF2* and *TET2* shown above the plot do indeed appear discordant with the ratio of *SRSF2*-mutant to *TET2*-mutant cells derived by Sanger sequencing in the unsorted population. This is likely to be a result of two factors i) clinical NGS is typically performed on fresh total bone marrow (BM) aspirate, whereas we used cryopreserved BM mononuclear cells (MNCs) for flowFISH. It is possible that differences in the proportion of *SRSF2*-mutant cells within these fractions may lead to discordance in VAF ratios when comparing total BM and BM MNCs. ii) Sanger sequencing performs very well for relative quantification of alternate alleles, but is not typically used for absolute VAF quantification, which is best done by NGS. Unfortunately, the *SRSF2*-V18 position mutated in the individual shown in 4E (4F in updated figure) is not captured by our targeted myeloid gene NGS panel. However, we were able to orthogonally validate our Sanger sequencing data with NGS for another two individuals (updated Extended Data Figure 13) and saw similar enrichment of *SRSF2*-mutant cells in the telomere^{>90%} fraction, relative to the telomere^{<10%} fraction, using both modalities. This confirms that Sanger sequencing provides robust relative quantification of VAFs and justifies its use to compare the *SRSF2/TET2*-mutant ratio between different cell fractions from the same individual. We now explain that the clinical VAF can in some cases differ from the BM MNC VAF due to different cellular composition in the Figure 4 legend (lines 810-813).

1.13 Line 248 - please clarify that "at the lower end of the spectrum" means "lower genetically predicted telomere length"

Thank you. We have amended the text to reference individuals with lower genetically predicted telomere length (now line 297).

1.14. Figure quality of Figure 4 is low resolution

We thank the reviewer for noticing this and have improved the resolution in the revised Figure 4.

1.15. If Figure 5 is a conceptual hypothesis - it perhaps may be better as an extended data figure. The data supporting 5D and 5E may not be the most robust. 5D is from one paper that characterized 17 patients and 21 controls. The data supporting 5E is from 1 published paper of 42 patients and 2 ASH abstracts from 2021 that have not yet been published.

We appreciate this comment and indeed the figure represents a hypothesis. However, since our original submission, a paper describing the landscape of CH in 207 TBD patients was published⁴ and its findings strongly endorse what we propose in Figure 5E. Also, as part of our revisions we identified *TERT* promoter mutations in UKB participants with significantly lower telomere PRS (compared to no-CH controls), at similar frequencies and with a similar age-related prevalence to splicing factor CH (see comment 3.6). Therefore, we now have stronger support for our hypothesis, which, in our view, justifies presenting it in a main figure as an overall conclusion of our manuscript. We hope our reviewer concurs with this approach considering these new findings.

Reviewer #2:

Remarks to the Author:

McLoughlin et al. describe the relationship between clonal hematopoiesis (CH) in specific CH driver genes and telomere length. As most prior studies investigated CH-TL associations at the aggregate level, this study provides novel insights into CH driven by splicing factor genes suggesting telomere attrition could select for CH clones harboring splicing factor gene mutations. The study data is from both a large population and specific individuals with clinical information and repeated samples. Approaches and methods used are appropriate, conclusions are supported by the data and the manuscript clearly presents the findings. I have a few comments worth consideration for further strengthening this already terrific investigation:

We thank our reviewer for their positive overall assessment of our manuscript.

2.1. Details on CH driver and mCA calling were sparse. As small changes in calling approach or filtering methods could have a substantial impact on study results I recommend a more detailed description of the methods used to facilitate readers in evaluating the approach and promote reproducibility of future investigations.

Thank you. We have now expanded the description of our CH driver calling in the methods (lines 424-443) and added a detailed description as Supplementary Note 3. mCA calls were obtained from Loh et al.¹³ and this is described in lines 444-448.

We also refined our filtering approach by additionally excluding a small number of individuals with a prevalent diagnosis of hematological malignancy (n=131) from analysis containing measured LTL. The majority of these individuals had mutations in *JAK2* (n=61), *CALR* (n=19), or *DNMT3A* (n=22) and removing them did not alter the significance of any of the observations in Figure 1 or 2. However, this change in filtering did alter the P value reported in Extended Data Figure 9 (lines 244, 990-1003) and the significance of some of the associations with myeloid malignancy reported in the revised Extended Data Figure 1 (lines 116-118 and 862-868). More specifically, the association between LTL-PRS and AML no longer reaches statistical significance, whereas the association between LTL and MPN now reaches significance. We have updated the manuscript to reflect these changes (lines 113-115).

2.2. The Codd et al investigation used to calculate the telomere length PRS and perform the Mendelian randomization was performed in the UK Biobank. As such the same dataset was used for GWAS discovery and phenotype testing, potentially resulting in overfitting or bias. For example, CH clones in UK Biobank could cause deviations in the UK Biobank TL measures and impact TL loci identified. This in turn could impact results between CH and TL in the current analysis. A sensitivity analysis employing the Taub et al TL GWAS in TOPMed showing very similar overall results with CH would help minimize any concerns. At minimum, I suggest a discussion of the current TL loci being discovered in the same population as a study limitation.

We are grateful to the reviewer for this important suggestion. Applying the Taub et al. (2022) PRS to the UK Biobank was also suggested by Reviewer 1 and we have now

performed this analysis, which corroborates our findings. A detailed response to this point can also be found under our response to comment 1.3 above and is included our revised manuscript (Extended Data Figure 4 and text lines 135-142)

2.3. The population-based analyses employ multiple tests of different CH clones, but use 0.05 as the statistical significance threshold. I recommend a more conservative threshold that factors in the number of tests performed.

We thank the reviewer for raising this point. Similar concerns regarding multiple testing were also raised by another reviewer and have now been addressed – please see comment 1.6.

2.4. It is unclear how individuals were selected for the WGS analysis of mutation carriers. Were these individuals chosen from multiple individuals with splicing factor gene mutations, or were they the only individuals with these alterations? Please better describe selection criteria and include a brief description of the mutation frequencies in the clinical sample from which the individuals for phylogenetic analyses were drawn.

We thank the reviewer for this point. The three individuals used for WGS studies (Figure 3) were the only ones with SF-mutant CH that we sequenced with sufficient colony numbers for analysis. Below is a description of exactly how we proceeded:

We selected individuals for colony WGS specifically because they harbored splicing factor gene mutations. PD34493 was recruited from the SardiNIA cohort¹⁸ and reported in a previous study from our group¹⁹. Another two individuals from the latter study could not be used here as they did not have sufficient numbers of either SF-mutant colonies (PD41305 had only one *SRSF2*-mutant colony) or WT colonies (PD42176 had 82 colonies with *SF3B1* K666N, but only one WT colony)¹⁹. For this reason, we sequenced one more individual from the SardiNIA cohort, PD41082, specifically for this study. We also sequenced PD48499, a local Cambridge patient/volunteer who was selected for WGS because he harbored the *SF3B1*-K700E mutation and had longitudinal samples available separated by several years. Our group has not performed WGS on any other individuals with SF-mutant CH.

We now state in the Methods, under “Construction of Phylogenetic Trees from Whole Genome Sequencing of Hematopoietic Cell Colonies”, that the individuals used for WGS analyses were chosen on the basis that they had SF-mutant CH, with no other selection criteria. We also now list the mutations detected in bulk peripheral blood (PD34493, PD41082) or bone marrow aspirate (PD48499) at time of sampling (lines 525-538).

2.5. While the relationship between senescent phenotypes and PPM1D is suggested as a mechanism for the relationship between PPM1D mutations and telomere length, the relationship between PPM1D and telomere length is not explored outside of the

population-level data like the splicing factor mutations. Were samples available for analysis? Were findings null?

At the initiation of this study, we set out to investigate the basis for the strong association of SF-mutant CH with advanced age and this has remained the focus of our paper. We agree that the increased risk of developing *PPM1D*-mutant CH in individuals with lower LTL-PRS is an important additional finding. We proposed a plausible explanation that takes account of what is known about the DNA damage response (DDR)-inhibitory function of normal *PPM1D*²⁰, the ability of mutant *PPM1D* to augment this²¹, and the fact that, in our study, individuals with large VAF *PPM1D*-mutant CH have shorter telomeres than those with small clones (i.e. supporting the hypothesis that mutant *PPM1D* inhibits DDR signaling rather than providing telomere length protection). It is also noteworthy that *PPM1D*-CH was recently found to be the most common form of rescue CH in patients with TBDs caused by germline mutations in *TERT* or *TERC*⁴, where replicative senescence of HSCs leads to aplastic anemia²².

However, we felt that additional experimental work to investigate the basis of the relationship between *PPM1D*-mutant CH and telomere length was outside the scope of this manuscript and its focus on splicing factor-mutant CH. Moreover, the types of experiments outlined in Figures 3 and 4 would have been very challenging in the context of *PPM1D*-mutant CH: First, unlike SF-mutant CH, we had no access to existing colony WGS data or to cryopreserved viable samples from individuals with *PPM1D*-mutant CH, which would have precluded us from replicating the equivalent work shown in Figure 3 for SF-mutant CH. Second, we also had no access to CH or MN cases with *PPM1D* plus another CH mutation, such as *TET2*, as such cases are rare, thus precluding the type of analyses described in Figure 4.

2.6. Analyses of cases with double mutant CH assume the same CH clone has both mutations. This might not always be the case and should be stated.

We thank the reviewer for this valid comment. Similar points regarding the assumptions made in the double-mutant individuals were raised by other reviewers and a justification can be found under comment 1.10. We have now added an explanation/justification for the above in Methods (lines 493-499).

2.7. In Figure 3 and Extended Figure 3, consider moving the mutation label to a location that does not cover the branch points on the dendrogram. It is difficult to see the estimated age at divergence when the label is covering the branch points.

We thank the reviewer for this suggestion. We have reformatted the labels in Figure 3 and the original Extended Data Figure 3 (now Extended Data Figure 6) to make the coalescences clearly visible. We have also colored the branches of clades with splicing factor mutations to highlight the estimated timing of mutation acquisition.

2.8. There are several discrepancies in the PD identifiers and descriptions of the WGS sequenced individuals in the Methods (pages 10-11) and Figure 3. In the Methods, there is a male aged 83.8 years with a SF3B1-mutant CH clone denoted with (PD48499), but

this is the label for the male with CCUS to MDS. In Figure 3 panel a, the individual with PD34493 is listed as male, while page 11 describes PD34493 as female. Please correct.

We thank the reviewer for noting these discrepancies, which we have now corrected in the text. For clarity, PD34493 is an 83.8-year-old male, PD41082 is a 73.9-year-old female, and PD48499 is a 50.2/53.8-year-old male (as detailed in our original Figure 3). The text now reflects this (lines 525-538).

2.9. The abbreviation HSPCs was not defined in text.

Thank you for noticing this oversight. We now define the abbreviation HSPC as hematopoietic stem and progenitor cell (HSPC) (line 194).

Reviewer #3:

Remarks to the Author:

In their manuscript McLoughlin et al., study the relationship between telomere length and clonal hematopoiesis showing that while most CH mutations are associated with longer telomere lengths (as has been previously shown) specific mutations including PPM1D and splicing mutations are more common among individuals with shorter telomere lengths. As would be predicted based on the finding of an excess of PPM1D and spliceosome CH among individuals with telomere biology disorders, they demonstrate that splicing factor CH appears to protect against telomere attrition. The absence of mechanistic data limits extrapolation to whether this might be harnessed therapeutically. Overall though the analysis was well executed and novel, the manuscript well-written with appropriate and interesting conclusions being drawn.

We thank our reviewer for the positive overall assessment of our manuscript.

Major comments:

3.1. They do not present any mechanistic data regarding how spliceosome and PPM1D mutations might protect against telomere attrition and whether this plays a role in leukemogenesis. Because of this, their data does not directly support statements that their findings offer clear therapeutic approaches in splicing factor mutant myeloid neoplasms. This includes the last statement in the abstract, introduction and discussion. These statements should be modified emphasizing that future work is needed to determine whether this might have therapeutic relevance.

We thank the reviewer for this comment and agree that our work does not identify specific therapeutic approaches. However, we believe that it is reasonable to suggest that investigating the mechanistic basis of our observation may open new therapeutic approaches. Taking on board our reviewer's point, we have amended the indicated sections of the text to reflect the need for further mechanistic studies to determine the therapeutic relevance of our findings (lines 55-56, 90-91 and 348).

3.2. It is interesting in Figure 1b and c, that there is no association between TP53 CH and telomere length. Is there an association between other common DDR CH genes and telomere length including ATM and CHEK2? Similarly, it would be interesting to compare

the strength of the association between telomere length and U2AF1. This could provide some additional biologic insight into differences between genes of the same class.

We thank the reviewer for this reasoned comment. We set out to investigate a possible link between telomere maintenance and the age-related rise in prevalence of SF-mutant CH, whilst confirming previously reported associations between LTL-PRS and other forms of CH. For this reason, we restricted our analysis to the most common CH/mCA subtypes, particularly as we would have been underpowered to investigate uncommon/rare forms of CH (please also see response to comment 3.9). With respect to *CHEK2*, we had not investigated this gene since mutations are uncommon in the general population and, with the exception of truncating variants²³, it is difficult to confidently identify driver/pathogenic mutations due to the lack of sufficient cases/recurrence in the literature. Also, we did not include *ATM*, as *ATM* mutations usually occur in a B-lymphoid cell, rather than an HSC and therefore do not represent conventional (myeloid) CH. This is mirrored in the fact that *ATM* mutations are common in chronic lymphocytic leukemia (CLL)²⁴ but not in MN^{15,16}.

Nevertheless, to address this comment, we have now extended our PRS analyses to include mutations in *CHEK2* (truncating only, n=93) and *ATM* (n=159), as well as another DDR gene, *BRCC3* (n=104) and a few other rare drivers of CH (**Figure R5**). Given the lack of significant association with PRS for any of these additional forms of CH, and as they are not central to our focus on SF-mutant CH, **we have opted to not include these results in our paper.**

Figure R5. PRS distribution amongst carriers of an extended panel of CH mutations
Upper panel: Distribution of UKB-derived LTL-PRS scores amongst UKB participants with or without different types of CH. White dots and black boxes mark the LTL-PRS median and interquartile range (IQR), respectively. Whiskers extend to the lowest and highest

data points within $Q1-1.5 \times IQR$ and $Q3+1.5 \times IQR$ where $Q1$ and $Q3$ represent the first and third quartiles, respectively. The control group includes all participants without any CH mutation or mCA. Lower panel: Odds ratio (OR) of developing different CH subtypes per 1 standard deviation increase in LTL-PRS. Bars represent the 95% CI for each OR. ORs for genes with an unadjusted $P < 0.05$ are depicted in dark grey with an asterisk (*) and those with an unadjusted $P \geq 0.05$ in light grey. The number of individuals in each group is indicated in brackets next to the gene names.

With regards to *U2AF1*, we had not included this gene because the total number of mutant cases in the UKB was below 100 and because its annotation in GRCh38 is erroneous such that mutations cannot be detected by standard pipelines such as Mutect2²⁵, which we used to detect other mutations in this study. However, we agree that inclusion of *U2AF1* is important and have now included *U2AF1*-mutant cases (n=82) identified using a pile-up as previously described²⁶. We repeated our analysis in the UKB with the inclusion of *U2AF1*-CH, observing a non-significant trend towards lower LTL-PRS (FDR-adjusted $P = 0.157$) (revised Figure 1). However, we did detect a significant association between *U2AF1*-CH and lower LTL-PRS when analyzing the All of Us cohort, which we used to validate our UKB findings (see response to comment 1.1, **Figure R1**). **These *U2AF1* analyses are now included in the revised manuscript (Figure 1-2 & Extended Data Figure 3-5) and discussed in relevant parts of the text (line 128-142).**

3.3. In their analysis of individuals with double mutant CH, they state that they used the VAF to determine mutation order. However, VAF is both a reflection of the time of mutation acquisition and mutation fitness. If the authors restricted their double hit analysis to individuals in whom mutant clones must be co-occurring based on the pigeonhole principle (using the cell fraction), VAF can be a proxy for order. However, if the clones are independent VAF cannot be used as a proxy. This should be clarified.

We thank the reviewer for this comment and agree that this assumption should have been clarified. Similar points have been made by other reviewers and a detailed response can be found under our response to comment 1.10. **We have added additional detail to the methods to clarify the assumptions surrounding double-mutant individuals and the potential caveats of this approach (lines 493-499).**

3.4. The authors state in the discussion that only 32 of the 640 cases of PPM1D-CH had documented history of prior chemotherapy. Is complete ascertainment of chemotherapy available in the UKBB including both oral and IV administration? Is radiation therapy data available? This would be important in determining the true number of PPM1D carriers that have no prior chemotherapy/radiation therapy.

We thank the reviewer for this comment and have now investigated whether any other carriers of *PPM1D*-mutant CH in the UKB were likely to have been exposed to cytotoxic therapy, but had been missed due to incomplete annotation.

We have identified a total of 647 cases of *PPM1D*-CH, a slightly higher number than previously stated due to the inclusion of an additional 7 individuals with multiple *PPM1D*-mutated clones, which had previously been filtered in error. Of the 647 individuals with *PPM1D*-CH, 32 had prior chemotherapy and 5 had prior radiotherapy. Given that cancers are by far the most common indication for cytotoxic therapy, we also searched if any of the 647 *PPM1D*-CH carriers had previously developed any malignant neoplasm that may have required such treatment (all cancers except non-melanoma skin cancer) and identified 98 such cases. The total number of cases after amalgamating all 3 categories (chemotherapy + radiotherapy + malignant neoplasm) was 100. This reassures both that the annotation of cytotoxic therapy in the UKB is of generally high quality and that, even if some of these individuals with malignant neoplasms had undocumented therapies, the majority of *PPM1D*-CH carriers had not received chemotherapy/radiation therapy. **We have now updated these numbers in the manuscript text (lines 318-322).**

3.5. In Figure 4, are there differences in proportion of mutant cells in the telomere high and low populations statistically significant? It is unclear whether there is formal hypothesis testing performed here.

We agree that formal hypothesis testing would strengthen the conclusions derived from the data presented in Figure 4. To do this, we analyzed data across the 3 individuals with *SRSF2/TET2* co-mutations and compared the proportion of splicing factor-mutant cells (expressed as the ratio of *SRSF2*-mutant VAF over *TET2*-mutant VAF) in MNC fractions that had been sorted by percentile telomere length (**Figure R6**). The VAFs used in the combined analysis were derived using Sanger sequencing, since the *SRSF2*-V18 position mutated in patient 8057 (Figure R6c) is not captured in our targeted myeloid NGS panel and the remaining two individuals with NGS data are insufficient for formal statistical hypothesis testing. Analysis of Sanger sequencing data revealed a significant enrichment of *SRSF2*-mutant cells in both the telomere^{>90%} ($P = 0.0059$) and telomere^{66-90%} ($P=0.0326$), relative to the telomere^{<10%} fraction (**Figure R6d**). We thank the reviewer for this suggestion and **have now included all Sanger sequencing data and combined analysis within an updated Figure 4, whilst the NGS data is now included as Extended Data Figure 13. We also discuss these findings in the text (line 263).**

Figure R6. Enrichment of splicing factor-mutant cells in cells sorted by percentile telomere length.

a-c, Ratio of *SRSF2*-mutant and *TET2*-mutant cells in PBMCs sorted by percentile telomere length from patients with AML and CMML with VAF quantified by Sanger sequencing. Absence of bars in 'Telomere<10%' (a-c) and 'Telomere10-33%' (c) indicates that no *SRSF2*-mutant cells were detected within those fractions. d, Combined analysis of data shown in a-c. Each point represents the ratio of *SRSF2*-mutant to *TET2*-mutant cells in a single individual. P values derived by one-way Anova with Tukey's multiple comparison test. Only adjusted P values below the significance threshold ($P < 0.05$) are shown.

3.6. The authors note that they did not observe CH in the *TERT* promoter as has been observed in TBD. Did they query the WGS of the UKBB for this region? Only analysis of the WES data is mentioned

We are grateful to our reviewer for this insightful and, in the end, very helpful comment. They correctly note that we did not detect *TERT* promoter (*TERTp*) mutations in WES data, which only gives shallow coverage of the *TERTp* region. We had analyzed WES data as WGS data were not available across the UKB at the time of our original analysis.

We have now gone on and performed analyses of WGS data as suggested. Relative to WES, WGS data in the UKB is substantially lower depth (typically ~34x). As a result, the use of stringent somatic mutation calling pipelines such as Mutect2 on these data would filter out many real mutations and reduce our sensitivity to detect somatic *TERTp* mutations. To circumvent this, we performed pileup analysis across the entire *TERT* promoter region in the UKB, including the three known *TERTp* mutation hotspots reported in cancer and blood cells of TBD patients, namely chr5:1295046:T:G NM_198253.3:c.-57A>C, chr5:1295113:G:A: NM_198253.3:c.-124C>T and chr5:1295135:G:A: NM_198253.3:c.-146C>T⁴. After implementing several filters to remove sequencing errors and enrich for 'real' somatic variants (see Supplementary Note 2), this yielded a total of 151 individuals within the UKB with a *TERTp* mutation at the three known hotspots. We excluded 3 individuals with a prevalent diagnosis of hematological malignancy, resulting in a total of 148 individuals with putative *TERTp*-CH in the UKB.

We went on to perform extensive orthogonal analyses that provided robust evidence that these changes represented real somatic mutations (See Methods & Supplementary Note 2). Importantly, the hotspot mutations were enriched in individuals with lower telomere PRS, had a VAF distribution that differed substantially from erroneous calls on non-hotspot sites on the *TERTp*, and had a prevalence and age-/sex-distributions (median age: 65 years, range: 45-70 years, sex: 76.4% male) very similar to SF-mutant CH (**Figure R7**). Further analysis of these 148 *TERTp*-CH cases revealed that 4 individuals went on to develop incident myeloid neoplasia.

To benchmark the veracity of our pileup approach and contrast the prevalence of *TERTp*-CH relative to other known drivers, we also performed pileup on WGS data at several splicing factor mutation hotspots (*SF3B1*-K700, *SF3B1*-K666, *SF3B1*-R625, *SRSF2*-P95,

U2AF1-S34, *U2AF1-Q157*) and filtered them in the same manner. We identified a total of 126 SF-mutant CH cases, the majority of which (60%) had also been identified by performing Mutect2 on WES data. We go on to provide orthogonal evidence that mutations identified only by WGS pileup were associated with known changes in complete blood indices seen in SF-mutant CH, corroborating their veracity. This suggests that pileup at known hotspots using WGS data from the UKB reliably identifies somatic variants, albeit at reduced sensitivity compared to the deeper-sequenced WES data. Our WGS pileup analysis also enabled us to compare the relative prevalence of *TERTp*-CH and SF-CH in the UKB identified using WGS data.

The finding of *TERTp*-CH in the UKB strongly endorses the main premise of our paper, namely that telomere attrition becomes an instrument for clonal selection in aging hematopoiesis and leukemogenesis. We again thank the reviewer for this suggestion, as this led us to an important novel finding that corroborates our hypothesis regarding the role of telomere attrition in ageing hematopoiesis and CH. We have included these data in the text (272-288), Extended Data Figure 15, Supplementary Note 2 and also refer to it in the revised Figure 5.

Figure R7. Age-related prevalence of *SF3B1*, *SRSF2* and *TERTp*-CH amongst UK Biobank participants.

The prevalence of *SF3B1* and *SRSF2* hotspot mutations increases sharply after the sixth decade of life. The sensitivity to detect CH clones is a function of sequencing depth and modality. Since *TERTp* is not adequately captured by WES, we identified *TERTp*-CH (red line) from blood DNA WGS data of (methodology described in Supplemental Note 2). When we call *SF3B1* and *SRSF2* hotspot mutations from WGS using the same filtering method, the prevalence of *TERTp*-CH is comparable to the combined prevalence of

SF3B1-CH (light green line) and *SRSF2*-CH (light blue line) and exhibits a similar association with advancing age.

Minor:

3.7. In the introduction, the association between SNPs at the TERT locus and CH are described as “powerful”. Qualifiers on strength are normally used in regard to the OR, not the p-value. But in the Kar et al paper the top hit rs2853677 was modestly associated with CH with an OR of 1.24.

We thank the reviewer for this clarification and have replaced the term “powerful” with “significant” to better reflect the findings from Kar et al. (2022) (line 71).

3.8. The results they show in Figure 3F for the patient with longitudinal sampling are not significant. I would advise emphasizing this on line 186-7 by emphasizing this was a “non-significant trend”.

We thank the reviewer for their suggestion and have amended the text to clarify that this finding did not reach statistical significance (line 200).

3.9. In the methods section they state that “People with mutations in less frequently mutated genes...were excluded”. What is the rationale for excluding individuals with CH in less common genes?

As mentioned earlier, the focus of our study was SF-mutant CH, with other types of CH included as comparators (please also see repose to comment 1.6). For this reason, we only wanted to include CH subtypes with sufficient cases to detect associations with LTL-PRS or refute them with some confidence. A threshold of 100 cases was selected to ensure sufficient power to detect significant associations between LTL-PRS and CH subtype risk. More specifically, in a case-control study with a continuous predictor and binary outcome, with 100 cases and approximately 450,000 controls, we have 80% power at an alpha of 0.05 to detect protective odds ratios of at least 0.75 and risk odds ratios up to 1.32 per standard deviation change in the continuous predictor. Considering that we are now adjusting for 18 multiple comparisons, this further lowers the alpha and leaves a power of 80% to detect protective and risk ORs of approximately 0.68 and 1.47, respectively. Given the lack of power to confidently detect and comment on subtle effects of either measured telomere length or LTL-PRS in CH driven by genes mutated in fewer than 100 cases, we believe that it is prudent to exclude these genes from our analyses, rather than report potentially false negative results (see also our remark about *CHEK2* in our response to comment 3.2). Moreover, including these groups in our analyses would have required more stringent correction for multiple testing, “penalizing” our analyses for investigating genes that we did not set out to study. This would have further restricted our power to detect associations of the more prevalent forms of CH that we were investigating. These power calculations were performed using the “samplesizelogisticcasecontrol” package in R version 2.0.2 and a brief summary is now included in Methods (lines 431-438).

Please note that we have included *U2AF1* in our analyses despite having slightly less cases than 100 (n=82), as this was specifically requested by one of our reviewers and also because the gene forms part of our core hypothesis (i.e. that telomere maintenance may underlie the unusual age distribution of sporadic splicing factor-mutant CH – see line 104-105). The recent identification of *U2AF1* mutations in TBD patients⁴ adds to the rationale for including this gene. This is also explained in Methods (lines 441-442).

3.10. The code used to generate all main and extended/supplemental datasets should be included.

We agree that all code used to generate main and extended datasets should be included. All code has now been uploaded to GitHub and links have been added to the code availability statement (line 393-408).

References

- 1 Wen S, Kuri-Morales P, Hu F *et al.* Comparative analysis of 136,401 Admixed Americans and 419,228 Europeans reveals ancestry-specific genetic determinants of clonal haematopoiesis. *medRxiv (Nature Genetics, accepted)*, 2024.2002.2007.24302442 (2024).
- 2 Ziyatdinov A, Torres J, Alegre-Díaz J *et al.* Genotyping, sequencing and analysis of 140,000 adults from Mexico City. *Nature* **622**, 784-793 (2023).
- 3 Investigators TAOURP. The “All of Us” Research Program. *New England Journal of Medicine* **381**, 668-676 (2019).
- 4 Gutierrez-Rodriguez F, Groarke EM, Thongon N *et al.* Clonal landscape and clinical outcomes of telomere biology disorders: somatic rescue and cancer mutations. *Blood* **144**, 2402-2416 (2024).
- 5 Codd V, Wang Q, Allara E *et al.* Polygenic basis and biomedical consequences of telomere length variation. *Nat Genet* **53**, 1425-1433 (2021).
- 6 Taub MA, Conomos MP, Keener R *et al.* Genetic determinants of telomere length from 109,122 ancestrally diverse whole-genome sequences in TOPMed. *Cell Genom* **2** (2022).
- 7 Cheloor Kovilakam S, Gu M, Dunn WG *et al.* Prevalence and significance of DDX41 gene variants in the general population. *Blood* **142**, 1185-1192 (2023).
- 8 Bick AG, Weinstock JS, Nandakumar SK *et al.* Inherited causes of clonal haematopoiesis in 97,691 whole genomes. *Nature* **586**, 763-768 (2020).
- 9 Kar SP, Quiros PM, Gu M *et al.* Genome-wide analyses of 200,453 individuals yield new insights into the causes and consequences of clonal hematopoiesis. *Nat Genet* **54**, 1155-1166 (2022).
- 10 Kessler MD, Damask A, O’Keeffe S *et al.* Common and rare variant associations with clonal haematopoiesis phenotypes. *Nature* **612**, 301-309 (2022).
- 11 Nakao T, Bick AG, Taub MA *et al.* Mendelian randomization supports bidirectional causality between telomere length and clonal hematopoiesis of indeterminate potential. *Sci Adv* **8**, eabl6579 (2022).

- 12 Tapper W, Jones AV, Kralovics R *et al.* Genetic variation at MECOM, TERT, JAK2 and HBS1L-MYB predisposes to myeloproliferative neoplasms. *Nature Communications* **6**, 6691 (2015).
- 13 Loh PR, Genovese G, Handsaker RE *et al.* Insights into clonal haematopoiesis from 8,342 mosaic chromosomal alterations. *Nature* **559**, 350-355 (2018).
- 14 Vlasschaert C, Mack T, Heimlich JB *et al.* A practical approach to curate clonal hematopoiesis of indeterminate potential in human genetic data sets. *Blood* **141**, 2214-2223 (2023).
- 15 Papaemmanuil E, Gerstung M, Malcovati L *et al.* Clinical and biological implications of driver mutations in myelodysplastic syndromes. *Blood* **122**, 3616-3627; quiz 3699 (2013).
- 16 Papaemmanuil E, Gerstung M, Bullinger L *et al.* Genomic Classification and Prognosis in Acute Myeloid Leukemia. *N Engl J Med* **374**, 2209-2221 (2016).
- 17 Kim E, Ilagan JO, Liang Y *et al.* SRSF2 Mutations Contribute to Myelodysplasia by Mutant-Specific Effects on Exon Recognition. *Cancer Cell* **27**, 617-630 (2015).
- 18 Orrù V, Steri M, Sole G *et al.* Genetic Variants Regulating Immune Cell Levels in Health and Disease. *Cell* **155**, 242-256 (2013).
- 19 Fabre MA, de Almeida JG, Fiorillo E *et al.* The longitudinal dynamics and natural history of clonal haematopoiesis. *Nature* **606**, 335-342 (2022).
- 20 Gräf JF, Mikicic I, Ping X *et al.* Substrate spectrum of PPM1D in the cellular response to DNA double-strand breaks. *iScience* **25**, 104892 (2022).
- 21 Zhang L, Hsu JI & Goodell MA. PPM1D in Solid and Hematologic Malignancies: Friend and Foe? *Mol Cancer Res* **20**, 1365-1378 (2022).
- 22 Kam MLW, Nguyen TTT & Ngeow JYY. Telomere biology disorders. *npj Genomic Medicine* **6**, 36 (2021).
- 23 Bolton KL, Ptashkin RN, Gao T *et al.* Cancer therapy shapes the fitness landscape of clonal hematopoiesis. *Nat Genet* **52**, 1219-1226 (2020).
- 24 Schaffner C, Stilgenbauer S, Rappold GA *et al.* Somatic ATM Mutations Indicate a Pathogenic Role of ATM in B-Cell Chronic Lymphocytic Leukemia. *Blood* **94**, 748-753 (1999).
- 25 Miller CA, Walker JR, Jensen TL *et al.* Failure to Detect Mutations in U2AF1 due to Changes in the GRCh38 Reference Sequence. *J Mol Diagn* **24**, 219-223 (2022).
- 26 Gu M, Kovilakam SC, Dunn WG *et al.* Multiparameter prediction of myeloid neoplasia risk. *Nat Genet* **55**, 1523-1530 (2023).